# UNDERSTANDING AUGMENTATION-BASED SELF-SUPERVISED REPRESENTATION LEARNING VIA RKHS APPROXIMATION AND REGRESSION

**Runtian Zhai, Bingbin Liu, Andrej Risteski, Zico Kolter, Pradeep Ravikumar**
Carnegie Mellon University
`{rzhai,bingbinl,aristesk,zkolter,pradeepr}@cs.cmu.edu`

## ABSTRACT

Data augmentation is critical to the empirical success of modern self-supervised representation learning, such as contrastive learning and masked language modeling. However, a theoretical understanding of the exact role of augmentation remains limited. Recent work has built the connection between self-supervised learning and the approximation of the top eigenspace of a graph Laplacian operator, suggesting that learning a linear probe atop such representation can be connected to RKHS regression. Building on this insight, this work delves into a statistical analysis of augmentation-based pretraining. Starting from the isometry property, a geometric characterization of the target function given by the augmentation, we disentangle the effects of the model and the augmentation, and prove two generalization bounds that are free of model complexity. Our first bound works for an arbitrary encoder, where the prediction error is decomposed as the sum of an estimation error incurred by fitting a linear probe with RKHS regression, and an approximation error entailed by RKHS approximation. Our second bound specifically addresses the case where the encoder is near-optimal, that is it approximates the top-$d$ eigenspace of the RKHS induced by the augmentation. A key ingredient in our analysis is the *augmentation complexity*, which we use to quantitatively compare different augmentations and analyze their impact on downstream performance.

## 1 INTRODUCTION

It is widely acknowledged that better data augmentation techniques have been a major driving force in many recent breakthroughs in self-supervised representation learning. For example, contrastive learning (Chen et al., 2020) with aggressive random cropping and strong color distortion greatly improves the performance of vision tasks, and masked prediction with random masking and replacement is among the state-of-the-art representation learning methods for both natural language processing (Devlin et al., 2019) and vision (He et al., 2022). Due to their extraordinary empirical successes, developing a rigorous theoretical understanding of augmentation-based representation learning is an important open problem, and is crucial for inventing new, better and principled augmentation techniques.

A key advance was recently made by HaoChen et al. (2021), who analyzed a variant of contrastive representation learning, termed spectral contrastive learning, by connecting it to learning eigenfunctions of a Laplacian operator over a population *augmentation graph*. This has been followed up by multiple works which extended these results to more general contrastive learning approaches (Saunshi et al., 2022; Johnson et al., 2023; Cabannes et al., 2023). However, the generalization guarantees in these works have to explicitly grapple with the function class being used to learn the eigenfunctions (typically deep neural networks in practice). Such a dependency was even deemed necessary by Saunshi et al. (2022) who argued that "function-class-agnostic analysis leads to vacuous guarantees." Consequently, the generalization bounds such as HaoChen et al. (2021, Theorem 4.2) need to depend on the Rademacher or other complexities of this encoder function class, yet finding proper complexity measures for flexible function classes such as neural networks remains an open problem. Moreover, with the effects of the augmentation and the encoder function class intertwined, these analyses cannot discern the exact role of augmentation in self-supervised representation learning.

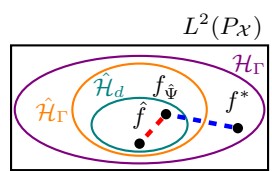

$L^2(P_\mathcal{X})$: Large space of $L^2$ bounded functions, containing the target function $f^*$
Induced RKHS $\mathcal{H}_\Gamma$: Functions with soft invariance to augmentation, $f^* \in \mathcal{H}_\Gamma$
Empirical RKHS $\hat{\mathcal{H}}_\Gamma$ : Obtained with $N$ samples to approximate inaccessible $\mathcal{H}_\Gamma$
Empirical top-$d$ RKHS $\hat{\mathcal{H}}_d$ : Induced by the pretrained $d$-dimensional encoder
Approximation error $\|f^* - f_{\hat{\Psi}}\|$ : Distance from $f^*$ to $\hat{\mathcal{H}}_d$ (Lemma 3, Theorem 2)
Estimation error $\|f_{\hat{\Psi}} - \hat{f}\|$: Entailed by downstream regression (Lemma 2)

Figure 1: Overall RKHS approximation/regression framework illustration and commentary.

In this work, we factor out the effect of the encoder function class completely. Our starting point is the observation that target functions that are "soft invariant" to the augmentation lie in a specific RKHS $\mathcal{H}_\Gamma$ which we call the *(augmentation) induced RKHS*. Meanwhile, the $d$-dimensional feature encoder we pretrain also spans an RKHS, and the closer this RKHS is to the induced RKHS, the better the encoder. Thus, pretraining can be viewed as approximating the induced RKHS with unlabeled data. Then, learning a linear probe atop corresponds to regression on the encoder's RKHS with labeled data. This perspective elucidates what roles the two stages of representation learning are playing:

- **Upstream:** The self-supervised pretraining stage can be viewed as doing RKHS approximation. It learns, with unlabeled data, an encoder $\hat{\Psi}$ whose learned RKHS $\mathcal{H}_{\hat{\Psi}}$ (Eqn. (9)) approximates $\mathcal{H}_\Gamma$.

- **Downstream:** The supervised stage is doing RKHS regression over $\mathcal{H}_{\hat{\Psi}}$ with labeled data.

Figure 1 provides an illustration of this RKHS approximation/regression perspective. The learning performance is measured by the prediction error of the downstream predictor, which decomposes into two parts based on the above framework: the approximation error upstream entailed by RKHS approximation, and the estimation error downstream entailed by RKHS regression.

Thus, using classical function analytic tools, this leads us to our first set of results: For an *arbitrary* pretrained encoder, we provide a generalization bound on the $L^2$ distance between the predictor and the target function (Theorem 1). We emphasize that the encoder can be *arbitrary*, *i.e.* any architecture and size. As a result, Theorem 1 starkly contrasts prior work in that it disentangles the effects of the model and the augmentation: our generalization bound (a) is nonparametric, (b) does not depend on any model complexity or model inductive bias, and hence (c) allows the user to choose any model class for the encoder. Our bound depends on two key quantities: (a) the augmentation complexity $\kappa$, which is smaller for stronger and "better behaved" augmentations; (b) the trace gap, which serves as a proxy of how well the encoder approximates the induced RKHS.

The bound for an arbitrary encoder is great, but one might also be interested in proving guarantees for the *optimal* encoder that minimizes the worst-case approximation error. We will show that the optimal $d$-dimensional encoder consists of the top-$d$ eigenfunctions of the induced RKHS (Proposition 4). However, these eigenfunctions are not accessible given only finite samples, so we instead study the near-optimal $d$-dimensional encoder, which is a Monte-Carlo approximation of the optimal one. Our second set of results provide a generalization bound for the near-optimal encoder, by bounding its trace gap (which captures its quality) with high probability in terms of some spectral aspects of the induced RKHS. The bound shows that: As the number of unlabeled samples goes to infinity, the near-optimal encoder will become optimal.

A practical implication of our framework is that the *augmentation complexity* $\kappa$, defined as the $L^\infty$ norm of the kernel of $\mathcal{H}_\Gamma$, can be used as a tool to quantitatively compare different augmentations and analyze their impact on downstream performance. In Section 5, we demonstrate this point with mask-type augmentations on synthetic and real datasets, and show that (i) $\kappa$ depends on both the augmentation strength and the augmentation strategy; (ii) a smaller $\kappa$ (*e.g.* stronger augmentation) leads to a smaller generalization gap, but an overly strong augmentation causes poor training performance. Thus, there is a "sweet spot" in the middle with the best test performance. Overall, we believe that this work places modern representation learning on a more scientific footing and can inspire future empirical advancements. Finally, it is worth emphasizing that this work is about studying self-supervised learning from a kernel perspective, rather than studying kernel methods.

## 2 PROBLEM SETUP, GEOMETRY OF THE AUGMENTATION INDUCED RKHS

Let $\mathcal{X} \subset \mathbb{R}^{d_\mathcal{X}}$ be the data space, and $P_\mathcal{X}$ be the data distribution. Let $L^2(P_\mathcal{X})$ be the $L^2$ function space, such that any $f \in L^2(P_\mathcal{X})$ satisfies $\mathbb{E}_{P_\mathcal{X}}[f(X)^2] < \infty$. Let $\langle f_1, f_2 \rangle_{P_\mathcal{X}} = \int f_1(x) f_2(x) dP_\mathcal{X}(x)$ and

$\|f\|_{P_{\mathcal{X}}} = \langle f, f \rangle_{P_{\mathcal{X}}}^{1/2}$ be the inner product and the norm of the Hilbert space $L^2(P_{\mathcal{X}})$. Let $f^* \in L^2(P_{\mathcal{X}})$ be the target function we want to learn. This work studies the least-squares regression problem (see Györfi et al. (2002) for an introduction), which is formally stated as follows:

> **Problem.** *Given unlabeled samples $x_1, \cdots, x_N$ and labeled samples $\tilde{x}_1, \cdots, \tilde{x}_n$ i.i.d. sampled from $P_{\mathcal{X}}$, and labels $\tilde{y}_k = f^*(\tilde{x}_k) + \nu_k$ for $k \in [n]$ and random noise $\nu_k$, find a predictor $\hat{f} \in L^2(P_{\mathcal{X}})$ with a low prediction error* $\mathrm{err}(\hat{f}, f^*) := \|\hat{f} - f^*\|_{P_{\mathcal{X}}}^2 = \mathbb{E}_{P_{\mathcal{X}}}[(\hat{f}(X) - f^*(X))^2]$.

## 2.1 THE AUGMENTATION OPERATOR $\Gamma$: A SOURCE OF PRIOR KNOWLEDGE

The core problem in self-supervised representation learning is how to leverage the unlabeled data $x_1, \cdots, x_N$. Without labels, the pretraining task must be built upon some *prior knowledge* we have about the target function $f^*$. Data augmentation based pretraining tasks are built upon the following prior knowledge: *Two augmentations of the same sample $x$ should be similar to each other.*

Formally, let $\mathcal{A}$ denote an augmented space that contains the augmentations of $x \in \mathcal{X}$. Data augmentation induces a joint distribution $P_{\mathcal{AX}}$, with marginal distributions $P_{\mathcal{A}}$ and $P_{\mathcal{X}}$. We will use capital letters $A$ and $X$ to denote random variables from $P_{\mathcal{A}}$ and $P_{\mathcal{X}}$. We define an *augmentation operator* $\Gamma = \Gamma_{x \to a} : L^2(P_{\mathcal{X}}) \to L^2(P_{\mathcal{A}})$ as $(\Gamma_{x \to a} f)(a) = \mathbb{E}[f(X)|a]$ for all $f \in L^2(P_{\mathcal{X}})$, and define its adjoint operator $\Gamma^* = \Gamma_{a \to x} : L^2(P_{\mathcal{A}}) \to L^2(P_{\mathcal{X}})$ as $(\Gamma_{a \to x} g)(x) = \mathbb{E}[g(A)|x]$ for all $g \in L^2(P_{\mathcal{A}})$. We can see that $\Gamma_{a \to x}$ is indeed the adjoint of $\Gamma_{x \to a}$, since for all $f \in L^2(P_{\mathcal{X}})$ and $g \in L^2(P_{\mathcal{A}})$, there is $\langle \Gamma_{x \to a} f, g \rangle_{P_{\mathcal{A}}} = \iint f(x) g(a) p(a, x) da dx = \langle f, \Gamma_{a \to x} g \rangle_{P_{\mathcal{X}}}$. Intuitively, the augmentation operator connects any single sample $x$ to the distribution of its augmentations $p(a|x)$.

**Example:** In BERT, suppose we use 15% random masking. Then, $\mathcal{X}$ is the space of original sentences, and $\mathcal{A}$ is the space of 15% masked sentences; $P_{\mathcal{X}}$ is the distribution over original sentences, $A \sim p(\cdot|x)$ is the 15% randomly masked version of an original sentence $x$, and $P_{\mathcal{AX}}(a, x) = P_{\mathcal{X}}(x) p(a|x)$. Thus, $(\Gamma_{a \to x} g)(x)$ is the mean of $g$ over all 15% randomly masked versions of $x$.

Now we define the "soft invariance" *w.r.t.* the augmentation. Denote the function class $\mathcal{F}(\Gamma; \epsilon)$ to be the set of function $f^*$ such that $\exists g^* \in L^2(P_{\mathcal{A}})$ such that $f^*(x) = (\Gamma_{a \to x} g^*)(x) = E[g^*(A)|x]$, and

$$\frac{1}{2} \mathbb{E}_{X \sim P_{\mathcal{X}}} \mathbb{E}_{A, A' \sim p(\cdot|X)} \left[ (g^*(A) - g^*(A'))^2 \right] \le \epsilon \|g^*\|_{P_{\mathcal{A}}}^2, \tag{1}$$

for some small positive constant $\epsilon$. Eqn. (1) is a regularization constraint induced by the random walk normalized Laplacian over the *augmentation graph* defined in HaoChen et al. (2021), where the edge weight between $a$ and $a'$ is given by $P_A^+(a, a')$ in Eqn. (2). Compared to Assumption 1.1 in Johnson et al. (2023), Eqn. (1) has an additional $\|g^*\|_{P_{\mathcal{A}}}^2$ term on the right-hand side so it is homogeneous. For any $f \in \mathcal{F}(\Gamma; \epsilon)$, if $\|f\|_{P_{\mathcal{X}}} \le B$ for some constant $B$, then we denote $f \in \mathcal{F}_B(\Gamma; \epsilon)$.

> **Assumption 1** (See discussions in Appendix A). *We assume that $f^* \in \mathcal{F}_B(\Gamma; \epsilon)$ for given $B, \epsilon$.*

Let the range of $\Gamma^*$ be $R(\Gamma^*) = \{f = \Gamma_{a \to x} g \mid g \in L^2(P_{\mathcal{A}})\}$. Then $f^* \in R(\Gamma^*)$ by Assumption 1.

## 2.2 INDUCED RKHS $\mathcal{H}_\Gamma$ AND THE ISOMETRY PROPERTY

Now, we show that $R(\Gamma^*)$ is an RKHS. Define the following kernel over $\mathcal{A} \times \mathcal{A}$:

$$K_A(a_1, a_2) := \frac{dP_A^+}{d(P_{\mathcal{A}} \otimes P_{\mathcal{A}})} = \frac{P_A^+(a_1, a_2)}{P_{\mathcal{A}}(a_1) P_{\mathcal{A}}(a_2)}, \; P_A^+(a_1, a_2) := \int p(a_1|x) p(a_2|x) dP_{\mathcal{X}}(x). \tag{2}$$

Here, $P_A^+$ is called the *associative distribution*, and $K_A$ is the *Radon-Nikodym derivative* of $P_A^+$ *w.r.t.* $P_{\mathcal{A}} \otimes P_{\mathcal{A}}$, also called the positive-pair kernel in Johnson et al. (2023). Next, we define a *dual kernel* as:

$$K_X(x_1, x_2) := \frac{dP_X^+}{d(P_{\mathcal{X}} \otimes P_{\mathcal{X}})} = \frac{P_X^+(x_1, x_2)}{P_{\mathcal{X}}(x_1) P_{\mathcal{X}}(x_2)} = \int \frac{p(a|x_1) p(a|x_2)}{P_{\mathcal{A}}(a)} da, \tag{3}$$

which is constructed by swapping $a$ and $x$ in $K_X$ and then applying the Bayes rule. It holds that:

$$\begin{cases} (\Gamma_{a \to x} \Gamma_{x \to a} f)(x) = (\Gamma^* \Gamma f)(x) = \int K_X(x, x') f(x') P_{\mathcal{X}}(x') dx'; \\ (\Gamma_{x \to a} \Gamma_{a \to x} g)(a) = (\Gamma \Gamma^* g)(a) = \int K_A(a, a') g(a') P_{\mathcal{A}}(a') da'. \end{cases}$$

See App. D for derivation. Let $\lambda_i \in \mathbb{R}$ and $\psi_i \in L^2(P_\mathcal{X})$ be the sorted *eigenvalue* and *eigenfunction* of $\Gamma^*\Gamma$, *i.e.* $\Gamma^*\Gamma\psi_i = \lambda_i\psi_i$, with $\lambda_1 \geq \lambda_2 \geq \cdots \geq 0$. Suppose $\int K_X(x,x')^2 dP_\mathcal{X}(x)dP_\mathcal{X}(x') < \infty$, so that $\Gamma^*\Gamma$ is a compact Hilbert-Schmidt integral operator. Then, by Hilbert-Schmidt theorem, we can choose $\psi_1, \psi_2, \cdots$ that form an orthonormal basis of $L^2(P_\mathcal{X})$, such that $\langle\psi_i,\psi_j\rangle_{P_\mathcal{X}} = \delta_{i,j}$, and any $f \in L^2(P_\mathcal{X})$ can be written as $f = \sum_i u_i\psi_i$ for some $\{u_i\}_i$.

Note that $\lambda_i \in [0,1]$, since $\|g\|^2_{P_\mathcal{A}} \geq \|\Gamma^*g\|^2_{P_\mathcal{X}}$ for any $g \in L^2(P_\mathcal{A})$ by Jensen's inequality. It is easy to check that $\lambda_1 = 1$ and $\psi_1 \equiv 1$ is always a pair of eigenvalue and eigenfunction of $\Gamma^*\Gamma$. Moreover, when $d$ is sufficiently large, $\lambda_d$ should be close to $0$. Similarly, we can define a set of orthonormal eigenfunctions of $\Gamma\Gamma^*$ denoted by $\{\phi_i\}$, and we can relate these two sets of eigenfunctions by:

**Proposition 1** (Duality). *Operators $\Gamma\Gamma^*$ and $\Gamma^*\Gamma$ share the same non-zero eigenvalues, and there exist eigenfunctions $\{\phi_i\}$ of $\Gamma\Gamma^*$ that form an orthonormal basis of $L^2(P_\mathcal{A})$, such that for any $\lambda_i > 0$,*

$$\psi_i = \lambda_i^{-1/2}\Gamma^*\phi_i = \lambda_i^{-1/2}\Gamma_{a\to x}\phi_i \quad and \quad \phi_i = \lambda_i^{-1/2}\Gamma\psi_i = \lambda_i^{-1/2}\Gamma_{x\to a}\psi_i.$$

*Moreover, we have the following spectral decomposition of the Radon-Nikodym derivative:*

$$\frac{dP_{\mathcal{AX}}}{d(P_\mathcal{A} \otimes P_\mathcal{X})} = \frac{P_{\mathcal{AX}}(a,x)}{P_\mathcal{A}(a)P_\mathcal{X}(x)} = \sum_i \lambda_i^{1/2}\phi_i(a)\psi_i(x). \tag{4}$$

Define a Hilbert space $\mathcal{H}_\Gamma := \left\{ f = \sum_i u_i\psi_i \in L^2(P_\mathcal{X}) \mid \sum_i \lambda_i^{-1}u_i^2 < \infty \right\}$, whose inner product is $\langle f_1, f_2\rangle_{\mathcal{H}_\Gamma} = \sum_i \lambda_i^{-1}u_iv_i$ for $f_1 = \sum_i u_i\psi_i$ and $f_2 = \sum_i v_i\psi_i$. It satisfies (proof in Appendix D):

(i) $K_X$ is the reproducing kernel of $\mathcal{H}_\Gamma$, such that for all $f \in \mathcal{H}_\Gamma$, $f(x) = \langle f, K_X(x,\cdot)\rangle_{\mathcal{H}_\Gamma}$.

(ii) $\mathcal{H}_\Gamma = R(\Gamma^*)$. We call this the augmentation induced RKHS.

(iii) $\mathcal{H}_\Gamma$ is isometric to $\text{span}(\{\phi_i\}_{\lambda_i>0})$, a subspace of $L^2(P_\mathcal{A})$, and $\|f\|_{\mathcal{H}_\Gamma} = \inf_{g:f=\Gamma^*g}\|g\|_{P_\mathcal{A}}$.

(iv) For any $f^* \in \mathcal{F}_B(\Gamma;\epsilon) \subset R(\Gamma^*)$, let $f^* = \sum_i u_i\psi_i$. Define $g_0 := \sum_i \lambda_i^{-1/2}u_i\phi_i$. Then, $g_0$ must satisfy Eqn. (1), so we can choose $g^* = g_0$, in which case Eqn. (1) is equivalent to:

$$\langle g^*, (I - \Gamma\Gamma^*)g^*\rangle_{P_\mathcal{A}} \leq \epsilon\|g^*\|^2_{P_\mathcal{A}} \Leftrightarrow \sum_i \frac{1-\lambda_i}{\lambda_i}u_i^2 \leq \epsilon\sum_i \frac{1}{\lambda_i}u_i^2. \tag{5}$$

Moreover, Eqn. (5) is equivalent to a simple and intuitive *isometry property*:

$$\boxed{(1-\epsilon)\|f^*\|^2_{\mathcal{H}_\Gamma} \leq \|f^*\|^2_{P_\mathcal{X}} \leq \|f^*\|^2_{\mathcal{H}_\Gamma}.} \tag{6}$$

**Understanding the isometry property:** This property essentially says that the operator $\Gamma^*\Gamma$ preserves most variance of the target function $f^*$ in the infinite-dimensional functional space. Thus naturally, the optimal $d$-dimensional encoder that minimizes the approximation error should keep as much variance as possible under $\Gamma^*\Gamma$. This isometry property can be considered as a counterpart of the restricted isometry property (RIP) in infinite-dimensional functional spaces: In Section 4, we will show that the optimal encoder consists of the top-$d$ eigenfunctions of $\Gamma^*\Gamma$. Analogously, for a finite-dimensional vector space, PCA finds the $d$ principal components that keep the most variance under a linear transformation $T$, and they consist of the top-$d$ eigenvectors of $T$.

## 3 LEARNING GUARANTEES FOR AN ARBITRARY ENCODER

We now present generalization bounds for an arbitrary encoder. But before that, we need to clarify which encoder we are talking about. In practice, people don't directly pretrain $\hat{\Psi}$ during upstream. Instead, the common practice is to first pretrain an encoder $\hat{\Phi} = [\hat{\phi}_1, \cdots, \hat{\phi}_d]$ on the augmented space with $\hat{\phi}_i \in L^2(P_\mathcal{A})$, and then transform it into $\hat{\Psi}$, on top of which a linear probe is learned downstream. For example, in contrastive learning, the encoder is trained on views of images instead of original images; in masked language modeling, the encoder is trained on masked sentences instead of full sentences. In practice, people usually apply $\hat{\Phi}$ to downstream tasks directly without transformation, but for theoretical analysis we need to explicitly write out the transformation since $\hat{\Psi}$ and $\hat{\Phi}$ work on different spaces. We consider the commonly used *average encoder* (Saunshi et al., 2022, Eqn. (4)):

$$\hat{\Psi}(x) = \mathbb{E}[\hat{\Phi}(A)|x] = \int \hat{\Phi}(a)p(a|x)da, \tag{7}$$

which is equivalent to $\hat{\Psi} = \Gamma^* \hat{\Phi}$, and thus $\hat{\psi}_i \in R(\Gamma^*) = \mathcal{H}_\Gamma$ for all $i \in [d]$. In this section, $\hat{\Phi}$ can be an arbitrary function, even with infinite dimensions, so that the model can have any architecture and size; whereas $\hat{\Psi}$ is always the average encoder of $\hat{\Phi}$. Moreover, we consider the following predictor:

> **Definition 1.** *The final predictor is the **nonparametric least-squares estimate** defined as*
> $$\hat{f} := \underset{f: f = w^\top \hat{\Psi} \in \mathcal{H}_{\hat{\Psi}}, \|f\|_{\mathcal{H}_\Gamma} \leq \frac{B}{\sqrt{1-\epsilon}}}{\arg\min} \left\{ \frac{1}{n} \sum_{k=1}^{n} (\tilde{y}_k - f(\tilde{x}_k))^2 \right\}. \tag{8}$$

Here, note that by Eqn. (6) and $\|f^*\|_{P_\mathcal{X}} \leq B$, there is $\|f^*\|_{\mathcal{H}_\Gamma} \leq \frac{B}{\sqrt{1-\epsilon}}$.

We next introduce two key ingredients crucial to our analyses. The first ingredient is what we term the *augmentation complexity*, which uniformly bounds the kernel and is typically required in RKHS generalization analyses; for instance, see Schölkopf & Smola (2002, Section 12.1.3).

> **Definition 2.** *Define the **augmentation complexity** as $\kappa := \|K_X\|_\infty^{1/2}$, i.e. for $P_\mathcal{X}$-almost all $x$,*
> $$K_X(x,x) = \sum_i \lambda_i \psi_i(x)^2 = \int \frac{p(a|x)^2}{P_\mathcal{A}(a)} da = D_{\chi^2}(P_\mathcal{A}(\cdot|x) \| P_\mathcal{A}) + 1 \leq \kappa^2.$$

Here, $D_{\chi^2}(P \| Q) := \int (\frac{dP}{dQ} - 1)^2 dQ$ is the $\chi^2$-divergence. Let $S_\lambda(d) := \sum_{i=1}^{d} \lambda_i$ and $S_\lambda := S_\lambda(\infty) = \sum_{i=1}^{\infty} \lambda_i$. Wang et al. (2022b) showed that $S_\lambda = D_{\chi^2}(P_{\mathcal{A}|\mathcal{X}} \| P_\mathcal{A}) + 1$. By convexity of $D_{\chi^2}$, we have $S_\lambda \leq \kappa^2$, i.e. $\Gamma^*\Gamma$ is a trace-class operator (or $\kappa^2 \geq \int K_X(x,x) dP_\mathcal{X}(x) = S_\lambda$). Thus, $\kappa \geq 1$ as $\lambda_1 = 1$. We will provide examples of this augmentation complexity in Section 5.

Our second ingredient is the *trace gap* that captures the quality of the encoder. It is based on the notion of the *ratio trace*. Given an encoder $\hat{\Phi}$, without loss of generality, suppose it is full-rank.

> **Definition 3.** *Define covariance matrices $\boldsymbol{F}, \boldsymbol{G}$ as $\boldsymbol{F}(i,j) = \langle \hat{\psi}_i, \hat{\psi}_j \rangle_{P_\mathcal{X}} = \langle \Gamma^* \hat{\phi}_i, \Gamma^* \hat{\phi}_j \rangle_{P_\mathcal{X}}$ and $\boldsymbol{G}(i,j) = \langle \hat{\phi}_i, \hat{\phi}_j \rangle_{P_\mathcal{A}}$. Then, the **ratio trace** is defined as $\mathrm{Tr}(\boldsymbol{G}^{-1}\boldsymbol{F})$, if $\boldsymbol{G}^{-1}$ is well-defined.*

Ratio trace is a classical quantity in linear discriminant analysis (LDA) (Wang et al., 2007) and, as we will show, controls the approximation error. The largest ratio trace of any $d$-dimensional $\hat{\Phi}$ is $\lambda_1 + \cdots + \lambda_d$, and can be achieved by the top-$d$ eigenspace of $\mathcal{H}_\Gamma$. Then, define the *learned kernel* as

$$\hat{K}_{\hat{\Psi}}(x,x') = \langle \Gamma^*(\boldsymbol{G}^{-1/2}\hat{\Phi})(x), \Gamma^*(\boldsymbol{G}^{-1/2}\hat{\Phi})(x') \rangle, \tag{9}$$

which is the reproducing kernel of $\mathcal{H}_{\hat{\Psi}} = \mathrm{span}(\hat{\psi}_1, \hat{\psi}_2, \cdots)$, a subspace of $\mathcal{H}_\Gamma$. Here $\boldsymbol{G}^{-1/2}$ is used for normalization. The ratio trace can be viewed as the trace of $\mathcal{H}_{\hat{\Psi}}$. Then, define the **trace gap** as:

$$\tau^2 := \inf_{d' \leq d} \inf_{h_1, \cdots, h_{d'}} S_\lambda(d'+1) - \mathrm{Tr}(\boldsymbol{G}_h^{-1}\boldsymbol{F}_h), \tag{10}$$

where $\tau \geq 0$, $h_i = w_i^\top \hat{\Phi}$, $\boldsymbol{G}_h = (\langle h_i, h_j \rangle_{P_\mathcal{A}})_{i,j \in [d']}$, and $\boldsymbol{F}_h = (\langle \Gamma^* h_i, \Gamma^* h_j \rangle_{P_\mathcal{X}})_{i,j \in [d']}$. Note that for any $d' \leq d$ there is $\mathrm{Tr}(\boldsymbol{G}_h^{-1}\boldsymbol{F}_h) \leq S_\lambda(d')$, so $\tau^2$ is always lower bounded by $\lambda_{d+1}$. And by choosing $h_i = \hat{\phi}_i$ for $i \in [d]$, we can see that $\tau^2 \leq S_\lambda(d+1) - \mathrm{Tr}(\boldsymbol{G}^{-1}\boldsymbol{F})$.

We now state our first result. For simplicity, we assume that the random noise $\nu_1, \cdots, \nu_n$ are *i.i.d.* $\mathcal{N}(0, \sigma^2)$ variates for some $\sigma > 0$, though this can be relaxed (Wainwright, 2019, Chapter 13).

> **Theorem 1.** *Let $\nu_1, \cdots, \nu_n$ be i.i.d. $\mathcal{N}(0, \sigma^2)$ variates. Let $\hat{f}$ be defined by Definition 1. If $\hat{\Phi}$ has $d$ dimensions and $\tau < 1$ (d can be $\infty$), then there are universal constants $c_0, c_1, c_2$ such that with probability at least $1 - c_1 \exp\left(-\frac{c_2\sqrt{2nS_\lambda(d+1)}}{\kappa}\right) - \exp\left(-\sqrt{\frac{2n\kappa^2 B^2}{1-\epsilon}}\right)$, there is*
> $$\|\hat{f} - f^*\|_{P_\mathcal{X}}^2 \leq \frac{9\tau^2(\tau + \epsilon)B^2}{(1-\tau^2)(1-\epsilon)} + \frac{c_0 \kappa (B^2 + \sigma B)}{1-\epsilon} \sqrt{\frac{S_\lambda(d+1)}{n}} \quad \text{for all } f^* \in \mathcal{F}_B(\Gamma; \epsilon).$$

*Remark.* The first term in the bound controls the approximation error entailed by the limited capacity of the $d$-dimensional encoder $\hat{\Phi}$. This term may not vanish as $N, n \to \infty$, since $\tau^2$ is lower bounded

by $\lambda_{d+1}$ which can be positive. For instance, if $d$ is too small for $\hat{\Phi}$ to represent $f^*$, then the approximation error cannot be zero regardless of the number of samples available. Nevertheless, we can show that with a proper pretraining algorithm, the trace gap $\tau^2$ can be very close to $\lambda_{d+1}$ as $N \to \infty$ (Theorem 2). A larger ratio trace leads to a smaller approximation error, and indeed many existing objectives are strongly connected to maximizing $\mathrm{Tr}(\boldsymbol{G}^{-1}\boldsymbol{F})$ (see Appendix C). The second term bounds the downstream estimation error, which vanishes as $n \to \infty$. Finally, this result requires full access to $p(a|x)$ for any $x$, which is available in theory since the augmentation scheme is of our choice. Practical considerations for estimating $p(a|x)$ is outside the scope of this work.

➢ **Proof Sketch of Theorem 1:** This result follows from two bounds:

**Lemma 2** (Estimation error bound). *Suppose $\nu_1, \cdots, \nu_n$ are i.i.d. $\mathcal{N}(0, \sigma^2)$ variates. If $\hat{\Phi}$ has $d$ dimensions ($d$ can be $\infty$), then we have the following uniform bound over all $f^* = \Gamma^* g^* \in \mathcal{F}_B(\Gamma; \epsilon)$:*

$$\mathbb{P}_{\tilde{x}_i, \nu_i} \left[ \forall f^* \in \mathcal{F}_B(\Gamma; \epsilon), \|\hat{f} - f^*\|_{P_{\mathcal{X}}}^2 \leq 9\|f_{\hat{\Psi}} - f^*\|_{P_{\mathcal{X}}}^2 + \frac{c_0\kappa(B^2 + \sigma B)}{1-\epsilon}\sqrt{\frac{S_\lambda(d+1)}{n}} \right]$$
$$\geq 1 - c_1 \exp\left(-\frac{c_2\sqrt{2nS_\lambda(d+1)}}{\kappa}\right) - \exp\left(-\sqrt{\frac{2n\kappa^2 B^2}{1-\epsilon}}\right),$$

*where $f_{\hat{\Psi}} = \Gamma^*(\Pi_{\hat{\Phi}} g^*)$ is the projection of $f^*$ onto $\mathcal{H}_{\hat{\Psi}}$ w.r.t. $\langle \cdot, \cdot \rangle_{\mathcal{H}_\Gamma}$, and $c_0, c_1, c_2$ are universal constants. Note that $\hat{f}$ depends on $f^*$. Moreover, $S_\lambda(d+1) \leq \min\left\{d+1, \kappa^2\right\}$.*

*Remark.* The constant 9 in the bound is loose and can be tightened arbitrarily close to 1 (at the cost of increasing other terms in the bound) by straightforward modifications to the proof, which we omit here. Such a greater-than-one constant is inevitable for any uniform deviation bound.

**Lemma 3** (Approximation error, upper bound). *If $\tau < 1$, then for any $f^* \in \mathcal{F}_B(\Gamma; \epsilon)$, there is*

$$\|f_{\hat{\Psi}} - f^*\|_{P_{\mathcal{X}}}^2 \leq \frac{\tau^2}{1-\tau^2}\frac{\tau+\epsilon}{1-\epsilon}B^2. \tag{11}$$

## 4 ANALYSES FOR THE NEAR-OPTIMAL ENCODER

In this section, we consider the special case where $\hat{\Psi}$ is near-optimal, *i.e.* the Monte-Carlo approximation of the optimal $d$-dimensional encoder that minimizes the worst-case approximation error. First, we show that the optimal encoder spans the *top-$d$ eigenspace*, which is the linear span of the top-$d$ eigenfunctions $\psi_1, \cdots, \psi_d$. Define the *worst-case approximation error* over $\mathcal{F}_B(\Gamma; \epsilon)$ as:

$$\mathrm{err}(\hat{\Psi}; \mathcal{F}_B(\Gamma; \epsilon)) := \sup_{f \in \mathcal{F}_B(\Gamma;\epsilon)} \min_{w \in \mathbb{R}^d} \mathrm{err}(w^\top \hat{\Psi}, f) = \sup_{f \in \mathcal{F}_B(\Gamma;\epsilon)} \min_{w \in \mathbb{R}^d} \|w^\top \hat{\Psi} - f\|_{P_{\mathcal{X}}}^2.$$

**Proposition 4** (Approximation error, lower bound). *For any $\hat{\Psi} = [\hat{\psi}_1, \cdots, \hat{\psi}_d]$ where $\hat{\psi}_i \in L^2(P_{\mathcal{X}})$,*

$$\mathrm{err}(\hat{\Psi}; \mathcal{F}_B(\Gamma; \epsilon)) \geq \frac{\lambda_{d+1}}{1-\lambda_{d+1}}\frac{\epsilon}{1-\epsilon}B^2 \quad \text{given that} \quad \frac{\lambda_{d+1}}{1-\lambda_{d+1}}\frac{\epsilon}{1-\epsilon} \leq \frac{1}{2}.$$

*To attain equality, it is sufficient for $\hat{\Psi}$ to span the top-$d$ eigenspace, and also necessary if $\lambda_{d+1} < \lambda_d$.* One can use kernel PCA to recover top-$d$ eigenspace (Belkin & Niyogi, 2003; Johnson et al., 2023), which is however not scalable. Alternatively, one can use variational objectives that attain minimum when $\hat{\Phi}$ spans the top-$d$ eigenspace (see Appendix C for examples). Also, when $\lambda_d > 0$, the top-$d$ eigenspace lies within $R(\Gamma^*)$, which is the function class of $\Psi$ when using the average encoder.

The optimal $d$-dimensional encoder is inaccessible since $\psi_1, \cdots, \psi_d$ cannot be extracted with only finite samples. What one can do instead is extracting the *empirical top-$d$ eigenspace* $\hat{\mathcal{H}}_d$. Given samples $x_1, \cdots, x_N$, define two Hilbert spaces $L^2(\hat{P}_{\mathcal{X}})$ and $L^2(\hat{P}_{\mathcal{A}})$, where $\langle f_1, f_2 \rangle_{\hat{P}_{\mathcal{X}}} = \frac{1}{N}\sum_{k=1}^N f_1(x_k)f_2(x_k)$, and $\langle g_1, g_2 \rangle_{\hat{P}_{\mathcal{A}}} = \int g_1(a)g_2(a)d\hat{P}_{\mathcal{A}}(a)$ for $\hat{P}_{\mathcal{A}}(a) = \frac{1}{N}\sum_{k=1}^N p(a|x_k)$. With these finite samples, $\Gamma^*$ will remain the same, but $\Gamma$ will become the empirical version $\bar{\Gamma}$:

$$(\bar{\Gamma}f)(a) = \frac{1}{N}\sum_{k=1}^N \frac{f(x_k)p(a|x_k)}{\hat{P}_{\mathcal{A}}(a)}.$$

Let the eigenvalues and eigenfunctions of $\Gamma^*\bar{\Gamma}$ be $\{(\bar{\lambda}_i, \bar{\psi}_i)\}$. Let $\bar{\phi}_i$ be the eigenfunctions of $\bar{\Gamma}\Gamma^*$. $\bar{\phi}_i$ and $\bar{\psi}_i$ have the duality property similar to Proposition 1. Let $\hat{\mathcal{H}}_\Gamma$ be the RKHS associated with

$\Gamma^* \bar{\Gamma}$, which is the span of $\{\bar{\psi}_i\}_{\bar{\lambda}_i > 0}$ and hence is a subspace of $\mathcal{H}_\Gamma = R(\Gamma^*)$. Let $\hat{\mathcal{H}}_d$ be the top-$d$ eigenspace of $\hat{\mathcal{H}}_\Gamma$. Let $\lambda_{\max}(\boldsymbol{G})$, $\lambda_{\min}(\boldsymbol{G})$ be the largest and smallest eigenvalue of $\boldsymbol{G}$. For the case of interest where $\mathcal{H}_{\hat{\Psi}} = \hat{\mathcal{H}}_d$, we have our second main learning guarantee that bounds the trace gap:

---

**Theorem 2.** *Suppose $\hat{\phi}_i = \bar{\phi}_i$ for $i \in [d]$. Let $\gamma_{\boldsymbol{G}} := \lambda_{\max}(\boldsymbol{G})/\lambda_{\min}(\boldsymbol{G})$ be the condition number of $\boldsymbol{G}$. Then, for any $\delta > 0$, it holds with probability at least $1 - \delta$ that*

$$\tau^2 \leq S_\lambda(d+1) - \mathrm{Tr}(\boldsymbol{G}^{-1}\boldsymbol{F}) \leq \lambda_{d+1} + \left(2 + \sqrt{2 \log \frac{2}{\delta}}\right) \frac{(\lambda_d^{-1} + \bar{\lambda}_d^{-1}\gamma_{\boldsymbol{G}}^{1/2} + 2)\kappa^2}{\sqrt{N}} d.$$

---

*Remark.* Combining this result with the first main Theorem 1 leads to the bound for $\mathcal{H}_{\hat{\Psi}} = \hat{\mathcal{H}}_d$. Since $\tau^2 \geq \lambda_{d+1}$, this result says that the gap between $\tau^2$ and its optimal value is $O(N^{-1/2})$, ignoring potential dependence of $\gamma_{\boldsymbol{G}}$ and $\bar{\lambda}_d^{-1}$ on $N$. Comparing the upper bound in Lemma 3 + Theorem 2 to the lower bound in Proposition 4, we note that the upper bound is near tight: The only difference is that in Eqn. (11), we have $\frac{\tau+\epsilon}{1-\epsilon}$ instead of $\frac{\epsilon}{1-\epsilon}$. Unlike the bounds in HaoChen et al. (2021); Saunshi et al. (2022), this bound only depends on $\lambda_d^{-1}$, $\bar{\lambda}_d^{-1}$ and $\gamma_{\boldsymbol{G}}$, but not $(\lambda_{d'} - \lambda_{d+1})^{-1}$ for some $d' \leq d$, *i.e.* it does not require the separability of the top-$d$ eigenfunctions, because it does not need the top-$d$ eigenspace to be close to the empirical top-$d$ eigenspace. Instead, it only requires the traces of the two top-$d$ eigenspaces to be close. This is a big improvement since the eigenvalues can have high multiplicity as pointed out by Saunshi et al. (2022).

➤ **Proof Sketch of Theorem 2:** Denote $S_{\bar{\lambda}}(d) := \sum_{i=1}^d \bar{\lambda}_i$. We define two empirical covariance matrices $\hat{\boldsymbol{F}}$ and $\hat{\boldsymbol{G}}$ as: $\hat{\boldsymbol{F}}(i,j) = \langle \hat{\psi}_i, \hat{\psi}_j \rangle_{\hat{P}_\mathcal{X}}$, and $\hat{\boldsymbol{G}}(i,j) = \langle \hat{\phi}_i, \hat{\phi}_j \rangle_{\hat{P}_\mathcal{A}}$. Since any invertible linear transformation to $\hat{\Phi}$ does not change $\mathrm{Tr}(\boldsymbol{G}^{-1}\boldsymbol{F})$ or the empirical ratio trace $\mathrm{Tr}(\hat{\boldsymbol{G}}^{-1}\hat{\boldsymbol{F}})$, we can see that for $\mathcal{H}_{\hat{\Psi}} = \hat{\mathcal{H}}_d$ there is $\mathrm{Tr}(\hat{\boldsymbol{G}}^{-1}\hat{\boldsymbol{F}}) = S_{\bar{\lambda}}(d)$ (simply consider $\hat{\phi}_i = \bar{\phi}_i$). Thus, it suffices to bound $|\mathrm{Tr}(\hat{\boldsymbol{G}}^{-1}\hat{\boldsymbol{F}}) - \mathrm{Tr}(\boldsymbol{G}^{-1}\boldsymbol{F})|$, and $S_\lambda(d) - S_{\bar{\lambda}}(d)$.

We start with bounding the gap between empirical and real ratio traces for any encoder $\hat{\Phi}$:

**Lemma 5.** *Suppose there exists a constant $C > 0$ such that $\mathbb{E}_{P_\mathcal{A}}[g^4] \leq C^2 \|g\|_{P_\mathcal{A}}^2$, for all $g = w^\top \hat{\Phi}$ where $\|g\|_{P_\mathcal{A}} \leq 1$. Then, for any $\delta > 0$, it holds with probability at least $1 - \delta$ that*

$$|\mathrm{Tr}(\hat{\boldsymbol{G}}^{-1}\hat{\boldsymbol{F}}) - \mathrm{Tr}(\boldsymbol{G}^{-1}\boldsymbol{F})| \leq \left(2 + \sqrt{2 \log \frac{2}{\delta}}\right) \frac{C\kappa + \kappa^2}{\sqrt{N}} d. \tag{12}$$

Lemma 5 considers a general $\hat{\Phi}$, and requires a fourth-moment control assumption (Wainwright, 2019, Eqn. (14.22a)). For the specific case $\hat{\phi}_i = \bar{\phi}_i$ studied in this section, we can prove that this assumption holds (that is why it does not appear in Theorem 2), *i.e.* the top-$d$ empirical eigenfunctions can be proved to be *delocalized* (Erdős et al., 2009). In particular, we prove the following:

**Lemma 6.** *Suppose $\hat{\phi}_i = \bar{\phi}_i$ for $i \in [d]$. Let $\gamma_{\boldsymbol{G}} := \lambda_{\max}(\boldsymbol{G})/\lambda_{\min}(\boldsymbol{G})$, which is the condition number of $\boldsymbol{G}$. Then, for any $\delta > 0$, both*

$$\sum_{j=1}^d \bar{\lambda}_j \geq \sum_{i=1}^d \lambda_i - \left(2 + \sqrt{2 \log \frac{2}{\delta}}\right) \frac{(\lambda_d^{-1} + 1)\kappa^2}{\sqrt{N}} d$$

*and Eqn. (12) with $C = \kappa \bar{\lambda}_d^{-1} \gamma_{\boldsymbol{G}}^{1/2}$ hold simultaneously for $\mathcal{H}_{\hat{\Psi}} = \hat{\mathcal{H}}_d$ with probability at least $1 - \delta$.*

## 5 ESTIMATING AND EXPLOITING THE AUGMENTATION COMPLEXITY

An important quantity from our analysis is the augmentation complexity $\kappa$ — both approximation and estimation error bounds get smaller when $\kappa$ is reduced. A natural way to reduce $\kappa$ is via a stronger augmentation, which has indeed been helpful in practice (Chen et al., 2020; Wettig et al., 2023).

In this section, we take a closer look at $\kappa$ of mask-type augmentations. We study $\kappa$ of different masking schemes on the hypercube data model introduced in Saunshi et al. (2022), and show that $\kappa$ depends on both the mask ratio (augmentation strength) and the masking strategy. Then, we empirically estimate $\kappa$ on the real-world NLP dataset `wikipedia-simple`, and demonstrate

its correlation with the generalization gap on real downstream tasks. Our results indicate that the augmentation complexity $\kappa$ offers a means to compare different augmentations quantitatively and implies a tradeoff in the choice of augmentations.

## 5.1 THEORETICAL HYPERCUBE DATA MODEL

Like Cabannes et al. (2023), we study three random masking methods (Figure 2): (i) Independent random masking, (ii) Cutout-like block masking, and (iii) BERT-like masking. Let $\alpha$ denote the mask rate. Consider the data space $\mathcal{X} = \{-1, 1\}^{d_{\mathcal{X}}}$ with $P_{\mathcal{X}}$ being the uniform distribution over $\mathcal{X}$. Denote the corresponding $\kappa$ of these three methods to be $\kappa_r, \kappa_c$, and $\kappa_b$. We will show that these $\kappa$'s will be very different even with the same $\alpha$.

Figure 2: Augmentation illustration.

Let's first consider the independently random masking (derivations are deferred to Appendix G):

**Example 1.** *Consider a* random masking *augmentation, i.e. for any $x \in \mathcal{X}$, each coordinate $x^{(i)}$ is randomly and independently masked to be $0$ (i.e. $0$ denotes the* [MASK] *token) with probability $\alpha \in (0, 1)$. Then, its augmentation complexity is given by $\kappa_r^2 = (2 - \alpha)^{d_{\mathcal{X}}}$.*

Next, consider the Cutout-like block masking (DeVries & Taylor, 2017), which has been successfully applied in practice such as ViT (Dosovitskiy et al., 2021) and MAE (He et al., 2022). Compared to the previous example, Cutout-like block masking leads to a smaller $\kappa$ with the same $\alpha$:

**Example 2.** *Consider a* random block masking *augmentation, i.e. masking $x^{(i)}, x^{(i+1)}, \cdots, x^{(i+r-1)}$ for $r = \lceil \alpha d_{\mathcal{X}} \rceil$ and a uniformly random $i \in [d_{\mathcal{X}} - r]$, for any $x \in \mathcal{X}$. Then, $\kappa_c^2 \leq [2^{(1-\alpha)}]^{d_{\mathcal{X}}}$.*

Our third and final example resembles the 80-10-10 strategy in BERT (Devlin et al., 2019): First randomly mask some tokens, and then randomly replace some unmasked tokens with other tokens.

**Example 3.** *Consider* random block masking with flipping*, where for any $x \in \mathcal{X}$, first mask $x^{(i)}, \cdots, x^{(i+r-1)}$ to be $0$ for $r = \lceil \alpha d_{\mathcal{X}} \rceil$ and a uniformly random $i \in [d_{\mathcal{X}} - r]$, then randomly flip the sign of each remaining coordinate w.p. $\frac{\alpha}{2}$ independently. Then, $\kappa_b^2 \leq \left[ (\alpha^2 - 2\alpha + 2)^{(1-\alpha/2)} \right]^{d_{\mathcal{X}}}$.*

While $\kappa_r, \kappa_c, \kappa_b$ all have an exponential dependency on $d_{\mathcal{X}}$, their bases are different, as shown in Figure 3a. This means these methods have different $\kappa$ despite having the same mask rate, which highlights the importance of the masking strategy.

**On the exponentiality of $\kappa$ in $d_{\mathcal{X}}$.** We suspect the exponential dependency on $d_{\mathcal{X}}$ is a manifestation of the typical curse of dimensionality in high-dimensional statistics in the absence of strong inductive bias. Bengio et al. (2013, Section 3.2) had a discussion on this point. What our analysis makes salient is that the inductive bias can be expressed via controlled augmentation complexity. One approach to obtain a polynomial augmentation complexity is by using very strong augmentations, such as using ImageNet pretraining where $\mathcal{A}$ is a finite set of size $1000$, *i.e.* the ImageNet label set. However, such simple strong augmentations usually lead to substantially worse performance in practice. We posit developing more interesting augmentations with bounded augmentation complexity as an important open problem facing the field of representation learning.

We postulate moreover that despite the worst-case bounds above come with exponential dependency on data dimension, the empirical success of existing augmentation-based self-supervised learning suggest that they implicitly adapt to the inherent low-dimensional manifold structure in real-world data. We conjecture that as long as the augmentation captures such a structure, it can evade the curse.

## 5.2 AUGMENTATIONS ON REAL DATASETS

The augmentation complexity $\kappa$ can be estimated on real datasets and used to quantitatively analyze and compare augmentations, which we demonstrate on the NLP dataset `wikipedia-simple`. We study masked language modeling, where $x$ is a full sentence and $a$ is a masked sentence. Recall that $\kappa^2$ upper bounds $\int \frac{p(x|a)}{p(x)} p(a|x) da$. Empirically, we replace the integration with sample average; namely, for $x = [x^{(1)}, \cdots, x^{(l)}]$ where $x^{(i)}$ is the $i^{th}$ token, we have

$$\log p(x|a) = \log p\left(x^{(1)} \middle| a\right) + \log p\left(x^{(2)} \middle| a, x^{(1)}\right) + \cdots + \log p\left(x^{(l)} \middle| a, x^{(1)}, \cdots, x^{(l-1)}\right).$$

Figure 3: Plots for Section 5. In (b), $\log \kappa^2$ is estimated on `wikipedia-simple`.

We leverage a bi-directional MLM such as a BERT, and compute $p(x^{(i)}|a, x^{(<i)})$ auto-regressively: For any $i \in [l]$, compute $p(x^{(i)}|a, x^{(<i)})$ with the BERT, and then replace $a^{(i)}$ with $x^{(i)}$ for $i+1$. We compute this for a random subset of samples. We report the $99^{\text{th}}$ percentile of $K_X(x, x)$ instead of $\sup_x K_X(x, x)$ due to the presence of outliers. Experimental details are deferred to Appendix H.

**Estimating $\kappa$.** We study Random masking, Block masking, Random masking + Flipping, and Block masking + Flipping. For Masking + Flipping, we randomly mask $\alpha/2$ of the tokens, and replace another $\alpha/2$ of the tokens with random tokens; this replace rate is much higher than standard practice, since we want to highlight the effect of flipping. Figure 3b plots the estimated $\log \kappa^2$ of the four augmentations *w.r.t.* the mask rate $\alpha$, where $\log \kappa^2$ is calculated as the average of five runs with different random seeds. The complexity drops as $\alpha$ increases as predicted by theory. One observation is that the "Random + Flip" curve intersects with "Block" and "Block + Flip", suggesting that block masking has a stronger effect when $\alpha$ is small, whereas flipping has a stronger effect when $\alpha$ is large.

**Downstream performance.** We also study how the mask ratio $\alpha$ affects the downstream performance, using QNLI (Wang et al., 2018) and SST-2 (Socher et al., 2013). For pretraining, we train `roberta-large` models with different mask ratios using the fast pretraining recipe in Wettig et al. (2023). We focus on Random masking and do not apply the 80-10-10 strategy. For downstream, we fine-tune the encoder together with the linear head following common practice. To align better with our theory, we explicitly use the average encoder Eqn. (7), estimated by sampling 16 $a$'s for each $x$.

We evaluate the train/test accuracies of the models, and plot the test accuracy (blue solid) and the train-test accuracy gap (green dashed) in Figure 3c. The highest test accuracy is achieved at $\alpha = 0.15$ on QNLI and at $\alpha = 0.40$ on SST-2 (marked in red). The test accuracy is low when $\alpha$ is too small due to the large generalization gap, and also low when $\alpha$ is too large due to low training accuracy. Regarding the train-test gap, QNLI shows a monotonic decrease in the gap as the mask ratio grows, but the gap on SST-2 is U-shaped, with the lowest point at $\alpha = 0.40$. This is likely because with $\alpha > 0.40$ is too strong an augmentation for SST-2 that Assumption 1 is broken, in which case our theoretical results will not hold. Thus, these results align with our theory that while augmentations should be sufficiently robust, they must not be so strong as to violate Assumption 1. This suggests the presence of a "sweet spot", which is also supported by evidence in prior work (Tian et al., 2020).

## 6 CONCLUSION

This work establishes an RKHS approximation/regression framework that completely disentangles the effects of the model and the augmentation. Our framework (i) clarifies the roles of the two stages of representation learning, as well as the role of augmentation with the isometry property Eqn. (6); (ii) leads to nonparametric statistical guarantees free of model complexity or inductive bias, so that they are valid for an *arbitrary* encoder; and (iii) formulates the augmentation complexity $\kappa$ as a tool to quantitatively analyze the effect of augmentations. We also refer our readers to two recent works that used similar techniques but derived quite different results: Wang et al. (2022b) studied an RKHS similar to $\mathcal{H}_\Gamma$, but in a completely different context (they studied conditional moment models); Cabannes et al. (2023) studied a similar problem, but they used a pre-defined kernel to define the function class and derived guarantees, while the RKHS $\mathcal{H}_\Gamma$ in this work is completely induced by the data augmentation. We defer discussions for more related work to Appendix B.

**Limitations and open problems.** The major limitation of this work is that $\kappa$ can be exponential in $d_{\mathcal{X}}$, which as we pointed out could be a natural consequence of the curse of dimensionality, and that the low-dimensional structure of real-world data could be a potential reason of circumventing the exponential dependency. Another limitation is that we studied the average encoder, while usually people directly apply the original encoder to downstream. The latter case requires the model to perform some type of implicit transformation as the upstream and downstream have different input spaces ($\mathcal{A}$ and $\mathcal{X}$), and this transformation cannot be studied within the architecture-free framework of this work. Additionally, an open problem is how to deal with the common upstream-to-downstream data distribution shift in practice, for which downstream fine-tuning could be critical. This has been discussed in Cabannes et al. (2023), but a more general analysis is desired.

CODE

The code of Section 5 can be found at https://colab.research.google.com/drive/1loSZLLI-qfoKE7BCIi1SWJKgruU6i4ku?usp=sharing.

ACKNOWLEDGMENTS

We would like to thank Yutong He, Di He and Elan Rosenfeld for their very useful discussions, and Jeremy Cohen, Amrith Setlur, Xiaoyu Huang, Samuel Sokota, Zhili Feng, Swaminathan Gurumurthy and Zhengyang Geng for their feedback on the early draft of this work. We are also very grateful to our anonymous ICLR reviewers, with whose help this work has been greatly improved. We acknowledge the support of NSF via IIS-2211907, IIS-1909816, CCF-2238523, ONR via N00014-23-1-2368, and Amazon.

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

# A   DISCUSSION

**Why do we need Assumption 1 and can it be relaxed?**   We first note that an assumption on the target function is necessary to establish any non-vacuous learning theory. This is implied by the No Free Lunch Theorem (Wolpert & Macready, 1997), which states that if the target function is sampled uniformly at random from all possible target functions, then any learner achieves the same performance as random guess.

Assumption 1 implies an isometry property. As pointed out by Goldblum et al. (2023), real-world datasets are highly structured, and the structures are largely shared across different domains. Assumption 1 and implied isometry property suggest that a well-designed augmentation is a key to capturing such a low-dimensional cross-domain structure. Indeed, better augmentation has been the main driving force in the recent progress in contrastive learning and masked language modeling. Our focus on augmentation is in stark contrast to prior work on the generalization of overparameterized models (see Appendix B), which can find the global minima fitting all samples, yet cannot capture the invariant structure within the data and the target concept.

If we were to relax Assumption 1 (*e.g.* the target function $f^*$ can be outside but very close to $\mathcal{H}_\Gamma$), additional assumptions on $f^*$ would be required for our proof with uniform deviation bounds to hold. Namely, $f^*$ can no longer be *any* $L^2$ function that is close to $\mathcal{H}_\Gamma$. For instance, we at least need to assume that $|f^*(x)|$ is *a.e.* uniformly bounded by some constant $C$ which the final bound will depend on, or assume that $f^*$ belongs to some predefined RKHS similar to Cabannes et al. (2023). We feel that these additional assumptions would complicate our analysis and obscure the role of augmentation, hence we choose to assume $f^* \in \mathcal{H}_\Gamma$ instead.

**The main theorem allows the encoder to use any model. Does it mean that the inductive bias of the model is not useful as opposed to prior work?**   No, the inductive bias is still useful. In fact, we believe that the model family and data augmentation are two key ingredients of modern representation learning. The former has been studied by prior work (Saunshi et al., 2022; HaoChen & Ma, 2023), while this work studies the latter and introduces a class of augmentation complexity based generalization bounds that is orthogonal to the class of classical model complexity based bounds, in the sense that the bounds are nonparametric and do not require any inductive biases from the model family. This new class of guarantees deepens our understanding because it clarifies the roles of these two ingredients. However, a downside of our analysis is that it cannot leverage the geometric structure of the data. For example, the data might be so good that it is linearly separable in the Euclidean space, but our nonparametric framework cannot make use of this information.

The inductive bias of the model has at least two roles: (i) It further reduces the complexity on top of the data augmentation, leading to even tighter bounds; (ii) While we used the average encoder in our analysis, in practice people usually directly apply the pretrained encoder to the data in the downstream task. In this case, the model architecture has to implicitly perform some kind of transformation, since the pretraining and downstream samples come from different spaces, in which case the inductive biases of the model will be helpful.

**Comparing to the claim in Saunshi et al. (2022).**   Saunshi et al. (2022) argued that there exist setups where any learning guarantees independent of the model inductive bias are necessarily vacuous. They presented an example of a disjoint augmentation, such that any two different original samples $x_1$ and $x_2$ cannot be augmented to the same $a$. In this case, our $K_X$ defined in Eqn. (3) is zero everywhere, so apparently our analysis will not work. The observation made by Saunshi et al. (2022) is that even for this augmentation, if there is a good model (feature map), then good generalization can still be achieved, and therefore they claimed that model inductive bias is irreplaceable.

The analysis in this work is compatible with their observation. Our analysis shows that if the augmentation is good enough (such that Assumption 1 is satisfied and the augmentation complexity is small enough, which is in fact a strong assumption), then we can achieve good generalization only with the augmentation and without any help of the model. In other words, the disjoint augmentation example showcases a poor choice of augmentation, which is orthogonal to the emphasis of this paper.

# B RELATED WORK

## B.1 LEARNING GUARANTEES FOR BIG MODELS

Big models with millions and even billions of parameters have been proved to work well on a wide variety of tasks. However, their empirical success cannot be explained by classical generalization bounds that use the VC-dimension or the Rademacher complexity. This discrepancy was made clear by Zhang et al. (2017), which shows that while enjoying good generalization on real tasks, modern neural networks have sufficient capacity to fit random labels. This challenges the classical generalization theory, which appeals to the Occam's razor argument that big models cannot generalize well in general. There is a large body of research on improving classical generalization bounds for neural networks and big models, and we highlight a few seminal works below.

**Algorithmic stability.** This line of research attributes the good generalization of complex models to the stability of the (random) training algorithm. There is a folklore that one key reason why neural networks generalize so well is that they are trained by stochastic gradient descent (SGD). Let $f(x; A, S)$ be the predictor trained by a random training algorithm $A$ on a finite training set $S$. $A$ is called *uniformly stable* (Kearns & Ron, 1997; Bousquet & Elisseeff, 2002), if $E_A|f(x; A, S) - f(x; A, S')|$ is uniformly bounded when $S$ and $S'$ only differ by one training sample. $f$ is guaranteed to enjoy good generalization if $A$ is uniformly stable. Then, Hardt et al. (2016) proved that SGD is uniformly stable under certain conditions. There are also weaker notions of algorithmic stability (Mukherjee et al., 2006; Shalev-Shwartz et al., 2010). It has also been shown in Li et al. (2018); Arora et al. (2019a) that gradient methods for overparameterized models provide an implicit regularization effect that helps generalization. However, Zhang et al. (2021) challenged this line of work by empirically showing that SGD on real neural networks is not very stable, even with regularization.

**PAC-Bayes and sharpness.** Compared to algorithmic stability, the PAC-Bayes line of analysis attributes generalization to model stability, that is the empirical risk is insensitive to perturbations in model parameters. If the model is stable, then the empirical risk is small within a large region around the global minima, meaning that the risk landscape has low sharpness (Keskar et al., 2017; Neyshabur et al., 2017) around the global minima. The PAC-Bayes generalization bound McAllester (1999); Langford & Seeger (2001) considers a prior distribution $P$ over the model's function space (usually Gaussian or uniform), and a posterior stationary distribution $Q$ found by a training algorithm. If $D_{KL}(Q \parallel P)$ is small, then good generalization is guaranteed. Thus, the more models there are in the support of $Q$, the better the generalization. So we can see that a lower sharpness around the global minima leads to better generalization. Sharpness-based bounds are shown to be empirically tighter than other bounds in Jiang et al. (2020), and inspire a class of training method called sharpness-aware minimization (SAM) (Foret et al., 2021; Bartlett et al., 2023; Wen et al., 2023). Other generalization guarantees derived from the PAC-Bayes analysis with similar ideas include low spectral norm (Bartlett et al., 2017; Neyshabur et al., 2018), flat minima (Hochreiter & Schmidhuber, 1997), and derandomization (Negrea et al., 2020).

One prominent class of analysis within the PAC-Bayes paradigm uses *compression* to obtain tighter, and empirically non-vacuous generalization bounds. For instance, Dziugaite & Roy (2017) proposed to directly optimize the PAC-Bayes bound for multi-layer perceptrons, and it was improved by Rivasplata et al. (2019) with a relaxation on the bound. Then, Arora et al. (2018) formulated the compression framework, and showed that models with noise stability are more compressible and thus lead to tighter bounds (also discovered in Nagarajan & Kolter (2019)). Zhou et al. (2019) then used the compression framework to obtain non-vacuous generalization bounds for ImageNet. More recently, Lotfi et al. (2022) compressed models by considering model equivariance, and Goldblum et al. (2023) verified that large language models favor simpler sequences. Both papers proposed to use generalization bounds based on the Kolmogorov complexity (Kolmogorov, 1963).

**Implicit bias of overparameterized models.** This line of analysis attributes the good generalization to the implicit bias of a specific model architecture, *i.e.* with certain training algorithms such as SGD, these models can find the global minima that can generalize. First, on the optimization side, there is a series of work that shows that gradient descent finds the global optima of overparameterized neural networks, starting from the results in Kawaguchi (2016) for linear neural networks, and then Li & Yuan (2017); Du et al. (2019b;a); Arora et al. (2019b) for two-layer fully-connected ReLU

networks, and Allen-Zhu et al. (2019b) for other architectures such as CNNs and ResNets. The analysis on overparameterized fully-connected networks with two or more layers is then extended to generalization in Du & Lee (2018); Allen-Zhu et al. (2019a). There is also analysis based on neural tangent kernels (NTK) (Jacot et al., 2018), including Lee et al. (2019); Arora et al. (2019c). Most of these work requires the neural network to be overparameterized, *i.e.* the number of neurons is polynomially large comparing to the input size (Allen-Zhu et al., 2019b). One caveat of this line of analysis, as argued in Chizat et al. (2019), is that these overparameterized models might fall into the "lazy training" regime, meaning that the training dynamics of these models resemble those of a linear model. They argued that the phenomenal success of deep neural networks is unlikely to be due to their similarity to linear models, and Wang et al. (2020) provided a counter-argument.

## B.2 Theoretical Study on Augmentation Based Representation Learning

Many recent studies have been devoted to theoretically understanding representation learning. One line of research studies the effectiveness of contrastive learning by showing its features are optimal for linear predictor of certain downstream tasks (Saunshi et al., 2019; Tosh et al., 2021a;b), robust to class imbalance (Liu et al., 2021), and suitable for unsupervised domain adaptation (Shen et al., 2022; HaoChen et al., 2022). Masked prediction tasks have been shown to be useful for reducing the downstream sample complexity (Lee et al., 2021) and for parameter identifiability (Liu et al., 2022). Relating to language applications, Saunshi et al. (2019) explained why next-word prediction can benefit sentiment classification, and Wei et al. (2021) studied the effect of prompt tuning through the lens of implicit Bayesian inference. Regarding the optimization in representation learning, there have been prior works on the training dynamics and loss landscapes of contrastive learning (Wen & Li, 2021; Jing et al., 2022; Tian, 2022), non-contrastive learning (Tian et al., 2021; Pokle et al., 2022; Wen & Li, 2022), and masked prediction (Xiong et al., 2020; Huang et al., 2020).

Representation learning has adopted the idea of compression. Classical representation learning aims to extract the low-dimensional manifold on which the data resides (Belkin & Niyogi, 2003; 2004; Bengio et al., 2004). More recently, Yu et al. (2020) proposes maximal coding rate reduction (MCR$^2$), which learns an explicitly meaningful low-dimensional representation by maximizing the rate reduction, defined as the difference in coding rate, *i.e.* the number of bits needed to encode the embedding before and after a certain partition (*e.g.* according to different classes). Yu et al. (2020) showed that MCR$^2$ is closely related to contrastive learning. Follow-up work includes Chan et al. (2022); Dai et al. (2022).

More closely related to this work is the line of work that formulates contrastive learning as a Laplacian operator over the augmentation graph. The idea of studying data augmentation from a kernel perspective was first explored in Mroueh et al. (2015); Raj et al. (2017); Dao et al. (2019). The augmentation graph was defined in HaoChen et al. (2021) in whose Theorem 4.2 proved a generalization bound for the spectral contrastive loss that includes a Rademacher complexity term *w.r.t.* $a_1, \cdots, a_N$, which is incurred by sampling only one $a$ from each $x$ (see their Definition D.3). Then, Saunshi et al. (2022) pointed out that this model-class-free bound could be vacuous with a hypercube construction. One thing to note is that while Saunshi et al. (2022) claimed that the bound in HaoChen et al. (2021) is "function class independent", actually it is not.[*] As a response to Saunshi et al. (2022), HaoChen & Ma (2023) included the effect of the encoder's inductive bias into their generalization bound. Then, Wang et al. (2023) connected contrastive learning to message passing on the augmentation graph.

In Section 6 we highlighted two related recent papers. Wang et al. (2022b) studied a similar RKHS as the $\mathcal{H}_\Gamma$ defined in this work. In particular, the kernels $k_x$ and $k_z$ in their work are similar to the kernels $K_A$ and $K_X$ in this work. However, their work studies the conditional moment model, which is a completely different problem than ours. Their bounds are also distinct from ours and are with stronger assumptions. Nevertheless, the fact that they derived similar kernels to ours from a completely different angle suggests a strong innate connection, which alludes to the role of augmentation in self-supervised pretraining, namely, it defines a ordered set of features as we articulated after Eqn. (6).

---

[*]Theorem 4.2 in the NeurIPS version and the arXiv [v7] version of HaoChen et al. (2021) have different forms. We have confirmed with the authors that this theorem does depend on the complexity of the function class.

Another related work is Cabannes et al. (2023), who studied the same problem as this work. The difference is that their work uses a pre-defined kernel, while the kernels $K_A$ and $K_X$ in this work are completely induced by the augmentation, *i.e.* there would be not be a kernel without the augmentation. Their work studied regularized kernel regression, while this work studies unregularized kernel regression. Moreover, the bounds derived in their work and ours are significantly different. We would also like to credit Cabannes et al. (2023) for studying the upstream-to-downstream distribution shift which provides inspiring preliminary analysis. Last but not least, the follow-up work by Zhai et al. (2024) applies the framework developed in this paper to a more general semi-supervised kernel learning setting.

Finally, similar to Saunshi et al. (2022), we point out one common issue of existing learning guarantees for contrastive learning: Most of them require the samples from the same downstream class to have some type of "overlap" in the augmentation graph, but we find that sometimes these assumptions could be too strong to be reasonable. For instance, Wang et al. (2022a) assumed intra-class connectivity, *i.e.* the subgraph of the augmentation graph given by each downstream class is connected; Huang et al. (2023) assumed that each downstream class has a "main part" of samples, in which any two samples can be augmented to be close in Euclidean distance. We find these two assumptions hard to be reasonable for any practical scenario.

## C VARIATIONAL OBJECTIVES THAT EXTRACT THE TOP-d EIGENSPACE

In Proposition 4, we demonstrated the optimality of the top-$d$ eigenspace. The question is: How to extract the top-$d$ eigenspace? We cannot do so via explicit eigendecomposition of the kernel in most real tasks, where the kernel is huge and sparse. Instead, we will show that several pretraining objectives can extract the top-$d$ eigenspace assuming optimization converges to the global minima, because these objective are uniquely optimized by the top-$d$ eigenspace.

**Contrastive learning.** Let us warm up with contrastive learning, whose connection to kernels has been studied in prior work, among which HaoChen et al. (2021) introduced the *spectral contrastive loss*:

$$\mathcal{L}_{\text{SCL}}(\hat{\Phi}) = -2\mathbb{E}\left[\langle\hat{\Phi}(A), \hat{\Phi}(A^+)\rangle\right] + \mathbb{E}\left[\langle\hat{\Phi}(A), \hat{\Phi}(A^-)\rangle^2\right], \tag{13}$$

where $(A, A^+)$ is a positive pair that is jointly sampled from $P_X(a, a^+)$, and $A^-$ is a negative sample that is sampled from $P_A$ independently of $A$. It has been shown that:

**Theorem 3.** *(HaoChen et al., 2021) Eqn. (13) is minimized by the top-$d$ eigenspace, and is uniquely minimized by the top-$d$ eigenspace if $\lambda_{d+1} < \lambda_d$.*

Here, "minimized by the top-$d$ eigenspace" means that the objective is minimized when $\hat{\Phi}$ spans the top-$d$ eigenspace, and "uniquely minimized" means that this condition is necessary. The proof is based on the following Eckart-Young-Mirsky Theorem:

**Theorem 4** (Eckart-Young-Mirsky). *Given a matrix $A \in \mathbb{R}^{m \times n}$, consider the following optimization problem:*

$$\min_{B \in \mathbb{R}^{m \times n}, \text{rank}(B) \leq k} \|A - B\|_F^2 \tag{14}$$

*where $1 \leq k \leq \min\{m, n\}$. Let the SVD of $A$ and $B$ be $A = U\Sigma V^\top$ and $B = U_1 \Sigma_1 V_1^\top$, where $\Sigma = \text{diag}(\sigma_1, \cdots, \sigma_{\min\{m,n\}})$, and $\sigma_1 \geq \sigma_2 \geq \cdots \geq \sigma_{\min\{m,n\}}$. Then $B$ is a minimizer of the above problem if $U_1$ is the first $k$ columns of $U$, $V_1$ is the first $k$ columns of $V$, and $\Sigma_1$ is the upper left $k \times k$ block of $\Sigma$. Moreover, if $\sigma_{k+1} < \sigma_k$, then this is the unique minimizer.*

To make this paper self-contained, here we recap the proof of Theorem 3 in HaoChen et al. (2021):

*Proof.* Define matrix $C$ as: $\hat{\Phi} = C\Phi^*$, and matrix $B$ as $B = C^\top C$. Then, $\langle\hat{\Phi}, \hat{\Phi}\rangle = \Phi^{*\top} B \Phi^*$. Let the SVD of $C$ be $C = U\Sigma V^\top$. Denote $D_\lambda = \text{diag}(\lambda_1, \lambda_2, \cdots)$. Then, the spectral contrastive loss is equivalent to

$$\mathcal{L}_{\text{SCL}}(\hat{\Phi}) = -2\sum_i \lambda_i b_{i,i} + \sum_{i,j} b_{i,j}^2 = \|B - D_\lambda\|_F^2 - \|D_\lambda\|_F^2. \tag{15}$$

So it suffices to minimize $\|\boldsymbol{B} - \boldsymbol{D}_\lambda\|_F^2$ where $\mathrm{rank}(\boldsymbol{B}) \leq d$. By Theorem 4, this is minimized if $\boldsymbol{V}(i,j) = \delta_{i,j}$ and $\boldsymbol{\Sigma} = \mathrm{diag}(\sqrt{\lambda_i})$, which means that $\hat{\Phi} = \boldsymbol{U}\mathrm{diag}(\sqrt{\lambda_1}, \cdots, \sqrt{\lambda_d})\boldsymbol{\Phi}^*$ for some orthogonal matrix $\boldsymbol{U} \in \mathbb{R}^{d \times d}$. Moreover, by Thm. 4, this is the unique minimizer if $\lambda_{d+1} < \lambda_d$. $\square$

**CLIP.** CLIP (Radford et al., 2021) maximizes the similarity between two samples from different modalities. There is an objective similar to the spectral contrastive loss, which we term the *spectral CLIP loss*:

$$\mathcal{L}_{\mathrm{SCLIP}}(\hat{\Phi}, \hat{\Xi}) = -2\mathbb{E}[\langle \hat{\Phi}(A), \hat{\Xi}(X^+)\rangle] + \mathbb{E}[\langle \hat{\Phi}(A), \hat{\Xi}(X^-)\rangle^2], \tag{16}$$

where $\hat{\Phi}$ and $\hat{\Xi}$ are encoders for different modalities (*e.g.* image and text). $(A, X^+)$ is a positive pair, while $X^-$ is a negative sample independent of $A$. This loss also appeared in Wang et al. (2022b) (see their Section 4.1). We will show that this loss also extracts the top-$d$ eigenspace.

**Theorem 5.** *Eqn. (16) is minimized by the top-d eigenspace, and is uniquely minimized by the top-d eigenspace if $\lambda_{d+1} < \lambda_d$.*

*Proof.* Let $\hat{\Phi} = \boldsymbol{C}\boldsymbol{\Phi}^*$ and $\hat{\Xi} = \boldsymbol{S}\boldsymbol{\Psi}^*$. Let $\Xi^\dagger = \Gamma\hat{\Xi} = \boldsymbol{S}\boldsymbol{D}_\lambda^{1/2}\boldsymbol{\Phi}^*$. Then, we have

$$\mathbb{E}[\langle \hat{\Phi}(A), \hat{\Xi}(X^+)\rangle] = \iint \left\langle \hat{\Phi}(a), \hat{\Xi}(x) \right\rangle p(x|a)p(a)dadx$$

$$= \int \left\langle \hat{\Phi}(a), \int \hat{\Xi}(x)p(x|a)dx \right\rangle p(a)da$$

$$= \int \left\langle \hat{\Phi}(a), \Xi^\dagger(a) \right\rangle p(a)da = \mathrm{Tr}\left(\boldsymbol{C}^\top \boldsymbol{S}\boldsymbol{D}_\lambda^{1/2}\right).$$

We also have

$$\mathbb{E}[\langle \hat{\Phi}(A), \hat{\Xi}(X^-)\rangle^2] = \iint \left(\sum_{k=1}^d \hat{\Phi}(a)_k\hat{\Xi}(x)_k\right)^2 p(a)p(z)dadz$$

$$= \sum_{k,l=1}^d \left(\int \hat{\Phi}(a)_k\hat{\Phi}(a)_l p(a)da\right)\left(\int \hat{\Xi}(x)_k\hat{\Xi}(x)_l p(x)dx\right)$$

$$= \mathrm{Tr}\left(\boldsymbol{C}\boldsymbol{C}^\top \boldsymbol{S}\boldsymbol{S}^\top\right) = \left\|\boldsymbol{C}^\top \boldsymbol{S}\right\|_F^2.$$

Therefore, the spectral CLIP loss is equivalent to

$$\mathcal{L}_{\mathrm{SCLIP}}(\hat{\Phi}, \hat{\Xi}) = \left\|\boldsymbol{C}^\top \boldsymbol{S}\right\|_F^2 - 2\,\mathrm{Tr}\left(\boldsymbol{C}^\top \boldsymbol{S}\boldsymbol{D}_\lambda^{1/2}\right) = \left\|\boldsymbol{C}^\top \boldsymbol{S} - \boldsymbol{D}_\lambda^{1/2}\right\|_F^2 - \left\|\boldsymbol{D}_\lambda^{1/2}\right\|_F^2. \tag{17}$$

Let the SVD of $\boldsymbol{C}$ be $\boldsymbol{C} = \boldsymbol{U}\boldsymbol{\Sigma}\boldsymbol{V}^\top$. Let $\boldsymbol{M} = \boldsymbol{\Sigma}\boldsymbol{U}^\top \boldsymbol{S}$. Then, $\boldsymbol{C}^\top \boldsymbol{S} = \boldsymbol{V}\boldsymbol{M}$. It is well known in linear regression that Eqn. (17) is minimized when $\boldsymbol{M} = \left(\boldsymbol{V}^\top \boldsymbol{V}\right)^{-1} \boldsymbol{V}^\top \boldsymbol{D}_\lambda^{1/2} = \boldsymbol{V}^\top \boldsymbol{D}_\lambda^{1/2}$, so

$$\min_{\hat{\Phi}, \hat{\Xi}} \mathcal{L}_{\mathrm{SCLIP}}(\hat{\Phi}, \hat{\Xi}) = \min_{\boldsymbol{V}} \left\|(\boldsymbol{I} - \boldsymbol{V}\boldsymbol{V}^\top)\boldsymbol{D}_\lambda^{1/2}\right\|_F^2 - \left\|\boldsymbol{D}_\lambda^{1/2}\right\|_F^2.$$

By Theorem 4, this is minimized when the top $d$ rows of $\boldsymbol{V}$ is an orthogonal matrix, and this condition is also necessary if $\lambda_{d+1} < \lambda_d$. This proves the result. Moreover, we can see that $\hat{\Xi}$ is the minimizer if it extracts the top-$d$ eigenspace of $\boldsymbol{\Psi}^*$. $\square$

**Regularized Barlow Twins.** Barlow twins (Zbontar et al., 2021) is a non-contrastive learning method that trains without negative sampling. We consider the following *regularized Barlow Twins loss*:

$$\mathcal{L}_{\mathrm{RBT}}(\hat{\Phi}; \alpha, \beta) = \sum_{k=1}^d \left(\mathbb{E}\left[\hat{\Phi}(A)_k\hat{\Phi}(A^+)_k\right] - 1\right)^2 + \alpha \sum_{k \neq l} \left(\mathbb{E}\left[\hat{\Phi}(A)_k\hat{\Phi}(A^+)_l\right]\right)^2 + \beta\mathbb{E}\left[\left\|\hat{\Phi}(A)\right\|_2^2\right]. \tag{18}$$

When the regularization term $\beta$ is close to 0 so that the last term is much smaller than the first two, the above can be considered as a constrained optimization task:

$$\min_\theta \quad \mathbb{E}\left[\left\|\hat{\Phi}(A)\right\|_2^2\right] \qquad \text{s.t.} \qquad \mathcal{L}_{\mathrm{RBT}}(\hat{\Phi}; \alpha, 0) = 0. \tag{19}$$

**Theorem 6.** *The constrained optimization problem Eqn. (19) is minimized by the top-$d$ eigenspace, and is uniquely minimized by the top-$d$ eigenspace if $\lambda_{d+1} < \lambda_d$.*

*Proof.* Define matrix $\boldsymbol{C} = (c_{i,j})$ as: $\hat{\Phi} = \boldsymbol{C}\boldsymbol{\Phi}^*$. Then, for all $k, l \in [d]$, there is

$$\mathbb{E}[\hat{\Phi}(A)_k \hat{\Phi}(A^+)_l] = \sum_i \lambda_i c_{k,i} c_{l,i}.$$

Thus, $\mathcal{L}_{\mathrm{RBT}}(\hat{\Phi}; \alpha, 0) = 0$ is equivalent to $\boldsymbol{C}\boldsymbol{D}_\lambda \boldsymbol{C}^\top = \boldsymbol{I}$. Meanwhile, $\mathbb{E}\left[\left\|\hat{\Phi}(A)\right\|_2^2\right] = \|\boldsymbol{C}\|_F^2$. So the optimization problem is equivalent to minimizing $\|\boldsymbol{C}\|_F^2$ subject to $\boldsymbol{C}\boldsymbol{D}_\lambda\boldsymbol{C}^\top = \boldsymbol{I}$.

Let $\boldsymbol{G} = \boldsymbol{C}\sqrt{\boldsymbol{D}_\lambda} = (g_{k,i})$, then $\boldsymbol{G}\boldsymbol{G}^\top = \sum_k \boldsymbol{g}_k \boldsymbol{g}_k^\top = \boldsymbol{I}$, where $\boldsymbol{g}_k = [g_{k,1}, \cdots, g_{k,d}]$. For all $j \in [d]$, define $\boldsymbol{M}_j = \sum_{k=1}^j \boldsymbol{g}_k \boldsymbol{g}_k^\top$. Then, $\mathrm{rank}(\boldsymbol{M}_j) \leq j$, and all eigenvalues of $\boldsymbol{M}_j$ belong to $[0, 1]$. So the sum of eigenvalues of $\boldsymbol{M}_j$ is at most $j$, which implies that

$$\sum_{k=1}^j \boldsymbol{g}_j^\top \boldsymbol{g}_j = \sum_{k=1}^j \mathrm{Tr}\left(\boldsymbol{g}_j^\top \boldsymbol{g}_j\right) = \sum_{k=1}^j \mathrm{Tr}\left(\boldsymbol{g}_j \boldsymbol{g}_j^\top\right) = \mathrm{Tr}(\boldsymbol{M}_j) \leq j.$$

Denote $\frac{1}{\lambda_0} = 0$. Thus, by Abel transformation, we have

$$\|\boldsymbol{C}\|_F^2 = \sum_i \frac{1}{\lambda_i} \boldsymbol{g}_i^\top \boldsymbol{g}_i = \sum_{j \geq 0} \left(d - \sum_{k=1}^j \boldsymbol{g}_k^\top \boldsymbol{g}_k\right)\left(\frac{1}{\lambda_{j+1}} - \frac{1}{\lambda_j}\right) \geq \sum_{j=0}^{d-1} (d-j)\left(\frac{1}{\lambda_{j+1}} - \frac{1}{\lambda_j}\right) = \sum_{i=1}^d \frac{1}{\lambda_i}.$$

A sufficient condition of achieving the above equality is $\sum_{k=1}^j \boldsymbol{g}_k^\top \boldsymbol{g}_k = j$ for all $j \in [d]$, and it is also necessary if $\lambda_{d+1} < \lambda_d$. This condition is equivalent to $\boldsymbol{g}_1, \cdots, \boldsymbol{g}_d$ forming an orthonormal basis of $\mathbb{R}^d$, *i.e.* $\hat{\Phi}$ extracts the top-$d$ eigenspace. $\qquad\square$

**Maximizing the ratio trace.** Now we demonstrate the connection of contrastive learning and Barlow Twins to maximizing the ratio trace defined in Definition 3. We start with the regularized Barlow Twins loss, which we have shown is equivalent to minimizing $\|\boldsymbol{C}\|_F^2$ subject to $\boldsymbol{C}\boldsymbol{D}_\lambda\boldsymbol{C}^\top = \boldsymbol{I}$. Now, observe that $\boldsymbol{G} = \boldsymbol{C}\boldsymbol{C}^\top$, and $\boldsymbol{F} = \boldsymbol{C}\boldsymbol{D}_\lambda\boldsymbol{C}^\top$. Thus, regularized Barlow Twins is essentially minimizing $\mathrm{Tr}(\boldsymbol{F}^{-1}\boldsymbol{G})$, which is similar but not equivalent to maximizing $\mathrm{Tr}(\boldsymbol{G}^{-1}\boldsymbol{F})$.

Now we consider a variant of VICReg (Bardes et al., 2022), also given by Cabannes et al. (2023, Eqn. (2)):

$$\mathcal{L}_{\mathrm{VICReg}}(\hat{\Phi}; \beta) = \left\|\mathbb{E}\left[\hat{\Phi}(A)\hat{\Phi}(A)^\top\right] - \boldsymbol{I}\right\|_F^2 + \beta \mathbb{E}\left[\left\|\hat{\Phi}(A) - \hat{\Phi}(A^+)\right\|_2^2\right]. \tag{20}$$

When the regularization term $\beta$ is close to 0, we have $\mathbb{E}\left[\hat{\Phi}(A)\hat{\Phi}(A)^\top\right] = \boldsymbol{G} = \boldsymbol{I}$. And we minimize $\mathbb{E}\left[\left\|\hat{\Phi}(A) - \hat{\Phi}(A^+)\right\|_2^2\right]$, which is equivalent to $-2\mathbb{E}\left[\mathrm{Tr}\left(\hat{\Phi}(A)\hat{\Phi}(A^+)\right)\right] + c = -2\mathrm{Tr}(\boldsymbol{F}) + c$ for some constant $c$. Thus, this objective is equivalent to maximizing $\mathrm{Tr}(\boldsymbol{G}^{-1}\boldsymbol{F})$, the ratio trace, when $\beta \approx 0$. And when $\beta = 1$, Eqn. (20) is equivalent to the spectral contrastive loss.

# D    PROOFS FOR SECTION 2

**$\Gamma^*\Gamma$ and $\Gamma\Gamma^*$ are integral operators.**

$$\begin{cases} (\Gamma_{a\to x}\Gamma_{x\to a}f)(x) = (\Gamma^*\Gamma f)(x) = \int K_X(x, x')f(x')p(x')dx'; \\ (\Gamma_{x\to a}\Gamma_{a\to x}g)(a) = (\Gamma\Gamma^*g)(a) = \int K_A(a, a')g(a')p(a')da'. \end{cases} \tag{21}$$

*Proof.* We only show the first equation, and the second one can be proved in the same way.

$$(\Gamma^*\Gamma f)(x) = \Gamma^* \left( \int f(x')p(x'|a)dx' \right) = \int \left( \int f(x')p(x'|a)dx' \right) p(a|x)da$$

$$= \iint f(x')p(a|x)p(x'|a)dadx' = \iint f(x')\frac{p(a|x)p(a|x')}{p(a)}p(x')dadx'$$

$$= \int K_X(x,x')f(x')p(x')dx'.$$

$\square$

**Proposition 1 (Duality).** *$\Gamma\Gamma^*$ shares the same non-zero eigenvalues as $\Gamma^*\Gamma$, and there exist eigenfunctions $\{\phi_i\}$ of $\Gamma\Gamma^*$ that form an orthonormal basis of $L^2(P_\mathcal{A})$, such that for any $\lambda_i > 0$,*

$$\psi_i = \lambda_i^{-1/2}\Gamma^*\phi_i \quad and \quad \phi_i = \lambda_i^{-1/2}\Gamma\psi_i, \tag{22}$$

*and we also have the following spectral decomposition of the Radon-Nikodym derivative:*

$$\frac{dP_{\mathcal{A}\mathcal{X}}}{d(P_\mathcal{A} \otimes P_\mathcal{X})} = \frac{p(a,x)}{p(a)p(x)} = \sum_i \lambda_i^{1/2}\phi_i(a)\psi_i(x). \tag{23}$$

*Proof.* Suppose $\lambda_i, \psi_i(x)$ is a pair of eigenvalue and eigenfunction of $\Gamma^*\Gamma$, and $\lambda_i > 0$. Then, we have $\Gamma\Gamma^*\Gamma\psi_i = \lambda_i\Gamma\psi_i$, which means that $\Gamma\psi_i$ is an eigenfunction of $\Gamma\Gamma^*$ with eigenvalue $\lambda_i$. The $\lambda_i^{-1/2}$ is used for normalization. To see this, let $\phi_i = \lambda_i^{-1/2}\Gamma\psi_i$. Then, we have

$$\langle\phi_i, \phi_j\rangle_{P_\mathcal{A}} = \lambda_i^{-1/2}\lambda_j^{-1/2}\langle\Gamma\psi_i, \Gamma\psi_j\rangle_{P_\mathcal{A}}$$

$$= \lambda_i^{-1/2}\lambda_j^{-1/2}\langle\Gamma^*\Gamma\psi_i, \psi_j\rangle_{P_\mathcal{X}}$$

$$= \lambda_i^{-1/2}\lambda_j^{-1/2}\langle\lambda_i\psi_i, \psi_j\rangle_{P_\mathcal{X}} = \delta_{i,j}.$$

We can prove the reverse direction similarly. And for any fixed $x$, there is

$$\left\langle \frac{p(a,x)}{p(a)p(x)}, \phi_i \right\rangle_{P_\mathcal{A}} = \int \frac{p(a,x)}{p(a)p(x)}\phi_i(a)p(a)da = \int p(a|x)\phi_i(a)da = \sqrt{\lambda_i}\psi_i(x). \tag{24}$$

which implies Eqn. (23). $\square$

**Basic properties of $\mathcal{H}_\Gamma$.**

(i) $K_X$ is the reproducing kernel of $\mathcal{H}_\Gamma$, such that for all $f \in \mathcal{H}_\Gamma$, $f(x) = \langle f, K_X(x,\cdot)\rangle_{\mathcal{H}_\Gamma}$.

(ii) $\mathcal{H}_\Gamma = R(\Gamma^*)$.

(iii) $\mathcal{H}_\Gamma$ is isometric to $\text{span}(\{\phi_i\}_{\lambda_i>0})$, a subspace of $L^2(P_\mathcal{A})$, and $\|f\|_{\mathcal{H}_\Gamma} = \inf_{g:f=\Gamma^*g}\|g\|_{P_\mathcal{A}}$.

(iv) For any $f^* \in \mathcal{F}_B(\Gamma;\epsilon) \subset R(\Gamma^*)$, let $f^* = \sum_i u_i\psi_i$. Define $g_0 := \sum_i \lambda_i^{-1/2}u_i\phi_i$. Then, $g_0$ must satisfy Eqn. (1), so we can choose $g^* = g_0$, in which case Eqn. (1) is equivalent to:

$$\langle g^*, (I - \Gamma\Gamma^*)g^*\rangle_{P_\mathcal{A}} \leq \epsilon\|g^*\|_{P_\mathcal{A}}^2 \Leftrightarrow \sum_i \frac{1-\lambda_i}{\lambda_i}u_i^2 \leq \epsilon\sum_i \frac{1}{\lambda_i}u_i^2. \tag{25}$$

*Proof.* (i) First, note that $\mathcal{H}_\Gamma = \{\sum_{i:\lambda_i>0} a_i\boldsymbol{e}_i \mid \sum_i a_i^2 < \infty\}$ where $\boldsymbol{e}_i = \lambda_i^{-1/2}\psi_i$, so it is isomorphic to $\ell^2((a_i)_{i:\lambda_i>0})$ and is thus a Hilbert space. Then, $K_X(x,x') = \sum_i \lambda_i\psi_i(x)\psi_i(x')$. For any $f \in \mathcal{H}_\Gamma$, let $f = \sum_i u_i\psi_i$, then

$$\langle f(x'), K_X(x,x')\rangle_{\mathcal{H}_\Gamma} = \sum_i \frac{1}{\lambda_i}u_i(\lambda_i\psi_i(x)) = \sum_i u_i\psi_i(x) = f(x).$$

(ii) For any $f = \sum_i u_i \psi_i \in \mathcal{H}_\Gamma$, there is $\sum_i \lambda_i^{-1} u_i^2 < \infty$ by definition. So for any $\lambda_i = 0$, there must be $u_i = 0$. Let $g = \sum_i \lambda_i^{-1/2} u_i \psi_i$. Then, $\|g\|_{P_\mathcal{A}}^2 = \sum_i \lambda_i^{-1} u_i^2 < \infty$, meaning that $g \in L^2(P_\mathcal{A})$. And there is $f = \Gamma^* g$, so $f \in R(\Gamma^*)$, which implies that $\mathcal{H}_\Gamma \subseteq R(\Gamma^*)$. Meanwhile, for any $f = \Gamma^* g \in R(\Gamma^*)$, let $g = \sum_i v_i \phi_i$, then $\sum_i v_i^2 < \infty$. Then, $f = \sum_i \lambda_i^{1/2} v_i \psi_i$ by duality, so $\sum_i \lambda_i^{-1} (\lambda_i^{1/2} v_i)^2 < \infty$, meaning that $f \in \mathcal{H}_\Gamma$, so $R(\Gamma^*) \subseteq \mathcal{H}_\Gamma$.

(iii) For any $f = \sum_i u_i \psi_i \in \mathcal{H}_\Gamma$, let $g = \sum_i \lambda_i^{-1/2} u_i \psi_i$. By the proof of (ii) we know that $f \mapsto g$ is bijective, and $\|f\|_{\mathcal{H}_\Gamma} = \|g\|_{P_\mathcal{A}}$. Moreover, $g \in \text{span}(\{\phi_i\}_{\lambda_i > 0})$.

(iv) Let $g^* = \sum_i v_i \phi_i$. Then, since we have $f^* = \Gamma^* g^* = \sum_i \lambda_i^{1/2} v_i \psi_i$, for any $\lambda_i > 0$, there is $v_i = \lambda^{-1/2} u_i$; and for any $\lambda_i = 0$, there is $u_i = 0$. Let $g^* = g_0 + g_1$, where $g_0 = \sum_i \lambda_i^{-1/2} u_i \phi_i$, and $g_1 \perp g_0$ and $\Gamma^* g_1 = 0$. By duality, $\Gamma\Gamma^* g_0$ belongs to the linear span of $\{\phi_i\}_{\lambda_i > 0}$, so $g_1 \perp \Gamma\Gamma^* g_0$. As we will show later, Eqn. (1) is equivalent to Eqn. (25), which is the random walk normalized Laplacian over the augmentation graph (Chung, 1997, Section 1.2). This is equivalent to $\langle g^*, (I - \Gamma\Gamma^*) g^* \rangle_{P_\mathcal{A}} \le \epsilon \|g^*\|_{P_\mathcal{A}}^2$, which is further equivalent to $\langle g_0, (I - \Gamma\Gamma^*) g_0 \rangle_{P_\mathcal{A}} + \|g_1\|_{P_\mathcal{A}}^2 \le \epsilon(\|g_0\|_{P_\mathcal{A}}^2 + \|g_1\|_{P_\mathcal{A}}^2)$ (note that $\Gamma\Gamma^* g_1 = 0$). This implies that $\langle g_0, (I - \Gamma\Gamma^*) g_0 \rangle_{P_\mathcal{A}} \le \epsilon \|g_0\|_{P_\mathcal{A}}^2$, i.e. $g_0$ satisfies Eqn. (1). Since we also have $f^* = \Gamma^* g_0$, $g_0$ must satisfy Assumption 1, so we can choose $g^* = g_0$.

Next, to show the equivalence to Eqn. (25), We just need to show that $\langle g^*, (I - \Gamma\Gamma^*) g^* \rangle_{P_\mathcal{A}} = \frac{1}{2} \mathbb{E}_{X \sim P_\mathcal{X}} \mathbb{E}_{A, A' \sim p(\cdot|X)} \left[ (g^*(A) - g^*(A'))^2 \right]$. And indeed, we have:

$$
\begin{aligned}
\langle g^*, (I - \Gamma\Gamma^*) g^* \rangle_{P_\mathcal{A}} &= \left\langle g^*, g^* - \int g^*(a') K_A(\cdot, a') p(a') da' \right\rangle_{P_\mathcal{A}} \\
&= \|g^*\|_{P_\mathcal{A}}^2 - \iint g(a) g(a') \frac{\int p(a|x) p(a'|x) p(x) dx}{p(a) p(a')} p(a') p(a) da da' \\
&= \frac{1}{2} \mathbb{E}[g^*(A)^2] + \frac{1}{2} \mathbb{E}[g^*(A')^2] - \frac{1}{2} \mathbb{E}_{X \sim P_\mathcal{X}} \mathbb{E}_{A, A' \sim p(\cdot|X)} [2 g^*(A) g^*(A')] \\
&= \frac{1}{2} \mathbb{E}_{X \sim P_\mathcal{X}} \mathbb{E}_{A, A' \sim p(\cdot|X)} \left[ (g^*(A) - g^*(A'))^2 \right],
\end{aligned}
$$

as desired. □

# E PROOFS FOR SECTION 3

## E.1 LOCAL GAUSSIAN COMPLEXITY AND LOCALIZED RADEMACHER COMPLEXITY

We first provide the definition of the two complexities we will use in our analysis. For a function $f$, let $\|f\|_n^2 := \frac{1}{n} \sum_{i=1}^n f(\tilde{x}_i)^2$ be its mean on the downstream samples.

**Definition 4.** *(Wainwright, 2019, Eqns. (13.16) & (14.3)) For any $B, \epsilon > 0$, define*

$$
\mathcal{F}_0 := \left\{ f_1 - f_2 \,\middle|\, f_i \in \mathcal{H}_{\hat{\Psi}}, \|f_i\|_{\mathcal{H}_\Gamma} \le \frac{B}{\sqrt{1-\epsilon}} \right\} = \left\{ f \in \mathcal{H}_{\hat{\Psi}} \,\middle|\, \|f\|_{\mathcal{H}_\Gamma} \le \frac{2B}{\sqrt{1-\epsilon}} \right\}. \tag{26}
$$

*Then, the local Gaussian complexity around $f_{\hat{\Psi}}$ at scale $\delta > 0$ is given by*

$$
\mathcal{G}_n(\delta; \mathcal{F}_0) := \mathop{\mathbb{E}}_{\omega_1, \cdots, \omega_n} \left[ \sup_{f \in \mathcal{F}_0, \|f\|_n \le \delta} \left| \frac{1}{n} \sum_{i=1}^n \omega_i f(\tilde{x}_i) \right| \right], \tag{27}
$$

*where $\omega_1, \cdots, \omega_n$ are i.i.d. $\mathcal{N}(0, 1)$ variates. And define*

$$
\mathcal{F}_* := \left\{ f = f_1 + \alpha f^* \,\middle|\, \alpha \in [-1, 1], f_1 \in \mathcal{H}_{\hat{\Psi}}, \|f_1\|_{\mathcal{H}_\Gamma} \le \frac{B}{\sqrt{1-\epsilon}} \right\}. \tag{28}
$$

*Then, the localized population Rademacher complexity of radius $\delta > 0$ is given by*

$$
\bar{\mathfrak{R}}_n(\delta; \mathcal{F}_*) := \mathop{\mathbb{E}}_{\sigma_1, \cdots, \sigma_n, x_1, \cdots, x_n} \left[ \sup_{f \in \mathcal{F}_*, \|f\|_{P_\mathcal{X}} \le \delta} \left| \frac{1}{n} \sum_{i=1}^n \sigma_i f(x_i) \right| \right], \tag{29}
$$

*where $\sigma_1, \cdots, \sigma_n$ are i.i.d. Rademacher variables taking values in $\{-1, +1\}$ equiprobably.*

Our master plan is to apply Theorems 13.13 and 14.1 of Wainwright (2019) to $f_{\hat{\Psi}} = \Gamma^*(\Pi_{\hat{\Phi}} g^*)$, where $\Pi_{\hat{\Phi}}$ is the projection operator onto $\hat{\Phi}$ in $L^2(P_{\mathcal{X}})$, and $f_{\hat{\Psi}}$ is the projection of $f^*$ onto $\mathcal{H}_{\hat{\Psi}}$ w.r.t. $\langle \cdot, \cdot \rangle_{\mathcal{H}_\Gamma}$. Therefore, we need to bound $\mathcal{G}_n(\delta; \mathcal{F}_0)$ and $\bar{\mathfrak{R}}_n(\delta; \mathcal{F}_*)$. We start with the following uniform bound:

**Proposition 7.** *If $f = \Gamma^* g$, and $\|g\|_{P_{\mathcal{A}}} \leq T$, then $|f(x)| \leq \kappa T$ for all $x$.*

*Proof.* By Eqn. (4), we have $p(a|x) = \sum_i \sqrt{\lambda_i} \phi_i(a) \psi_i(x) p(a)$. For any $g = \sum_i u_i \phi_i \in L^2(P_{\mathcal{A}})$ such that $\|g\|_{P_{\mathcal{A}}} \leq T$, $(\Gamma^* g)(x) = \int g(a) p(a|x) da = \sum_i \sqrt{\lambda_i} u_i \psi_i(x)$. Then, by Cauchy-Schwarz inequality, we have for all $x$, $f(x)^2 = (\Gamma^* g)(x)^2 \leq (\sum_i \lambda_i \psi_i(x)^2)(\sum_i u_i^2) \leq \kappa^2 T^2$. $\square$

This proposition immediately implies that $f^*$ and $f_{\hat{\Psi}}$ are uniformly bounded:

**Corollary 8.** *For any $f^* \in \mathcal{F}_B(\Gamma; \epsilon)$, Eqn. (6) ensures that $\|g^*\|_{P_{\mathcal{A}}}^2 \leq \frac{B^2}{1-\epsilon}$, so $|f^*(x)| \leq \frac{\kappa B}{\sqrt{1-\epsilon}}$ for all $x$. Moreover, $\|\Pi_{\hat{\Phi}} g^*\|_{P_{\mathcal{A}}} \leq \|g^*\|_{P_{\mathcal{A}}}$ implies that $\|f_{\hat{\Psi}}\|_{\mathcal{H}_\Gamma} \leq \frac{B}{\sqrt{1-\epsilon}}$, and $|f_{\hat{\Psi}}(x)| \leq \frac{\kappa B}{\sqrt{1-\epsilon}}$ for all $x$.*

We will also use the following simple result in linear algebra:

**Lemma 9.** *Let $\boldsymbol{D}_\lambda = \mathrm{diag}(\lambda_1, \lambda_2, \cdots)$ where $\lambda_1 \geq \lambda_2 \geq \cdots \geq 0$ and $\lambda_i \to 0$. Let $\boldsymbol{Q}$ be a matrix with $d$ rows that are unit vectors. Then, $\mathrm{Tr}(\boldsymbol{Q} \boldsymbol{D}_\lambda \boldsymbol{Q}^\top) \leq \lambda_1 + \cdots + \lambda_d$.*

*Proof.* Let $\boldsymbol{q}_i$ be the $i$-th column of $\boldsymbol{Q}$. Then for all $j \in [d]$, there is $\sum_{i=1}^j \boldsymbol{q}_i^\top \boldsymbol{q}_i \leq j$. And for $j > d$, $\sum_{i=1}^j \boldsymbol{q}_i^\top \boldsymbol{q}_i \leq d$. Thus, using Abel transformation, we have

$$\mathrm{Tr}(\boldsymbol{Q} \boldsymbol{D}_\lambda \boldsymbol{Q}^\top) = \mathrm{Tr}(\boldsymbol{D}_\lambda \boldsymbol{Q}^\top \boldsymbol{Q}) = \sum_{i=1}^\infty \lambda_i \boldsymbol{q}_i^\top \boldsymbol{q}_i = \sum_{j=1}^\infty \left( \sum_{i=1}^j \boldsymbol{q}_i^\top \boldsymbol{q}_i \right) (\lambda_j - \lambda_{j+1}) \leq \sum_{i=1}^d \lambda_i,$$

which proves the assertion. $\square$

This result has many implications. For instance, for any rank-$d$ subspace of $\mathcal{H}_\Gamma$, its trace (the sum of its eigenvalues) is at most $S_\lambda(d)$.

Now, let us bound $\mathcal{G}_n(\delta; \mathcal{F}_0)$ with the following result:

**Lemma 10.** *(Application of Wainwright (2019, Lemma 13.22)) Let $\mathcal{H}$ be an RKHS with reproducing kernel $K$. Given samples $\tilde{x}_1, \cdots, \tilde{x}_n$, let $\boldsymbol{K}$ be the normalized kernel matrix with entries $\boldsymbol{K}(i, j) = K(\tilde{x}_i, \tilde{x}_j)/n$. Let $\mu_1 \geq \cdots \geq \mu_n \geq 0$ be the eigenvalues of $\boldsymbol{K}$. Then for all $\delta > 0$, we have*

$$\mathbb{E} \left[ \sup_{\|f\|_{\mathcal{H}} \leq T, \|f\|_n \leq \delta} \left| \frac{1}{n} \sum_{i=1}^n \omega_i f(\tilde{x}_i) \right| \right] \leq \sqrt{\frac{2}{n}} \sqrt{\sum_{j=1}^n \min\{\delta^2, \mu_j T^2\}}, \tag{30}$$

*where $\omega_1, \cdots, \omega_n$ are i.i.d. $\mathcal{N}(0, 1)$ variates. We apply this result to $K = K_X$. By Definition 2, all elements on the diagonal of $\boldsymbol{K}$ are at most $\kappa^2/n$, so $\sum_j \mu_j = \mathrm{Tr}(\boldsymbol{K}) \leq \kappa^2$. Thus, we have*

$$\mathcal{G}_n(\delta; \mathcal{F}_0) \leq \sqrt{\frac{8 \kappa^2 B^2}{n(1-\epsilon)}} \qquad \text{for any } \mathcal{H}_{\hat{\Psi}}. \tag{31}$$

Regarding $\bar{\mathfrak{R}}_n(\delta; \mathcal{F}_*)$, $\mathcal{F}_*$ is also a subset of RKHS $\hat{\mathcal{H}}_*$, which is the linear span of $\hat{\Psi}$ and $f^*$, and is a subspace of $\mathcal{H}_\Gamma$ whose rank is at most $(d+1)$. By Lemma 9, the sum of eigenvalues of $\hat{\mathcal{H}}_*$ is at most $S_\lambda(d+1)$. Since $\|f^*\|_{\mathcal{H}_\Gamma} \leq \frac{B}{\sqrt{1-\epsilon}}$, all $f \in \mathcal{F}_*$ satisfy $\|f\|_{\mathcal{H}_\Gamma} \leq \frac{2B}{\sqrt{1-\epsilon}}$. So we have the following bound for $\bar{\mathfrak{R}}_n(\delta; \mathcal{F}_*)$:

**Lemma 11.** *(Application of Wainwright (2019, Corollary 14.5)) Let $\mu_1, \mu_2, \cdots$ be the eigenvalues of the RKHS $\hat{\mathcal{H}}_*$. Since $\mathrm{rank}(\hat{\mathcal{H}}_*) \leq \mathrm{rank}(\mathcal{H}_{\hat{\Psi}}) + 1$, we have*

$$\bar{\mathfrak{R}}_n(\delta; \mathcal{F}_*) \leq \sqrt{\frac{2}{n}} \sqrt{\sum_{j=1}^\infty \min\left\{\delta^2, \frac{4\mu_j B^2}{1-\epsilon}\right\}} \leq \sqrt{\frac{8B^2}{n(1-\epsilon)} S_\lambda(d+1)} \quad \text{if } \mathrm{rank}(\mathcal{H}_{\hat{\Psi}}) \leq d, \tag{32}$$

*and for an arbitrary $\mathcal{H}_{\hat{\Psi}}$, we can simply replace $S_\lambda(d+1)$ with $S_\lambda$.*

### E.2 PROOFS

**Lemma 2.** *Suppose $\nu_1, \cdots, \nu_n$ are i.i.d. $\mathcal{N}(0, \sigma^2)$ variates. If $\hat{\Phi}$ has $d$ dimensions ($d$ can be $\infty$), then we have the following uniform bound over all $f^* = \Gamma^* g^* \in \mathcal{F}_B(\Gamma; \epsilon)$:*

$$\underset{\tilde{x}_i, \nu_i}{\mathbb{P}} \left[ \forall f^* \in \mathcal{F}_B(\Gamma; \epsilon), \|\hat{f} - f^*\|_{P_\mathcal{X}}^2 \leq 9\|f_{\hat{\Psi}} - f^*\|_{P_\mathcal{X}}^2 + \frac{c_0 \kappa (B^2 + \sigma B)}{1 - \epsilon} \sqrt{\frac{S_\lambda(d+1)}{n}} \right]$$

$$\geq 1 - c_1 \exp\left( -\frac{c_2 \sqrt{2n S_\lambda(d+1)}}{\kappa} \right) - \exp\left( -\sqrt{\frac{2n\kappa^2 B^2}{1 - \epsilon}} \right),$$

*where $f_{\hat{\Psi}} = \Gamma^*(\Pi_{\hat{\Phi}} g^*)$ is the projection of $f^*$ onto $\mathcal{H}_{\hat{\Psi}}$ w.r.t. $\langle \cdot, \cdot \rangle_{\mathcal{H}_\Gamma}$, and $c_0, c_1, c_2$ are universal constants. Moreover, $S_\lambda(d+1) \leq \min\left\{d+1, \kappa^2\right\}$.*

*Proof.* By Proposition 7, all functions in $\mathcal{F}_*$ are $b$-uniformly bounded, with $b = \frac{2\kappa B}{\sqrt{1-\epsilon}}$. And obviously $\mathcal{F}_*$ is star-shaped, meaning that for all $f \in \mathcal{F}_*$ and all $\beta \in [0, 1]$, $\beta f \in \mathcal{F}_*$. Let $t^2 = b \cdot \sqrt{\frac{8B^2}{n(1-\epsilon)} S_\lambda(d+1)} \geq b\bar{\mathfrak{R}}_n(\delta; \mathcal{F}_*)$. Then, by Wainwright (2019, Theorem 14.1), we have

$$\mathbb{P} \left[ \left| \|f\|_n^2 - \|f\|_{P_\mathcal{X}}^2 \right| \geq \frac{1}{2}\|f\|_{P_\mathcal{X}}^2 + \frac{t^2}{2} \right] \leq c_1 \exp\left( -c_2 \frac{nt^2}{b^2} \right) \qquad \text{for all } f \in \mathcal{F}_* \tag{33}$$

for universal constant $c_1, c_2$. We know that $\hat{f} - f^* \in \mathcal{F}_*$ and $f_{\hat{\Psi}} - f^* \in \mathcal{F}_*$, which means that

$$\mathbb{P} \left[ \left( \|\hat{f} - f^*\|_{P_\mathcal{X}}^2 \geq 2\|\hat{f} - f^*\|_n^2 + t^2 \right) \vee \left( \|f_{\hat{\Psi}} - f^*\|_n^2 \geq \frac{3}{2}\|f_{\hat{\Psi}} - f^*\|_{P_\mathcal{X}}^2 + \frac{t^2}{2} \right) \right]$$
$$\leq c_1 \exp\left( -c_2 \frac{nt^2}{b^2} \right). \tag{34}$$

Let $\delta_n^2 = 2\sigma\sqrt{\frac{8\kappa^2 B^2}{n(1-\epsilon)}}$. By Lemma 10, we have $\delta_n^2 \geq 2\sigma\mathcal{G}_n(\delta_n; \mathcal{F}_0)$. And $\mathcal{F}_0$ is also star-shaped. Thus, by setting $\gamma = 1/2$ in Wainwright (2019, Theorem 13.13), we have[†]

$$\mathbb{P} \left[ \|\hat{f} - f^*\|_n^2 \geq 3\|f_{\hat{\Psi}} - f^*\|_n^2 + 32\delta_n^2 \right] \leq \exp\left( -\frac{n\delta_n^2}{2\sigma^2} \right). \tag{35}$$

Combining the two inequalities above with the union bound, we obtain the result. □

Now we prove Lemma 3. Without loss of generality, suppose $h_1, \cdots, h_{d'}$ are linearly independent. Let $\hat{\mathcal{H}}_{d'} := \text{span}\{h_1, \cdots, h_{d'}\}$. Let $g^* = g_0 + \beta g_1$, where $g_0 = \Pi_{\hat{\mathcal{H}}_{d'}} g^*$, $g_1 \perp g_0$, and $\|g_1\|_{P_\mathcal{A}} = 1$. So by Lemma 9, we have:

**Proposition 12.** $\|\Gamma^*(G_h^{-1/2} h_1)\|_{P_\mathcal{X}}^2 + \cdots + \|\Gamma^*(G_h^{-1/2} h_{d'})\|_{P_\mathcal{X}}^2 + \|\Gamma^* g_1\|_{P_\mathcal{X}}^2 \leq \lambda_1 + \cdots + \lambda_{d'+1}$.

*Proof.* Let $\left[ G_h^{-1/2} h_1, \cdots, G_h^{-1/2} h_{d'}, g_1 \right] = Q\Phi^*$, where $Q$ is a matrix with $(d'+1)$ orthonormal rows. Then, $\left[ \Gamma^*(G_h^{-1/2} h_1), \cdots, \Gamma^*(G_h^{-1/2} h_{d'}), \Gamma^* g_1 \right] = QD_\lambda^{1/2}\Phi^*$. Thus, we have

$$\|\Gamma^*(G_h^{-1/2} h_1)\|_{P_\mathcal{X}}^2 + \cdots + \|\Gamma^*(G_h^{-1/2} h_{d'})\|_{P_\mathcal{X}}^2 + \|\Gamma^* g_1\|_{P_\mathcal{X}}^2 = \text{Tr}(QD_\lambda Q^\top).$$

Then, applying Lemma 9 completes the proof. □

*Remark.* This proposition is the functional version of Fan (1949, Theorem 1).

Notice that $\|\Gamma^*(G_h^{-1/2} h_1)\|_{P_\mathcal{X}}^2 + \cdots + \|\Gamma^*(G_h^{-1/2} h_{d'})\|_{P_\mathcal{X}}^2 = \text{Tr}(G_h^{-1/2} F_h G_h^{-1/2}) = \text{Tr}(G_h^{-1} F_h)$. With this, we can prove Lemma 3:

---

[†]Please refer to the proof of Wainwright (2019, Theorem 13.13) for removing the universal constants in this theorem.

**Lemma 3.** *For any $f^* \in \mathcal{F}_B(\Gamma; \epsilon)$, there is*

$$\|f_{\hat{\Psi}} - f^*\|_{P_{\mathcal{X}}}^2 \leq \frac{\tau^2}{1 - \tau^2} \frac{\tau + \epsilon}{1 - \epsilon} B^2.$$

*Proof.* Let $\alpha^2 = \|g_0\|_{P_{\mathcal{A}}}^2$, and $\beta^2 = \|g_0 - g^*\|_{P_{\mathcal{A}}}^2$. By Corollary 8, $\alpha^2 + \beta^2 \leq \frac{B^2}{1-\epsilon}$. Eqn. (6) implies that

$$(1 - \epsilon)(\alpha^2 + \beta^2) \leq \|\Gamma^*(g_0 + \beta g_1)\|_{P_{\mathcal{X}}}^2 \leq \alpha^2 + \beta^2 \tau^2 + 2\alpha\beta\tau,$$

since $\|\Gamma^* g_0\|_{P_{\mathcal{X}}}^2 \leq \|g_0\|_{P_{\mathcal{A}}}^2 = \alpha^2$, and $\|\Gamma^* g_1\|_{P_{\mathcal{X}}}^2 \leq \tau^2$ by Proposition 12. Thus,

$$(1 - \tau^2)\beta^2 \leq \epsilon(\alpha^2 + \beta^2) + 2\alpha\beta\tau \leq (\epsilon + \tau)(\alpha^2 + \beta^2) \leq (\epsilon + \tau)\frac{B^2}{1 - \epsilon}.$$

Thus, we have $\|f_{\hat{\Psi}} - f^*\|_{P_{\mathcal{X}}}^2 = \|\Gamma^*(g_0 - g^*)\|_{P_{\mathcal{X}}}^2 = \beta^2 \|\Gamma^* g_1\|_{P_{\mathcal{X}}}^2 \leq \beta^2 \tau^2$, which leads to the inequality we need to prove. Finally, by setting $h_i = \hat{\phi}_i$, we can see that $\tau^2 \leq S_\lambda(d+1) - \text{Tr}(\boldsymbol{G}^{-1}\boldsymbol{F})$. And for all $d' \leq d$, $\text{Tr}(\boldsymbol{G}_h^{-1}\boldsymbol{F}_h) \leq S_\lambda(d')$, so $\tau^2 \geq \lambda_{d+1}$. $\qquad\square$

## F   PROOFS FOR SECTION 4

**Proposition 4.** *For any $\hat{\Psi} = [\hat{\psi}_1, \cdots, \hat{\psi}_d]$ where $\hat{\psi}_i \in L^2(P_{\mathcal{X}})$, it holds that*

$$\text{err}(\hat{\Psi}; \mathcal{F}_B(\Gamma; \epsilon)) \geq \frac{\lambda_{d+1}}{1 - \lambda_{d+1}} \frac{\epsilon}{1 - \epsilon} B^2 \quad \text{given that} \quad \frac{\lambda_{d+1}}{1 - \lambda_{d+1}} \frac{\epsilon}{1 - \epsilon} \leq \frac{1}{2}. \qquad (36)$$

*To attain equality, it is sufficient for $\hat{\Psi}$ to span the top-d eigenspace, and also necessary if $\lambda_{d+1} < \lambda_d$.*

*Proof. Necessity:* Since $\hat{\Psi}$ is at most rank-$d$, there must be a function in $\text{span}\{\psi_1, \cdots, \psi_{d+1}\}$ that is orthogonal to $\hat{\Psi}$. Thus, we can find two functions $f_1, f_2 \in \text{span}\{\psi_1, \cdots, \psi_{d+1}\}$ such that: $\|f_1\|_{P_{\mathcal{X}}} = \|f_2\|_{P_{\mathcal{X}}} = 1$, $f_1$ is orthogonal to $\hat{\Psi}$, $f_2 = \boldsymbol{u}^\top \hat{\Psi}$ (which means that $f_2 \perp f_1$), and $\psi_1 \in \text{span}\{f_1, f_2\}$. Recall that $\lambda_1 = 1$, and $\psi_1 \equiv 1$. Let $\psi_1 = \alpha_1 f_1 + \alpha_2 f_2$, then $\alpha_1^2 + \alpha_2^2 = 1$. Without loss of generality, suppose $\alpha_1, \alpha_2 \in [0, 1]$. Let $f_0 = \alpha_2 f_1 - \alpha_1 f_2$. Then, $\|f_0\|_{P_{\mathcal{X}}} = 1$, $f_0 \perp \psi_1$. Note that we also have $\langle \psi_1, f_0 \rangle_{\mathcal{H}_\Gamma} = 0$ by duality. Let $\beta_1, \beta_2 \in [0, 1]$ be any value such that $f = \beta_1 \psi_1 + \beta_2 f_0$ satisfies $\|f\|_{P_{\mathcal{X}}}^2 = \beta_1^2 + \beta_2^2 = 1$, and $\|f\|_{\mathcal{H}_\Gamma}^2 \leq \frac{1}{1-\epsilon}$. This is satisfied as long as $\beta_2^2 \leq \frac{\epsilon}{1-\epsilon} \frac{\lambda_{d+1}}{1-\lambda_{d+1}}$, because $\|f\|_{\mathcal{H}_\Gamma}^2 \leq \beta_1^2 + \frac{\beta_2^2}{\lambda_{d+1}} = 1 + \frac{1-\lambda_{d+1}}{\lambda_{d+1}} \beta_2^2 \leq \frac{1}{1-\epsilon}$. Moreover, we have $Bf \in \mathcal{F}_B(\Gamma; \epsilon)$.

It is easy to show that $F(\alpha_1) = \alpha_1 \beta_1 + \alpha_2 \beta_2 = \alpha_1 \beta_1 + \sqrt{1 - \alpha_1^2} \beta_2$ ($\alpha_1 \in [0, 1]$) first increases then decreases, so $F(\alpha_1)^2 \geq \min\{F(0)^2, F(1)^2\} = \min\{\beta_1^2, \beta_2^2\}$, which can be $\frac{\epsilon}{1-\epsilon} \frac{\lambda_{d+1}}{1-\lambda_{d+1}}$ in the worst case given that it is at most $\frac{1}{2}$, in which case the prediction error of $Bf$ is $\|B(\alpha_1\beta_1 + \alpha_2\beta_2)f_1\|_{P_{\mathcal{X}}}^2 = F(\alpha_1)^2 B^2 = \frac{\epsilon}{1-\epsilon} \frac{\lambda_{d+1}}{1-\lambda_{d+1}} B^2$. Thus, for any $\hat{\Psi}$, we can find a function $Bf \in \mathcal{F}_B(\Gamma; \epsilon)$ such that $\min_w \text{err}(w^\top \hat{\Psi}, Bf) \geq \frac{\epsilon}{1-\epsilon} \frac{\lambda_{d+1}}{1-\lambda_{d+1}} B^2$.

When $\lambda_d > \lambda_{d+1}$, to attain equality, we need $\alpha_1 = 0$, and $\|f\|_{\mathcal{H}_\Gamma}^2 = \beta_1^2 + \frac{\beta_2^2}{\lambda_{d+1}}$, which means that $f_0 = \psi_{d+1}$. Thus, only $f_1 = f_0 = \psi_{d+1}$ is orthogonal to $\hat{\Psi}$, so $\hat{\Psi}$ must span the top-$d$ eigenspace.

*Sufficiency:* Suppose $\hat{\Psi}$ spans the top-$d$ eigenspace. For any $f \in \mathcal{F}_B(\Gamma; \epsilon)$ such that $f = \sum_i u_i \psi_i$, we have $\sum_i u_i^2 \leq B^2$, and $\sum_i \frac{1-\epsilon-\lambda_i}{\lambda_i} u_i^2 \leq 0$. Let $a = \sum_{i \geq d+1} u_i^2$ and $b = \sum_{i=1}^d u_i^2$. Then, $a = \min_w \text{err}(w^\top \hat{\Psi}, f)$, and $a + b \leq B^2$. So we have

$$0 \geq \sum_i \frac{1 - \epsilon - \lambda_i}{\lambda_i} u_i^2 \geq -\epsilon b + \frac{1 - \epsilon - \lambda_{d+1}}{\lambda_{d+1}} a \quad \left(\text{since } \frac{1 - \epsilon - \lambda}{\lambda} \text{ decreases with } \lambda\right)$$

$$\geq -\epsilon(B^2 - a) + \frac{1 - \epsilon - \lambda_{d+1}}{\lambda_{d+1}} a$$

$$= -\epsilon B^2 + (1 - \epsilon)\frac{1 - \lambda_{d+1}}{\lambda_{d+1}} a,$$

which combined with the necessity part implies that $\text{err}(\hat{\Psi}; \mathcal{F}_B(\Gamma; \epsilon)) = \frac{\epsilon}{1-\epsilon} \frac{\lambda_{d+1}}{1-\lambda_{d+1}} B^2$. $\qquad\square$

**Lemma 5.** *Suppose there exists a constant $C > 0$ such that $\mathbb{E}_{P_{\mathcal{A}}}[g^4] \leq C^2 \|g\|_{P_{\mathcal{A}}}^2$, for all $g = w^\top \hat{\Phi}$ where $\|g\|_{P_{\mathcal{A}}} \leq 1$. Then, for any $\delta > 0$, it holds with probability at least $1 - \delta$ that*

$$| \operatorname{Tr}(\hat{\boldsymbol{G}}^{-1}\hat{\boldsymbol{F}}) - \operatorname{Tr}(\boldsymbol{G}^{-1}\boldsymbol{F})| \leq \left( 2 + \sqrt{2 \log \frac{2}{\delta}} \right) \frac{C\kappa + \kappa^2}{\sqrt{N}} d.$$

*Proof.* Since multiplying an invertible $d \times d$ matrix to $\hat{\Phi}$ does not change either $\operatorname{Tr}(\hat{\boldsymbol{G}}^{-1}\hat{\boldsymbol{F}})$ or $\operatorname{Tr}(\boldsymbol{G}^{-1}\boldsymbol{F})$, for simplicity let us multiply $\boldsymbol{G}^{-1/2}$ to $\hat{\Phi}$, so that $\langle \hat{\phi}_i, \hat{\phi}_j \rangle_{P_{\mathcal{A}}} = \delta_{i,j}$ for all $i, j \in [d]$ (*i.e.* $\boldsymbol{G} = \boldsymbol{I}$). Define $\mathcal{F}_1 = \{f \in \mathcal{H}_\Gamma \mid \|f\|_{\mathcal{H}_\Gamma} \leq 1\}$. Its Rademacher complexity is given by

$$\mathfrak{R}_N(\mathcal{F}_1) = \mathbb{E}_{x_1, \cdots, x_N} \mathbb{E}_{\sigma_1, \cdots, \sigma_N} \left[ \sup_{f \in \mathcal{F}_1} \frac{1}{N} \sum_{k=1}^{N} \sigma_k f(x_k) \right]. \tag{37}$$

By Mohri et al. (2018, Theorem 6.12), we have $\mathfrak{R}_N(\mathcal{F}_1) \leq \kappa N^{-1/2}$. Moreover, by Proposition 7, all $f \in \mathcal{F}_1$ satisfy $|f(x)| \leq \kappa$ for all $x$. Thus, by Wainwright (2019, Theorem 4.10), for any $\delta > 0$, with probability at least $1 - \delta/2$, it holds for all $f \in \mathcal{F}_1$ that

$$\left| \frac{1}{N} \sum_{k=1}^{N} f(x_k) - \mathbb{E}[f(X)] \right| \leq 2\mathfrak{R}_N(\mathcal{F}_1) + \kappa \sqrt{\frac{2}{N} \log \frac{2}{\delta}} \leq \left( 2 + \sqrt{2 \log \frac{2}{\delta}} \right) \frac{\kappa}{\sqrt{N}}. \tag{38}$$

Define matrix $\boldsymbol{M} = \hat{\boldsymbol{G}}^{-1/2} \hat{\boldsymbol{F}} \hat{\boldsymbol{G}}^{-1/2} = (m_{i,j})_{i,j \in [d]}$. $\|\boldsymbol{M}\|_2 \leq 1$, so $\sum_{i=1}^{d} m_{i,j}^2 \leq 1$ for all $j \in [d]$. Consider $\operatorname{Tr}((\boldsymbol{I} - \hat{\boldsymbol{G}})\boldsymbol{M})$. For any $j \in [d]$, we have

$$((\boldsymbol{I} - \hat{\boldsymbol{G}})\boldsymbol{M})(j,j) = \left\langle \hat{\phi}_j, \sum_{i=1}^{d} m_{i,j} \hat{\phi}_i \right\rangle_{\hat{P}_{\mathcal{A}}} - \left\langle \hat{\phi}_j, \sum_{i=1}^{d} m_{i,j} \hat{\phi}_i \right\rangle_{P_{\mathcal{A}}}.$$

Note that $\left\| \sum_{i=1}^{d} m_{i,j} \hat{\phi}_i \right\|_{P_{\mathcal{A}}} \leq 1$, so $\left\| \hat{\phi}_j \left( \sum_{i=1}^{d} m_{i,j} \hat{\phi}_i \right) \right\|_{P_{\mathcal{A}}}^2 \leq \sqrt{\mathbb{E}[\hat{\phi}_j^4] \mathbb{E} \left[ \left( \sum_{i=1}^{d} m_{i,j} \hat{\phi}_i \right)^4 \right]} \leq C^2$, which means that $C^{-1} \Gamma^* \left( \hat{\phi}_j \left( \sum_{i=1}^{d} m_{i,j} \hat{\phi}_i \right) \right) \in \mathcal{F}_1$. So if Eqn. (38) holds, then for all $j \in [d]$, we have

$$((\boldsymbol{I} - \hat{\boldsymbol{G}})\boldsymbol{M})(j,j) = \left| \frac{1}{N} \sum_{k=1}^{N} \Gamma^* \left( \hat{\phi}_j \left( \sum_{i=1}^{d} m_{i,j} \hat{\phi}_i \right) \right) (x_k) - \mathbb{E} \left[ \Gamma^* \left( \hat{\phi}_j \left( \sum_{i=1}^{d} m_{i,j} \hat{\phi}_i \right) \right) (X) \right] \right|$$

$$\leq \left( 2 + \sqrt{2 \log \frac{2}{\delta}} \right) \frac{C\kappa}{\sqrt{N}},$$

which implies that

$$\operatorname{Tr} \left( \hat{\boldsymbol{G}}^{-1} \hat{\boldsymbol{F}} - \hat{\boldsymbol{F}} \right) = \operatorname{Tr} \left( \hat{\boldsymbol{G}}^{-1/2} (\boldsymbol{I} - \hat{\boldsymbol{G}}) \hat{\boldsymbol{G}}^{-1/2} \hat{\boldsymbol{F}} \right) = \operatorname{Tr} \left( (\boldsymbol{I} - \hat{\boldsymbol{G}}) \boldsymbol{M} \right) \leq \left( 2 + \sqrt{2 \log \frac{2}{\delta}} \right) \frac{C\kappa d}{\sqrt{N}}.$$

Next, define $\mathcal{F}_2 = \{f_1 f_2 \mid f_1, f_2 \in \mathcal{H}_\Gamma, \|f_1\|_{\mathcal{H}_\Gamma} \leq 1, \|f_2\|_{\mathcal{H}_\Gamma} \leq 1\}$. By Proposition 13 (proved after this lemma), we have $\mathfrak{R}_N(\mathcal{F}_2) \leq \kappa^2 N^{-1/2}$. And all $f \in \mathcal{F}_2$ satisfy $|f(x)| \leq \kappa^2$ for all $x$ by Proposition 7. So with probability at least $1 - \delta/2$, we have for all $f \in \mathcal{F}_2$,

$$\left| \frac{1}{N} \sum_{k=1}^{N} f(x_k) - \mathbb{E}[f(X)] \right| \leq 2\mathfrak{R}_N(\mathcal{F}_2) + \kappa^2 \sqrt{\frac{2}{N} \log \frac{2}{\delta}} \leq \left( 2 + \sqrt{2 \log \frac{2}{\delta}} \right) \frac{\kappa^2}{\sqrt{N}}. \tag{39}$$

Note that $\|\hat{\psi}_i\|_{\mathcal{H}_\Gamma} \leq 1$. So under Eqn. (39), we have for all $i, j \in [d]$,

$$\left| \langle \hat{\psi}_i, \hat{\psi}_j \rangle_{\hat{P}_{\mathcal{X}}} - \langle \hat{\psi}_i, \hat{\psi}_j \rangle_{P_{\mathcal{X}}} \right| = \left| \frac{1}{N} \sum_{k=1}^{N} \hat{\psi}_i(x_k) \hat{\psi}_j(x_k) - \mathbb{E}[\hat{\psi}_i \hat{\psi}_j] \right| \leq \left( 2 + \sqrt{2 \log \frac{2}{\delta}} \right) \frac{\kappa^2}{\sqrt{N}},$$

which implies that $\mathrm{Tr}\left(\hat{\boldsymbol{F}} - \boldsymbol{G}^{-1}\boldsymbol{F}\right) = \mathrm{Tr}\left(\hat{\boldsymbol{F}} - \boldsymbol{F}\right) \leq \left(2 + \sqrt{2\log\frac{2}{\delta}}\right)\frac{\kappa^2 d}{\sqrt{N}}$.

Finally, applying the union bound completes the proof. $\qquad\square$

**Proposition 13.** *Let $\mathcal{F}_2 = \{f_1 f_2 \mid f_1, f_2 \in \mathcal{H}_\Gamma, \|f_1\|_{\mathcal{H}_\Gamma} \leq 1, \|f_2\|_{\mathcal{H}_\Gamma} \leq 1\}$. Then, $\mathfrak{R}_N(\mathcal{F}_2) \leq \frac{\kappa^2}{\sqrt{N}}$.*

*Proof.* For any $h(x) = f_1(x)f_2(x) \in \mathcal{F}_2$, let $f_1 = \Gamma^* g_1$ and $f_2 = \Gamma^* g_2$, where $\|g_1\|_{P_{\mathcal{A}}} \leq 1$ and $\|g_2\|_{P_{\mathcal{A}}} \leq 1$. Let $g_1 = \sum_i u_i \phi_i$ and $g_2 = \sum_i v_i \phi_i$. Let $\boldsymbol{u} = [u_1, u_2, \cdots]$ and $\boldsymbol{v} = [v_1, v_2, \cdots]$. Then, $\|\boldsymbol{u}\|_2 \leq 1$ and $\|\boldsymbol{v}\|_2 \leq 1$. And we have $f_1 = \sum_i \lambda_i^{1/2} u_i \psi_i$, and $f_2 = \sum_i \lambda_i^{1/2} v_i \psi_i$.

For any $x \in \mathcal{X}$, let $\Psi(x) = [\lambda_1^{1/2}\psi_1(x), \lambda_2^{1/2}\psi_2(x), \cdots]$. Then, $f_1(x) = \boldsymbol{u}^\top \Psi(x)$ and $f_2(x) = \boldsymbol{v}^\top \Psi(x)$. Denote $\Psi_k = \Psi(x_k)$. Then, $\Psi_k^\top \Psi_k \leq \kappa^2$ for all $k \in [N]$. So for any $S = \{x_1, \cdots, x_N\}$, the empirical Rademacher complexity satisfies

$$
\hat{\mathfrak{R}}_S(\mathcal{F}_2) \leq \mathbb{E}_{\boldsymbol{\sigma}}\left[\sup_{\|\boldsymbol{u}\|_2 \leq 1, \|\boldsymbol{v}\|_2 \leq 1}\left|\frac{1}{N}\sum_{k=1}^N \sigma_k \boldsymbol{u}^\top \Psi_k \Psi_k^\top \boldsymbol{v}\right|\right]
$$

$$
\leq \frac{1}{N}\mathbb{E}_{\boldsymbol{\sigma}}\left[\left\|\sum_{k=1}^N \sigma_k \Psi_k \Psi_k^\top\right\|_2\right]
$$

$$
\leq \frac{1}{N}\mathbb{E}_{\boldsymbol{\sigma}}\left[\left\|\sum_{k=1}^N \sigma_k \Psi_k \Psi_k^\top\right\|_F\right]
$$

$$
= \frac{1}{N}\mathbb{E}_{\boldsymbol{\sigma}}\left[\mathrm{Tr}\left(\left(\sum_{k=1}^N \sigma_k \Psi_k \Psi_k^\top\right)^\top \left(\sum_{l=1}^N \sigma_l \Psi_l \Psi_l^\top\right)\right)^{1/2}\right]
$$

$$
\leq \frac{1}{N}\sqrt{\mathbb{E}_{\boldsymbol{\sigma}}\left[\mathrm{Tr}\left(\sum_{k,l=1}^N \sigma_k \sigma_l \Psi_k \Psi_k^\top \Psi_l \Psi_l^\top\right)\right]} \qquad \text{(Jensen)}
$$

$$
= \frac{1}{N}\sqrt{\mathrm{Tr}\left(\sum_{k,l=1}^N \mathbb{E}[\sigma_k \sigma_l]\Psi_k \Psi_k^\top \Psi_l \Psi_l^\top\right)}
$$

$$
= \frac{1}{N}\sqrt{\mathrm{Tr}\left(\sum_{k=1}^N \Psi_k \Psi_k^\top \Psi_k \Psi_k^\top\right)}
$$

$$
\leq \frac{1}{N}\sqrt{N\kappa^4} = \frac{\kappa^2}{\sqrt{N}}.
$$

Then, since $\mathfrak{R}_N(\mathcal{F}_2) = \mathbb{E}_S[\hat{\mathfrak{R}}_S(\mathcal{F}_2)]$, we obtain the result. $\qquad\square$

**Lemma 6.** *Suppose $\hat{\bar{\phi}}_i = \bar{\phi}_i$ for $i \in [d]$. Let $\gamma_{\boldsymbol{G}} := \lambda_{\max}(\boldsymbol{G})/\lambda_{\min}(\boldsymbol{G})$, which is the condition number of $\boldsymbol{G}$. Then, for any $\delta > 0$, both*

$$
\sum_{j=1}^d \bar{\lambda}_j \geq \sum_{i=1}^d \lambda_i - \left(2 + \sqrt{2\log\frac{2}{\delta}}\right)\frac{(\lambda_d^{-1}+1)\kappa^2}{\sqrt{N}}d
$$

*and Eqn. (12) with $C = \kappa \bar{\lambda}_d^{-1}\gamma_{\boldsymbol{G}}^{1/2}$ hold simultaneously for $\mathcal{H}_{\hat{\Psi}} = \hat{\mathcal{H}}_d$ with probability at least $1 - \delta$.*

*Proof.* Denote $\boldsymbol{\Phi}_d^* = [\phi_1, \cdots, \phi_d]$ and $\bar{\boldsymbol{\Phi}}_d^* = [\bar{\phi}_1, \cdots, \bar{\phi}_d]$. Let $\bar{\boldsymbol{\Phi}}_d^* = \boldsymbol{P}\boldsymbol{\Phi}^*$, where $\boldsymbol{P}$ is a matrix with $d$ rows. Observe that for any $g = \sum_i u_i \bar{\phi}_i$ such that $\|g\|_{P_{\mathcal{A}}} \leq 1$, we have $g = \bar{\Gamma}\Gamma^*\left(\sum_i \bar{\lambda}_i^{-1}u_i\bar{\phi}_i\right)$. Let $\boldsymbol{u} = (u_1, \cdots, u_d)$, then there is $g = \boldsymbol{u}^\top \bar{\boldsymbol{\Phi}}_d^* = \boldsymbol{u}^\top \boldsymbol{P}\boldsymbol{\Phi}^*$, so $\|\boldsymbol{P}^\top \boldsymbol{u}\|_2 \leq 1$. Thus, we have $\|\sum_i \bar{\lambda}_i^{-1}u_i\bar{\phi}_i\|_{P_{\mathcal{A}}} = \|\boldsymbol{P}^\top \boldsymbol{D}_{\bar{\lambda}^d}^{-1}\boldsymbol{u}\|_2 = \|\boldsymbol{P}^\top \boldsymbol{D}_{\bar{\lambda}^d}^{-1}(\boldsymbol{P}\boldsymbol{P}^\top)^{-1}\boldsymbol{P}\boldsymbol{P}^\top \boldsymbol{u}\|_2$.

So we just need to show that $\|\boldsymbol{P}^\top \boldsymbol{D}_{\bar{\lambda}^d}^{-1}(\boldsymbol{P}\boldsymbol{P}^\top)^{-1}\boldsymbol{P}\|_2 \leq \bar{\lambda}_d^{-1}\gamma_{\boldsymbol{G}}^{1/2}$. $\|\boldsymbol{P}^\top \boldsymbol{D}_{\bar{\lambda}^d}^{-1}(\boldsymbol{P}\boldsymbol{P}^\top)^{-1}\boldsymbol{P}\|_2$ is equal to the square root of the largest eigenvalue of $\boldsymbol{P}^\top \boldsymbol{D}_{\bar{\lambda}^d}^{-1}(\boldsymbol{P}\boldsymbol{P}^\top)^{-1}\boldsymbol{D}_{\bar{\lambda}^d}^{-1}\boldsymbol{P}$, and by using two simple linear algebra exercises: (i) $\lambda_{\max}(\boldsymbol{A}\boldsymbol{B}) \leq \lambda_{\max}(\boldsymbol{A})\lambda_{\max}(\boldsymbol{B})$ for positive definite matrices $\boldsymbol{A}$ and $\boldsymbol{B}$, and (ii) $\boldsymbol{A}\boldsymbol{B}$ and $\boldsymbol{B}\boldsymbol{A}$ share the same non-zero eigenvalues (Sylvester's Theorem), and the fact that $\boldsymbol{G} = \boldsymbol{P}\boldsymbol{P}^\top$, we can show that the largest eigenvalue of this matrix is at most $\bar{\lambda}_d^{-2}\gamma_{\boldsymbol{G}}$.

Therefore, we have $\|\boldsymbol{P}^\top \boldsymbol{D}_{\bar{\lambda}^d}^{-1}(\boldsymbol{P}\boldsymbol{P}^\top)^{-1}\boldsymbol{P}\|_2 \leq \bar{\lambda}_d^{-1}\gamma_{\boldsymbol{G}}^{1/2}$, which combined with $\|\boldsymbol{P}^\top\boldsymbol{u}\|_2 \leq 1$ implies that $\|\sum_i \bar{\lambda}_i^{-1}u_i\bar{\phi}_i\|_{P_{\mathcal{A}}} \leq \bar{\lambda}_d^{-1}\gamma_{\boldsymbol{G}}^{1/2}$. By Proposition 7, $|\Gamma^*\left(\sum_i \bar{\lambda}_i^{-1}u_i\bar{\phi}_i\right)(x)| \leq \kappa\bar{\lambda}_d^{-1}\gamma_{\boldsymbol{G}}^{1/2}$ for all $x$, so we have $|\bar{\Gamma}\Gamma^*\left(\sum_i \bar{\lambda}_i^{-1}u_i\bar{\phi}_i\right)(a)| = |\int \Gamma^*\left(\sum_i \bar{\lambda}_i^{-1}u_i\bar{\phi}_i\right)(x)p(x|a)dx| \leq \kappa\bar{\lambda}_d^{-1}\gamma_{\boldsymbol{G}}^{1/2}$ for all $a$. This means that with $C = \kappa\bar{\lambda}_d^{-1}\gamma_{\boldsymbol{G}}^{1/2}$, $g$ satisfies the condition of Lemma 5. Therefore, with probability at least $1 - \delta$, both Eqn. (38) and Eqn. (39) hold and they lead to Eqn. (12).

Now let $\boldsymbol{\Phi}_d^* = \boldsymbol{Q}\bar{\boldsymbol{\Phi}}^*$, where $\boldsymbol{Q}$ is a matrix with $d$ rows. Consider two matrices $\boldsymbol{Q}\boldsymbol{Q}^\top, \boldsymbol{Q}\boldsymbol{D}_{\bar{\lambda}}\boldsymbol{Q}^\top \in \mathbb{R}^{d\times d}$ where $\boldsymbol{D}_{\bar{\lambda}} = \mathrm{diag}(\bar{\lambda}_1, \bar{\lambda}_2, \cdots)$, for which we have

$$(\boldsymbol{Q}\boldsymbol{Q}^\top)(i,j) = \langle\phi_i, \phi_j\rangle_{\hat{P}_{\mathcal{A}}} \qquad \text{and} \qquad (\boldsymbol{Q}\boldsymbol{D}_{\bar{\lambda}}\boldsymbol{Q}^\top)(i,j) = \langle\Gamma^*\phi_i, \Gamma^*\phi_j\rangle_{\hat{P}_{\mathcal{X}}}.$$

We have $(\langle\phi_i, \phi_j\rangle_{P_{\mathcal{A}}})_{i,j\in[d]} = \boldsymbol{I}$ and $(\langle\Gamma^*\phi_i, \Gamma^*\phi_j\rangle_{P_{\mathcal{X}}})_{i,j\in[d]} = \boldsymbol{D}_{\lambda^d} := \mathrm{diag}(\lambda_1, \cdots, \lambda_d)$. Moreover, for any $g = \boldsymbol{u}^\top\boldsymbol{\Phi}_d^*$ such that $\|g\|_{P_{\mathcal{A}}} \leq 1$, there is $g = \Gamma\Gamma^*\left(\sum_i \lambda_i^{-1}u_i\phi_i\right)$, and obviously $\|\sum_i \lambda_i^{-1}u_i\phi_i\|_{P_{\mathcal{A}}} \leq \lambda_d^{-1}$. Thus, we can show that for all $a$, $|g(a)| \leq \kappa\lambda_d^{-1}$, which means that $\boldsymbol{\Phi}_d^*$ satisfies the fourth-moment control assumption in Lemma 5 with $C' = \kappa\lambda_d^{-1}$. So similar to the proof of Lemma 5, for all $\boldsymbol{u} \in \mathbb{R}^d$ such that $\|\boldsymbol{u}\|_2 \leq 1$, we can show that

$$\left|\boldsymbol{u}^\top(\boldsymbol{Q}\boldsymbol{Q}^\top - \boldsymbol{I})\boldsymbol{u}\right| = \left|\langle\boldsymbol{u}^\top\boldsymbol{\Phi}_d^*, \boldsymbol{u}^\top\boldsymbol{\Phi}_d^*\rangle_{\hat{P}_{\mathcal{A}}} - \langle\boldsymbol{u}^\top\boldsymbol{\Phi}_d^*, \boldsymbol{u}^\top\boldsymbol{\Phi}_d^*\rangle_{P_{\mathcal{A}}}\right| \leq \left(2 + \sqrt{2\log\frac{2}{\delta}}\right)\frac{\kappa^2\lambda_d^{-1}}{\sqrt{N}},$$

which implies that $\|\boldsymbol{Q}\boldsymbol{Q}^\top\|_2 \leq 1 + \left(2 + \sqrt{2\log\frac{2}{\delta}}\right)\frac{\kappa^2\lambda_d^{-1}}{\sqrt{N}}$. It is easy to show that all non-zero eigenvalues of $\boldsymbol{Q}^\top\boldsymbol{Q}$ are also eigenvalues of $\boldsymbol{Q}\boldsymbol{Q}^\top$, so $\|\boldsymbol{Q}^\top\boldsymbol{Q}\|_2 \leq 1 + \left(2 + \sqrt{2\log\frac{2}{\delta}}\right)\frac{\kappa^2\lambda_d^{-1}}{\sqrt{N}}$. Moreover, similar to the proof of Lemma 5, we can show that for all $i, j \in [d]$,

$$
\begin{cases}
\left|(\boldsymbol{Q}\boldsymbol{Q}^\top - \boldsymbol{I})(i,j)\right| \leq \left(2 + \sqrt{2\log\frac{2}{\delta}}\right)\frac{\kappa^2\lambda_d^{-1}}{\sqrt{N}}; & (40) \\[2ex]
\left|(\boldsymbol{Q}\boldsymbol{D}_{\bar{\lambda}}\boldsymbol{Q}^\top - \boldsymbol{D}_{\lambda^d})(i,j)\right| \leq \left(2 + \sqrt{2\log\frac{2}{\delta}}\right)\frac{\kappa^2}{\sqrt{N}}. & (41)
\end{cases}
$$

Let $\boldsymbol{q}_i$ be the $i$-th column of $\boldsymbol{Q}$. Then for all $i \in [d]$, $\boldsymbol{q}_i^\top\boldsymbol{q}_i \leq 1 + \left(2 + \sqrt{2\log\frac{2}{\delta}}\right)\frac{\kappa^2\lambda_d^{-1}}{\sqrt{N}}$. And we also have $\sum_{i=1}^\infty \boldsymbol{q}_i^\top\boldsymbol{q}_i = \mathrm{Tr}(\boldsymbol{Q}^\top\boldsymbol{Q}) = \mathrm{Tr}(\boldsymbol{Q}\boldsymbol{Q}^\top) \leq d + \left(2 + \sqrt{2\log\frac{2}{\delta}}\right)\frac{\kappa^2\lambda_d^{-1}d}{\sqrt{N}}$. Thus, we have

$$
\begin{aligned}
\sum_{i=1}^d \lambda_i - \left(2 + \sqrt{2\log\frac{2}{\delta}}\right)\frac{\kappa^2}{\sqrt{N}}d &\leq \mathrm{Tr}(\boldsymbol{Q}\boldsymbol{D}_{\bar{\lambda}}\boldsymbol{Q}^\top) = \mathrm{Tr}(\boldsymbol{D}_{\bar{\lambda}}\boldsymbol{Q}^\top\boldsymbol{Q}) \\
&= \sum_{i=1}^\infty \bar{\lambda}_i\boldsymbol{q}_i^\top\boldsymbol{q}_i = \sum_{j=1}^\infty\left(\sum_{i=1}^j \boldsymbol{q}_i^\top\boldsymbol{q}_i\right)(\bar{\lambda}_j - \bar{\lambda}_{j+1}) \\
&\leq \sum_{i=1}^d \bar{\lambda}_i\left[1 + \left(2 + \sqrt{2\log\frac{2}{\delta}}\right)\frac{\kappa^2\lambda_d^{-1}}{\sqrt{N}}\right] \leq \sum_{i=1}^d \bar{\lambda}_i + \left(2 + \sqrt{2\log\frac{2}{\delta}}\right)\frac{\kappa^2\lambda_d^{-1}d}{\sqrt{N}},
\end{aligned}
$$

which proves the assertion. $\qquad\square$

## G   PROOFS FOR SECTION 5

**Example 1.**   *Consider $\mathcal{X} = \{-1, 1\}^{d_{\mathcal{X}}}$, where each $x \in \mathcal{X}$ is a vector of $-1$ and $1$ with length $d_{\mathcal{X}}$. Let $p_{\mathcal{X}}$ be the uniform distribution over $\mathcal{X}$. Consider a random masking augmentation, where for any*

$x \in \mathcal{X}$, each coordinate $x^i$ is randomly masked to be $0$ with probability $\alpha \in (0, 1)$ independently, where $\alpha$ is the mask ratio ($0$ is the `[MASK]` token). Then, $\kappa^2 = (2 - \alpha)^{d_\mathcal{X}}$.

*Proof.* We know that $\kappa^2 \geq \int \frac{p(a|x)^2}{p(a)} da$, and by symmetry, the right-hand-side is the same for all $x$. Given an $a$, suppose $a$ has $r$ coordinates masked and $(d_\mathcal{X} - r)$ coordinates unmasked. Then, there are $2^r$ possible $x$ that can be augmented to $a$. For each of these $x$, $p(a|x) = \alpha^r (1 - \alpha)^{d_\mathcal{X} - r}$. So $p(a) = \int p(a|x)p(x)dx = 2^{r-d_\mathcal{X}} \alpha^r (1 - \alpha)^{d_\mathcal{X} - r}$. Thus, we have

$$\kappa^2 = \int \frac{p(a|x)^2}{p(a)} da = \sum_{r=0}^{d_\mathcal{X}} \binom{d_\mathcal{X}}{r} \frac{\alpha^{2r}(1 - \alpha)^{2d_\mathcal{X} - 2r}}{2^{r-d_\mathcal{X}} \alpha^r (1 - \alpha)^{d_\mathcal{X} - r}}$$

$$= \sum_{r=0}^{d_\mathcal{X}} \binom{d_\mathcal{X}}{r} \alpha^r (2 - 2\alpha)^{d_\mathcal{X} - r}$$

$$= (\alpha + 2 - 2\alpha)^{d_\mathcal{X}} = (2 - \alpha)^{d_\mathcal{X}},$$

which completes the proof. $\square$

**Example 2.** *Consider a random block masking augmentation with ratio $\alpha$, which for any $x \in \mathcal{X}$ masks $x^i, x^{i+1}, \cdots, x^{i+r-1}$ for a uniformly random $i$, and $r = \lceil \alpha d_\mathcal{X} \rceil$. Then, $\kappa^2 \leq [2^{(1-\alpha)}]^{d_\mathcal{X}}$.*

*Proof.* For any $a$, we have $p(a) = \frac{1}{d_\mathcal{X} - r + 1} \frac{1}{2^{d_\mathcal{X} - r}}$, and $p(a|x) = \frac{1}{d_\mathcal{X} - r + 1}$ if $x$ can be augmented to $a$. So there always is $\frac{p(a|x)}{p(a)} = 2^{d_\mathcal{X} - r} \leq 2^{(1-\alpha)d_\mathcal{X}}$. Thus, we have $\kappa^2 \leq 2^{(1-\alpha)d_\mathcal{X}}$. $\square$

**Example 3.** *Consider random block masking + flipping with ratio $\alpha$, where for any $x \in \mathcal{X}$, first $x^i, \cdots, x^{i+r-1}$ are masked to be $0$ for a uniformly random $i$ and $r = \lceil \frac{\alpha}{2} d_\mathcal{X} \rceil$, and then each remaining coordinate is randomly flipped sign ($1 \to -1$ and $-1 \to 1$) with probability $\frac{\alpha}{2}$ independently. Then, $\kappa^2 \leq \left[ (\alpha^2 - 2\alpha + 2)^{(1-\alpha/2)} \right]^{d_\mathcal{X}}$.*

*Proof.* For any $a$, we have $p(a) = \frac{1}{d_\mathcal{X} - r + 1} \frac{1}{2^{d_\mathcal{X} - r}}$. Suppose $a$ is augmented from $x$, and among the unmasked $(d_\mathcal{X} - r)$ coordinates, $a$ and $x$ have $k$ disagreeing coordinates. For a given $k$, there are $(d_\mathcal{X} - r + 1)\binom{d_\mathcal{X} - r}{k}$ possible $a$, and we have $p(a|x) = \frac{1}{d_\mathcal{X} - r + 1} (\frac{\alpha}{2})^k (1 - \frac{\alpha}{2})^{d_\mathcal{X} - r - k}$. Thus, we have

$$\int \frac{p(a|x)^2}{p(a)} da = \sum_{k=0}^{d_\mathcal{X} - r} (d_\mathcal{X} - r + 1) \binom{d_\mathcal{X} - r}{k} \frac{\frac{1}{(d_\mathcal{X} - r + 1)^2} (\frac{\alpha}{2})^{2k} (1 - \frac{\alpha}{2})^{2d_\mathcal{X} - 2r - 2k}}{\frac{1}{d_\mathcal{X} - r + 1} \frac{1}{2^{d_\mathcal{X} - r}}}$$

$$= \sum_{k=0}^{d_\mathcal{X} - r} \binom{d_\mathcal{X} - r}{k} 2^{d_\mathcal{X} - r} \left( \frac{\alpha^2}{4} \right)^k \left( 1 - \alpha + \frac{\alpha^2}{4} \right)^{d_\mathcal{X} - r - k}$$

$$= 2^{d_\mathcal{X} - r} \left( \frac{\alpha^2}{4} + 1 - \alpha + \frac{\alpha^2}{4} \right)^{d_\mathcal{X} - r}$$

$$\leq \left( \alpha^2 - 2\alpha + 2 \right)^{d_\mathcal{X} - r} \leq \left( \alpha^2 - 2\alpha + 2 \right)^{(1-\alpha/2)d_\mathcal{X}},$$

which completes the proof. $\square$

## H   EXPERIMENT DETAILS

This section contains the details of the experiments we conducted in Section 5.2.

### H.1   ESTIMATING THE AUGMENTATION COMPLEXITY OF MASKED LANGUAGE MODELING

We estimate the $\kappa$ of four augmentations, namely Random masking, Block masking, Random masking + Flipping, and Block masking + Flipping, on the widely used NLP dataset `wikipedia-simple`. Since it is prohibitively expensive to iterate over the entire dataset, we instead aim to estimate $\kappa$ with a subset of the dataset. In our experiments, instead of estimating $\sup_x K_X(x, x)$, we estimate its $99^{th}$ percentile. This is because of two reasons:

(i) $\sup_x K_X(x, x)$ is statistically impossible to estimate from a subset of data without any extra assumptions on the distribution of $K_X(x, x)$. This means that without iterating over the entire dataset, we cannot get a finite confidence interval. The percentile, on the other hand, can be estimated with a finite confidence interval via sampling regardless of the distribution of $K_X(x, x)$, which is a well-known result in statistics (Hahn & Meeker, 2011, Section 5.2).

(ii) Almost all real datasets contain outliers, *i.e.* samples that are very different from the other samples. These samples will have a very large $K_X(x, x)$, because their augmentations rarely overlap with the augmentations of other samples. Therefore, $\sup_x K_X(x, x)$ itself is not really meaningful because it is too sensitive to outliers, and cannot show how the augmentation works on most part of the population.

As a demonstration of (ii), the following is an example outlier in the `wikipedia-simple` dataset,[‡] and it has a very large $K_X(x, x)$:

```
Geroldsgrün is a municipality in Hof, Bavaria, Germany.

Geography

Boroughs

 Dürrenwaid
 Dürrenwaiderhammer
 Langenbachtal
 Langenau
 Langenbach
 Lotharheil
 Mühlleiten
 Geroldsgrün
 Geroldsreuth
 Großenreuth
 Hermesgrün
 Hertwegsgrün
 Hirschberglein
 Silberstein
 Steinbach
 Untersteinbach

References
```

We can see that this sample is very different from a typical sample in the dataset in that (a) It contains lots of German words while this is an English dataset; (b) It mostly consists of individual words, while most other samples are comprised of sentences and paragraphs. Such outliers are rare in the dataset, but they result in a large $K_X(x, x)$. Therefore, we instead estimate the $\beta$-th percentile of $K_X(x, x)$ for some $\beta$ close to 100. This is a more robust estimator that can be computed with a subset of data.

Specifically, we sample $m = 1000$ samples $x_1, \cdots, x_m$ from the dataset uniformly at random. The maximum length is set at $l = 64$. For each $x_i$, we estimate $\int \frac{p(x_i|a)}{p(x_i)} p(a|x_i) da$ via sampling $r = 255$ *i.i.d.* augmentations from $p(a|x_i)$ and computing the mean of $\frac{p(x_i|a_j)}{p(x_i)}$. So it suffices to estimate $p(x_i|a_j)$ and $p(x_i)$. To do this, we leverage a `bert-base-uncased`, which is a bi-directional language model. Then, we can estimate $p(x|a)$ for any $x$ and $a$ as follows: For a sample $x = [x^{(1)}, \cdots, x^{(l)}]$ ($x^{(i)}$ is a token) and its augmentation $a$, there is

$$\log p(x|a) = \log p\left(x^{(1)}\middle|a\right) + \log p\left(x^{(2)}\middle|a, x^{(1)}\right) + \cdots + \log p\left(x^{(l)}\middle|a, x^{(1)}, \cdots, x^{(l-1)}\right).$$

So we can estimate $\log p(x|a)$ in an autoregressive fashion: First estimate $\log p\left(x^{(1)}\middle|a\right)$, then replace $a^{(1)}$ with $x^{(1)}$ and estimate $\log p\left(x^{(2)}\middle|a, x^{(1)}\right)$, then replace $a^{(2)}$ with $x^{(2)}$ and estimate

---

[‡]Dataset available at huggingface.co/datasets/wikipedia. The index of the example outlier is 199562.

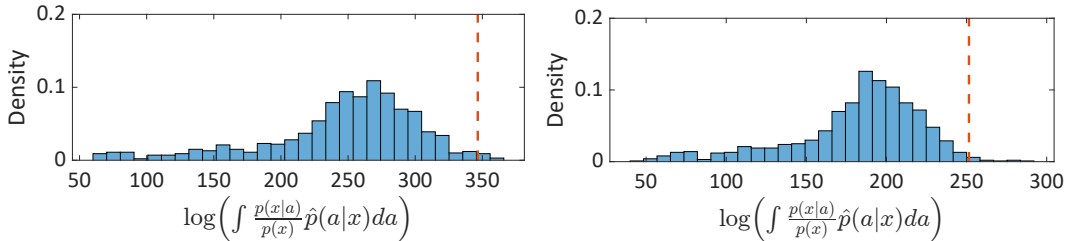

Figure 4: Plots of $\log\left(\int \frac{p(x|a)}{p(x)}\hat{p}(a|x)da\right)$ for random masking. **Left:** $\alpha = 0.15$; **Right:** $\alpha = 0.4$.

Table 1: $99^{th}$ percentile of estimated $\log K_X(x,x)$ on `wikipedia-simple`.

|  | Random Mask | Random Mask + Flip | Block Mask | Block Mask + Flip |
|---|---|---|---|---|
| $\alpha = 0.05$ | $353.94 \pm 4.63$ | $350.99 \pm 3.20$ | $346.77 \pm 4.47$ | $350.46 \pm 1.43$ |
| $\alpha = 0.10$ | $348.45 \pm 2.94$ | $349.14 \pm 1.15$ | $339.86 \pm 5.62$ | $343.73 \pm 5.76$ |
| $\alpha = 0.15$ | $346.40 \pm 1.94$ | $342.77 \pm 4.17$ | $327.13 \pm 2.33$ | $336.81 \pm 5.27$ |
| $\alpha = 0.20$ | $339.03 \pm 2.88$ | $335.01 \pm 2.65$ | $313.90 \pm 2.55$ | $322.87 \pm 3.55$ |
| $\alpha = 0.30$ | $325.08 \pm 2.35$ | $312.21 \pm 5.96$ | $280.65 \pm 1.87$ | $302.60 \pm 2.49$ |
| $\alpha = 0.40$ | $307.00 \pm 4.55$ | $280.76 \pm 0.71$ | $254.40 \pm 1.48$ | $279.24 \pm 4.59$ |
| $\alpha = 0.50$ | $278.59 \pm 2.46$ | $241.59 \pm 4.50$ | $217.35 \pm 2.81$ | $240.73 \pm 1.58$ |
| $\alpha = 0.60$ | $250.01 \pm 3.43$ | $198.15 \pm 4.01$ | $187.60 \pm 2.45$ | $205.45 \pm 2.63$ |
| $\alpha = 0.70$ | $214.14 \pm 0.88$ | $151.58 \pm 3.32$ | $152.70 \pm 1.60$ | $164.48 \pm 2.59$ |
| $\alpha = 0.80$ | $160.24 \pm 2.44$ | $84.51 \pm 2.35$ | $113.86 \pm 2.39$ | $108.44 \pm 1.43$ |

$\log p\left(x^{(3)}\big|a, x^{(1)}, x^{(2)}\right)$, and so on. The bi-directionality of BERT is important for such estimation. To compute $p(x)$, we set $a$ to be a fully masked sentence, and then compute $p(x|a) = p(x)$.

First, we decide the percentile of $K_X(x,x)$ to report. In Figure 4, we plot the histogram of $\log\left(\int \frac{p(x|a)}{p(x)}\hat{p}(a|x)da\right)$ estimated using the above method of random masking with mask ratio $\alpha = 0.15$ and $\alpha = 0.4$. The red dashed lines indicate the $99^{th}$ percentile, and we can see that they cut the tail of the data while preserving the bulk of the mass. Thus, we report the $99^{th}$ percentile for this dataset.

We study four masking augmentations: Random masking (mask $r = \lceil \alpha d_\mathcal{X}\rceil$ tokens uniformly at random), block masking (mask a block of $r$ tokens uniformly at random), random masking and flipping (mask $r/2$ tokens uniformly at random and then replace $r/2$ remaining tokens with random tokens), and block masking and flipping (mask a block of $r/2$ tokens uniformly at random and then replace $r/2$ remaining tokens with random tokens). In standard BERT, the number of randomly replaced tokens is much smaller than the number of masked tokens, but here we make them the same in order to magnify their difference.

In Table 1, we report the estimated $99^{th}$ percentile of $\log K_X(x,x)$. Each experiment is run five times with different random seeds, and we report the average and standard deviation. We can see that:

(i) $\kappa$ decreases as $\alpha$ grows, showing that a stronger augmentation has a lower complexity.

(ii) Random masking always has the highest complexity. Regarding the effect of block masking and flipping, block masking has a greater effect when $\alpha$ is small and it makes the complexity lower, whereas flipping has a greater effect when $\alpha$ is larger.

To conclude, as long as there is a way to estimate $\frac{p(x|a)}{p(x)}$ or $\frac{p(a|x)}{p(a)}$, we can estimate $\kappa$ via sampling from the dataset. Note that in our estimation, BERT is only used for estimating $p(x|a)$. $\kappa$ itself is defined free of any model.

## H.2 Analyzing the Effect of Augmentation on Downstream Performance

Our theoretical analysis implies that as long as Assumption 1 is satisfied, a smaller $\kappa$, *i.e.* a stronger augmentation, leads to better generalization. However, this does not mean that a stronger augmentation will always lead to a better test performance of tighter generalization gap, due to the following reasons:

(i) If the augmentation is too strong, then the model might have low training performance. In the extreme case, if the sentences are 100% masked, then the language model can learn nothing.

(ii) Stronger augmentations usually work better with larger models. For instance, Wettig et al. (2023) demonstrated that large models with strong augmentations can achieve higher performance than standard models and augmentations. However, for small models, stronger augmentations can lead to lower performance. One possible reason is that small models are not expressive enough, but we conjecture that this is not the true cause since a Transformer with a moderate size is already sufficiently complex. We conjecture that the real reason lies in optimization, in that it would be harder for the optimizer to find a point close to the global minima with a small model and a strong augmentation. The big model, on the other hand, has more parameters and could be easier to optimize in practice.

(iii) If the augmentation is too strong, then Assumption 1 will be violated, and our results will not hold so that the generalization gap can be large.

We study the effect of the augmentation on two real NLP downstream tasks, namely QNLI and SST-2. We study random masking with different mask ratios. Unlike the experiments in Wettig et al. (2023) that applied the pretrained encoder directly to downstream tasks, in our experiments we explicitly use the average encoder Eqn. (7), which is estimated by sampling 16 augmentations for each sample $x$ and then averaging their embeddings. We do, however, fine-tune the encoder during downstream as people do in common practice, which has been proven useful, [§] even though our theory does not analyze fine-tuning. We acknowledge that there is a discrepancy between the theory and real practice, which we aim to address in the future.

Details of our experiments:

(i) We train `roberta-large` models using the fast training recipe provided by Wettig et al. (2023). For $\alpha \geq 0.15$, we directly use the Huggingface checkpoints they provide. For $\alpha = 0.05, 0.10$, we pretrain new encoders using their source code[¶] without any modification. Note that these models use pre-layer normalization (see the original github repository regarding this detail). We use 8 NVIDIA A6000 GPUs for pretraining.

(ii) For downstream fine-tuning and evaluation, we use the official repository provided by Huggingface.[‖] The only modification we make is that we explicitly use the average encoder. The classifiers are trained for 3 epochs on QNLI and 6 epochs on SST-2. All hyperparameters are kept the same as the official repository. We use 4 NVIDIA A6000 GPUs for downstream training and evaluation.

The results are plotted in Figure 3c in Section 5.2. As discussed there, the "sweet spot" where the model achieves the highest test performance is in the middle, where the augmentation is strong enough to have good generalization, yet not too strong to break Assumption 1 or to make the training performance too low.

---

[§]We find in our experiments that without fine-tuning, the downstream classifiers cannot achieve meaningful training accuracy, merely higher than 70% for a binary classification task. With the training accuracy so low, there is no empirical generalization gap, and the test accuracy is even higher than the training accuracy in many cases, so these results are not meaningful.

[¶]https://github.com/princeton-nlp/DinkyTrain

[‖]https://github.com/huggingface/transformers/tree/main/examples/pytorch/text-classification

