# OpenReview forum: "Understanding Augmentation-based Self-Supervised Representation Learning via RKHS Approximation and Regression"
_ICLR.cc/2024/Conference — ICLR 2024 spotlight_

### Official Review · Reviewer_dUJ6 · 2023-10-16

**Soundness:** 3 good
**Presentation:** 3 good
**Contribution:** 3 good
**Rating:** 8
**Confidence:** 4

**Summary:**

This paper gives a theoretical analysis of how the distribution of augmentations affects the accuracy of contrastive learning methods, building on previous work that has studied this problem through the lens of RKHS regression, and focusing on nonparametric encoder-complexity-independent bounds. The authors first discuss the augmentation operator and show that it induces a pair of dual operators with corresponding eigenfunction decompositions. They then introduce the concepts of *augmentation complexity* (based on the infinity norm of a kernel defined by the augmentations) and the *ratio trace* (which roughly characterizes "how much" of the eigenspaces of the corresponding RKHS are captured by a given encoder).

The authors then prove a number of results based on these quantities. First, they show that the error of a downstream predictor can be decomposed into an approximation-error term (how closely the function can be recovered from the learned encoder's representation) and a downstream-estimation-error term (how well the linear readout actually recovers this function). Second, they consider approximating the top-d eigenspace of the RKHS using only a finite pretraining dataset, and show that this can be used to bound the approximation error term.

Finally, the authors consider how their "augmentation complexity" measurement translates to real augmentations, focusing on masking transformations. They give theoretical values for it for the hypercube examples discussed in Cabannes et al. (2023), and also show how to estimate it for masked language modeling on the `wikipedia-simple` dataset.

**Strengths:**

**[S1]** The paper is overall clearly written and easy to follow, and the authors do a good job introducing the problem and explaining their analysis. I found the "proof sketches" for the main results to be very useful as well.

**[S2]** The formalism of the dual operators and the isometry property are quite elegant. The authors build on previous work but present the previously-studied concepts in an insightful way, and clearly lay out the underlying structure produced by the augmentation distribution.

**[S3]** The main results in sections 3 and 4 go further than previous work in characterizing how much information the augmentations alone can provide about the function of interest. In particular, Theorem 1 shows how the error of a function decomposes into approximation and estimation error, and furthermore identifies how the approximation error component depends on the "trace gap"; Theorem 2 then shows that finding the empirical top eigenspace can lead to small approximation error even if $\lambda_{d+1}$ and $\lambda_d$ are close.

**[S4]** The analysis of the augmentation complexity of real augmentations is quite interesting, both for the theoretical values for particular masking strategies and for the empirical estimation of the augmentation complexity of real augmentations used for masked language modeling.

**Weaknesses:**

**[W1]** This work focuses on analyzing the effect of the augmentations alone, without considering the impact of inductive biases. I understand that this was an intentional choice, and I do think that this makes the analysis of augmentations more "pure". However, such an approach means that the theoretical results here may not have much explanatory power for real augmentations.

~~Relatedly, I think the paper makes some unjustified claims regarding the power of their analysis for tasks where inductive biases are important.~~

- ~~(a) The authors state that they "disentangle the effects of the model and the augmentation", but it's not clear to me what is being disentangled. This seems to be primarily an analysis of the augmentation distribution, with the model's role being just to achieve a good ratio trace with respect to those annotations. I'm not sure how easy it would be to extend this analysis to models with more structured inductive biases.~~
- ~~(b) In the appendix, the authors state that their analysis "clearly refutes the claim in Saunshi et al. (2022) that learning guarantees independent of the model inductive bias are necessarily vacuous", but I disagree with this claim and don't think it is justified (see questions below).~~

*Edit: The authors have addressed the unjustified claims. Also, I think the theoretical contributions of studying augmentations alone are valuable even if they don't fully explain the success of large-scale self-supervised learning, so this isn't a major concern.*

**[W2]** The analysis starts from a strong assumption that the target function $f^*$ lies in a particular RKHS and satisfies a particular soft invariance condition. However, it's difficult to tell what this assumption actually means, or whether it would be true in practice. In particular, the RKHS membership seems quite strict, and adding even an arbitrarily small perturbation to $f^*$ might cause the assumption to no longer hold. (There is a bit of discussion in the appendix, but it's not clear to me what effect this would actually have on the main results.)

*Edit: The authors have noted that assuming functions lie in a particular RKHS is standard for learning theory work and necessary to get statistical guarantees.*

~~**[W3]** I think part of Proposition 1 is incorrect, or at least incorrectly stated. I also have a small concern about the asymptotic statements concerning Theorem 2. (These should hopefully be minor issues that are easy to address.)~~

*Edit: Addressed in the revision.*

**[W4]** The empirical results about real augmentations suggest that the true augmentation complexity of real-world datasets is *massive* (on the order of $\kappa \approx 10^{100}$). As such, it seems likely that the bounds in Theorem 1 and Theorem 2 would be effectively vacuous for real-world data.

*Edit: As the authors point out, this is analogous to the standard curse of dimensionality for bounds based on model complexity, so it's not specific to their analysis.*

**Questions:**

**On inductive biases.** In the appendix, you state "the nonparametric analysis established in this work clearly refutes the claim in Saunshi et al. (2022) that learning guarantees independent of the model inductive bias are necessarily vacuous". I don't think this is true. As I understand it, Saunshi et al. show that when augmentations are approximately disjoint, the eigenspectrum of $\Gamma\Gamma^*$ decays extremely slowly, with many of the eigenvalues of $\Gamma\Gamma^*$ being close to 1 (or, under their notation, many eigenvalues of the normalized Laplacian $L_\circ = I - A_\circ$ being close to 0). This means that the approximation error of the top-d eigenspace is necessarily very large. I think this same analysis also applies to your results; Saunshi et al.'s approximately-disjoint augmentations would imply $\lambda_{d+1} \approx 1$ and so your $\tau \approx 1$, leading to a blowup in the first term of your equation 14.

If I'm correct, I think this deserves more discussion, and I don't think you can say that you've refuted the claims of Saunshi et al. I also think the paper could be improved by giving more context about the relationship of your results to those of Saunshi et al. For instance, is there any relationship between your augmentation complexity and Saunshi et al.'s approximately-disjoint augmentations? One might expect that approximately-disjoint augmentations would imply a high augmentation complexity; if so, that could be useful to discuss.

Also, what do you mean by "disentangle the effects of the model and the augmentation"? It's not clear to me how the presented results disentangle these effects, although I do think they give a clear explanation of the role of the augmentations in particular.


**Interpreting the soft invariance criterion.** Can you give any intuition about what satisfying Assumption 1 actually means in practice? It seems unlikely to me that functions of practical interest could be *exactly* represented as expectations of some function $g$ on augmentations, and I also don't see any way to verify whether or not Assumption 1 holds.

**Part (iv) of Proposition 1.** Proposition 1 part (iv) states that $g^*$ satisfies Assumption 1. This doesn't make sense to me; Assumption 1 is a statement about $f^*$, not $g^*$. Do you mean to say that $g^*$ satisfies Equation (1)?

The proof of part (iv) also seems potentially incorrect. In particular I'm not sure how $\langle g^*, (I - \Gamma \Gamma^*) g^* \rangle_{P_A} \le \epsilon ||g^*||^2_{P_A}$ implies $\langle g_0, (I - \Gamma \Gamma^*) g_0 \rangle_{P_A} + ||g_1||^2_{P_A}  \le \epsilon (||g_0||^2_{P_A} + ||g_1||^2_{P_A})$; shouldn't $||g_1||^2_{P_A}$ be $\langle g_1, (I - \Gamma \Gamma^*) g_1 \rangle_{P_A}$ instead? It also seems like your proof here implies that $R(\Gamma^*) \subseteq \mathcal{F}_B(\Gamma; \epsilon)$ for all $\epsilon$. But that doesn't seem like it could be true; perhaps I'm misunderstanding something.

**Asymptotic behavior of Theorem 2.** In Theorem 2, don't the quantities $\overline{\lambda}_d^{-1}$ and $\gamma_G$ potentially depend on $N$? It seems like that could affect the asymptotic behavior of the bound in equation (20). Can you say anything about the convergence or boundedness of these quantities?

**Relationship of this analysis to that of Johnson et al. (2023).** I noticed that parts of your analysis resemble some of the statements of Johnson et al. (2023) at a high level; in particular, your soft invariance assumption is similar to their "approximate view invariance" assumption, and the decomposition in Theorem 1 is somewhat similar to their bound in Proposition 4.2. The details are different (their analysis focuses on approximating functions $\mathcal{A} \to \mathbb{R}$ rather than $\mathcal{X} \to \mathbb{R}$ and doesn't include the approximation error of the encoder) but it might still be worth including a brief discussion of the similarities.

**Other suggestions / comments.**
- Section 2 states "Similar to RIP, this property clarifies the the role of augmentation: The augmentation defines a set of features (eigenfunctions) ordered by their a priori relevance to the target function." This wasn't clear to me; how does the isometry property relate the relevance of the eigenfunctions to the target function?
- The definition of $||f||\_n^2$ in section 3 doesn't seem to be used anywhere in the main paper.
- It took me a while to locate the definition of $\hat{f}$ when reading Theorem 1; perhaps you could add a reference back to equation (9) where it is defined.
- Section 4 discusses the empirical eigenspace $\hat{\mathcal{H}}\_d$ before actually defining what it is, which was a bit disorienting; it might make sense to define it at the start of section 4 instead of at the end of page 6.

---

> ### Author Response · Authors · 2023-11-17
> **Clarification**
>
> Dear Reviewer dUJ6,
>
> Thank you for your time in reviewing this paper! We are still finalizing our response to you, and will submit it within the next 10 hours. Your review has an exceptionally high quality, and we want to make sure that our response could be equally good. We are definitely looking forward to having a discussion with you.
>
> Best regards,
> Authors

---

> ### Author Response · Authors · 2023-11-18
> **Author Response (1/2)**
>
> Thank you so much for your constructive comments! We have uploaded a new version of our paper, and would like to answer your questions below.
> ### Comments
> 1. **What we mean by “disentangle the effects of the model and the augmentation”**:
> We mean that our bounds (a) are nonparametric, (b) do not depend on any model complexity or inductive bias, and hence (c) allow the user to choose any model of any size and architecture for the encoder $\hat{\Phi}$, which is what’s actually learned during upstream (see our remark in Section 3). Comparing our bounds to prior work:
>     - HaoChen et al. (2021) and Saunshi et al. (2022): Their bounds contain model complexity (e.g. Rademacher complexity) and thus depend on the model.
>     - Cabannes et al. (2023): Their bounds are nonparametric, similar to ours. However, they assumed the function class to be a subset of the RKHS of a predefined kernel $K$, which is determined by the model architecture. Thus, their results are still model-dependent.
>
> A major novelty of our work compared to prior work is removing the effect of the model and only focusing on the augmentation. We believe that this is important, because it helps “clearly lay out the underlying structure produced by the augmentation distribution”, as the reviewer also pointed out.
>
> 2. **What does $f^{\star}$ lie in a particular RKHS mean, and is this assumption reasonable?**
>     - First, we’d like to note that: Assuming the target function to belong to some (usually compact) function class is necessary for proving any nonparametric learning guarantee. Such assumptions are known as *priors* in statistical learning; the seminal works (DeVore et al.,  2004; Bauer et al., 2007) have discussed in detail why priors are inevitable for statistical guarantees.
>     - An RKHS (or more generally, a separable Hilbert space, e.g. Fischer & Steinwart (2020)) is a common choice of such a prior in most existing learning theory work. A concern is that, regardless of the definition of this Hilbert space, a small perturbation can make $f^*$ no longer belong to the space, as the reviewer pointed out. However, this has not been deemed a serious issue in learning theory community, since if the perturbation is small enough (e.g. on a subset of measure zero), then it won’t have a large impact on the statistical guarantees.
>     - **Interpreting our Assumption 1**: Our induced RKHS is determined by the augmentation, which is selected based on human prior knowledge of $f^*$. Thus, so long as the human prior knowledge is correct, “$f^*$ belonging to the induced RKHS” can be derived from the soft invariance condition. For instance, if the augmentation is image translation and we know that image semantics are invariant w.r.t. translation, then soft invariance implies that $f^*$ belongs to the RKHS induced by translation.
>
> Of course, a good model architecture also can contain human prior knowledge, like the design of convolutional networks. Cabannes et al. (2023) had a discussion on this in their Appendix D.3, where they considered linearization of one-block convolutional networks. However, in general, it is not quite clear how the inductive bias of a deeper, more complicated practical model (that cannot be linearized) can be formulated as a kernel.
>
> DeVore et al. Mathematical methods for supervised learning. 2004.
> Bauer et al. On Regularization Algorithms in Learning Theory. 2007.
> Fischer & Steinwart. Sobolev Norm Learning Rates for Regularized Least-Squares Algorithms. 2020.
>
> 3. **Regarding the claim in Saunshi et al. (2022):**
>     - We’d like to first clarify that we interpret “non-vacuous” as “meaningful”, rather than “close to empirical observations”. We do believe that our bounds, which do not contain any model complexity, are meaningful. We are definitely not claiming anything strong such as “augmentation is all you need”, and we emphasize in Appendix A that combining inductive biases of the model class can lead to tighter bounds. What we do claim is that, if the augmentation is good enough (which is a strong assumption), then we can achieve good generalization without the help of model inductive bias. This is why we disagree with Saunshi et al.’s strong claim that any guarantees independent of the inductive bias are vacuous.
>     - Regarding the disjoint augmentation example in Saunshi et al. (2022): Thank you for this great question! This example showcases a poor choice of augmentation, for which $K_X$ defined in our Eqn. (3) is zero everywhere, so surely our bounds won’t work. Their observation was that even for this very bad augmentation, we can still obtain generalization guarantees if there is a good model (feature map). To see why: Suppose there is a good feature map $\Phi$. Then, the augmentation $x \mapsto a$ is disjoint and thus is very bad. However, the augmentation $x \mapsto \Phi(a)$ is not disjoint and in fact could be quite good. And if we view $x \mapsto \Phi(a)$ as the augmentation, then our model-free bounds work again.

---

> > ### Author Response · Authors · 2023-11-18
> > **Author Response (2/2)**
> >
> > 4. **Regarding the exponential dependency of $\kappa$:**
> >     - We would like to clarify that we are not claiming our bounds to be “close to empirical observations”, as the reviewer also pointed out that $\kappa$ could have an exponential order. However, we’d like to point out that model complexity based generalization bounds cannot evade the curse of dimensionality either, which we discussed in Section 5. The model complexity is polynomial, only when the task is simple enough so that it can be done by simple models like linear models, or it can be sufficiently compressed (see the paragraph in our related work about “PAC-Bayes bounds”). Analogously, if the target function is simple enough so that it is invariant w.r.t. a very strong augmentation, then $\kappa$ is also polynomial in data dimension. However, for almost all popular big foundation models used on popular datasets today, neither model complexity based bounds nor augmentation complexity based bounds are practical. With that said, **this work introduces a class of generalization bounds that is orthogonal to classical model complexity based bounds, which we believe is an important contribution.**
> >
> > We are very grateful to the reviewer for raising so many great questions. We plan to add more discussions in Appendix A, after having a thorough discussion with the reviewer. These discussions are definitely helpful to general readers.
> > ### Questions
> > 1. **Part (iv) of Proposition 1**: We apologize for the typo in the original submission. The problem is fixed. Regarding the proof:
> >     - $\lVert g _1 \rVert _{P _A}^2$ is equal to $\langle g _1, (I - \Gamma \Gamma^*) g _1 \rangle _{P _A}$, because there is $\Gamma^* g _1 = 0$. We clarified this point in the updated pdf.
> >     - It should be $F_B(\Gamma; \epsilon) \subset R(\Gamma^*)$. The proof of (iv) only involves $f^* \in R(\Gamma^*)$, which is naturally implied by Assumption 1 (see the sentence after Assumption 1).
> >
> > 2. **Asymptotic behavior of Theorem 2**: You are right that $\gamma _G$ and $\bar{\lambda} _d$ could potentially depend on $N$, and we do not know how to remove such independence at the moment. Intuitively, to remove such dependence, we need a quantitative measurement of the “smoothness” of the real top-d eigenfunctions, where better smoothness corresponds to fewer samples. Note that this dependence on the spectrum of the pretrained encoder also appeared in previous results (e.g. Theorem 4.5 of Saunshi et al. (2022), in which $\lambda _i$ depends on the pretrained encoder $\phi$).
> >
> > 3. **Comparing to Johnson et al. (2023)**:
> >     - The soft constraint is equivalent to Assumption 1.1 in Johnson et al. (2023). We have clarified this point in the new Section 2.
> >     - Our Theorem 1 is in fact far distinct from Proposition 4.2 of Johnson et al. (2022), for the following reasons:
> >         1. Proposition 4.2 of Johnson et al. assumed that there is a pretrained upstream encoder $r _d^*$, fixing which they bound the excess risk of the linear probe. So essentially, their bound is the same as a bound of a linear model, where the inputs are $r _d^*(x)$ and the labels are the downstream labels. And note that their bound is only for the estimation error. They do not consider the approximation error because $r _d^*$ is fixed.
> >         2. Regarding our Theorem 1: First, it bounds both approximation error and estimation error. The approximation error is bounded using $\tau$, and Theorem 2 is bounding this $\tau$ when the encoder is the optimal encoder. Second, regarding the estimation error, while we also consider the linear probe, we use a completely different set of proving tools (local Gaussian and localized Rademacher complexity). Third, we are bounding the risk itself instead of the excess risk (that is we don’t assume that there is a good linear probe on top of the pretrained encoder), and our result also includes Gaussian label noise.
> >
> > We did not feel the need to add this comparison in the remark of Theorem 1 given how different the results are, but we are happy to include it in our revision if the reviewer finds it necessary.
> >
> > ### Other suggestions / comments
> > 1. Regarding RIP: Thanks for bringing this up. We have revised the remark after Eqn. (8) in the updated pdf.
> > 2. Regarding $\lVert f \rVert _n$: This notation is only used in the proof, so we have moved it to the appendix.
> > 3. Reminding $\hat{f}$ in Theorem 1: Thanks for this great suggestion! Added.
> > 4. Definition of $\hat{H} _d$ in Section 4: Problem fixed.
> >
> >
> > We hope that this response has addressed all your questions and concerns. This review has an exceptionally high quality, and we really appreciate the reviewer for their effort and expertise. We are looking forward to having further discussions with the reviewer.

---

> > > ### Comment · Reviewer_dUJ6 · 2023-11-20
> > > **Discussion**
> > >
> > > Thanks for the responses and the updated manuscript. Some follow-up comments:
> > >
> > > **Disentangling:** I see. I interpreted "disentangle the effects of the model and the augmentation" as meaning something like "separately account for the effects of model inductive biases and augmentation choices", but it sounds like you intended something like "account for the effects of augmentations without making unnecessary assumptions about the model"? I'd suggest rewording the statements in the conclusion (and appendix A) to make this a bit more explicit and to emphasize that you're focusing on the effects of the augmentation, not the effects of the model.
> > >
> > > **Relationship to Saunshi et al. (2022)** I agree you've shown that, if the augmentation is good enough, you can achieve good generalization without inductive biases. But I don't think you're correctly interpreting the claims of Saunshi et al. (2022). Looking back at their paper, I interpret their main claims to be:
> > > - "previous analyses for contrastive learning are vacuous in the disjoint augmentation setting, due to existence of bad minimizers of the contrastive loss",
> > > - there exist function-class-dependent bounds that are non-vacuous under disjoint augmentations,
> > > - in practice, real-world contrastive learning tasks may be in the disjoint augmentations regime, and the contrastive loss does not necessarily correlate with downstream performance
> > >
> > > As far as I can tell, Saunshi et al. (2022) only claim that model-agnostic bounds are vacuous "in some settings", and in particular in the context of the disjoint or approximately-disjoint augmentations regimes. I do not think they are claiming that these bounds are vacuous for *every* augmentation distribution, just that they are vacuous for *some* realistic augmentation distributions. So I think your results are still compatible with their claims.
> > >
> > > **Exponential dependency:** That's a good point, I agree model-complexity-based bounds are likely to have this problem as well in practice, and that giving tight bounds for big foundation models isn't necessary for the contribution to be meaningful.
> > >
> > > **Part (iv) of Proposition 1:** This still looks incorrect to me. I interpret part (iv) as implying that, for any $f^* \in R(\Gamma^*)$, we also have $f^* \in F(\Gamma; \epsilon)$ for all $\epsilon$, because it is of the form $f^* = \Gamma^* g^*$ for a $g^*$ that satisfies Equation 1. So that would imply $R(\Gamma^*) \subseteq F(\Gamma; \epsilon)$ for all $\epsilon$. But that doesn't make sense. The proof also seems circular to me, since it argues that Equation 1 is equivalent to something that implies Equation 1.
> > >
> > > I'm actually not sure what you're trying to show here, so maybe I've been misinterpreting it. Maybe you're trying to say something like "if $f^*$ satisfies Assumption 1 for any $g^*$, then it must also satisfy it for this particular $g^*$"? In that case, don't you need to start with $f^* \in F(\Gamma; \epsilon)$ rather than $f^* \in R(\Gamma^*)$? (It would be redundant to explicitly state that $f^* \in R(\Gamma^*)$ if you were already assuming Assumption 1, which is why I interpreted this as an arbitrary $f^* \in R(\Gamma^*)$.)
> > >
> > > Also, your proof of Proposition 1 also still references "property (v)" which has been removed.
> > >
> > > **Asymptotic behavior of Theorem 2:** Thanks, that makes sense. I'd suggest adding a qualification to your statement "the gap between $\tau$ and its optimal value is $O(N^{−1/2})$" then, maybe something like "(ignoring potential dependence of $\gamma_G$ and $\overline{\lambda}_d^{-1}$ on $N$)".
> > >
> > > **Comparing to Johnson et al. (2023)**: Thanks for the clarifications and for discussing the equivalence of the assumptions in Section 2. I  agree the theorem and proposition are quite different so I don't think adding a comparison in the remark of Theorem 1 is necessary. (I guess the closest connection to Prop 4.2 of Johnson et al. (2023) would be to use the true top-$d$ eigenspace and obtain $\tau^2 = \lambda_{d+1}$, which you discuss a bit in Section 4 already.)
> > >
> > > **Regarding RIP:** I still find the remark after Eqn. (8) to be confusing even in the new version. It seems like existing uses of the RIP are about matrices with nearly-orthonormal columns. I don't see how this connects to preserving variance, or how it relates to PCA. Also, I think the RIP needs to hold for every vector, whereas your Eqn. (8) seems to only hold for functions in $F_B(\Gamma; \epsilon)$. And it's still not clear to me why "The augmentation defines a set of features (eigenfunctions) ordered by their a priori relevance to the target function" is related to the isometry property; I agree that it's true but it seems unrelated to Equation 8.

---

> > > > ### Author Response · Authors · 2023-11-21
> > > > **Reply to Discussion**
> > > >
> > > > Thank you very much for your engagement and the follow-up discussion! We have uploaded a revision of this paper.
> > > >
> > > > 1. **Disentangling**: Thank you for this suggestion! We have revised our introduction, conclusion and Appendix A accordingly.
> > > >
> > > > 2. **Claims in Saunshi et al. (2022)**: Thank you for this remark. We agree with your interpretation of their claims and have revised our discussion in Appendix A accordingly.
> > > >
> > > > 3. **Part (iv) of Proposition 1**: We have revised Part (iv) as follow; please let us know if this statement is clear to you:
> > > > >  For any $f^* \in F _B(\Gamma; \epsilon) \subset R(\Gamma^*)$, let $f^* = \sum _i u _i \psi _i$. Define $g _0 := \sum _i \lambda _i^{-1/2} u _i \phi _i$. Then, $g _0$ must satisfy Eqn. (1), so we can choose $g^* = g _0$, in which case Eqn. (1) is equivalent to Eqn. (7).
> > > >
> > > > 4. **Asymptotic behavior of Theorem 2**: Thank you for your suggestion. We have revised the remark of Theorem 2 accordingly.
> > > >
> > > > 5. **Regarding RIP**: Since the connection to RIP is not very easy to understand, we have changed it to a connection to PCA, which we hope is more intuitive. We have revised the remark after Eqn. (8). Please let us know if the new remark is clear.
> > > >
> > > > We are very happy to have further discussions with you, thank you again!

---

> > > > > ### Comment · Reviewer_dUJ6 · 2023-11-21
> > > > > **Discussion and updated review**
> > > > >
> > > > > Thanks for making those changes and for clarifying part (iv) of Prop 1.
> > > > >
> > > > > Fairly minor suggestion: I think the statement of Part (iv) in the above OpenReview comment is clear, but I noticed that the version in the current paper revision is a bit different and unnecessarily redundant (it says "$g_0$ satisfies Eqn. (1). Choose $g^* = g_0$, then $g^*$ satisfies Eqn. (1)"), so I'd suggest using the version from the comment above instead.
> > > > >
> > > > > Based on this discussion and the changes to the paper, I've raised my score to 8.

---

> > > > > > ### Author Response · Authors · 2023-11-21
> > > > > > **Thank you**
> > > > > >
> > > > > > We have revised Part (iv) of Prop 1. Thank you for your discussion! Your feedback has really helped us improve our work and we really appreciate your time!
> > > > > >
> > > > > > Best regards,
> > > > > > Authors

---

### Official Review · Reviewer_kPdq · 2023-11-06

**Soundness:** 2 fair
**Presentation:** 2 fair
**Contribution:** 2 fair
**Rating:** 5
**Confidence:** 2

**Summary:**

The authors consider augmentation-based representation learning, where a feature map is learned from an unlabelled dataset augmented using some mechanism.
The paper provides a learning theory for this setup using a regression problem, where the regressor is linear in the learned features.

The authors consider an augmentation mechanism defined by a conditional distribution $p(a|x)$, which modifies a data point $x$ to an augmented sample $a$ stochastically.
Given this mechanism, the regression function is defined by the conditional expectation $\mathbb{E}[g(A)|x]$ of some function $g$.
For this, the authors show that the problem may be described by an RKHS, and using this framework, they investigate the role of the feature map and augmentation.

**Strengths:**

The theoretical framework introduced in the paper is interesting and original.
The authors introduce an RKHS structure to the representation learning problem. Specifically, the target function $\mathbb{E}_A[g(A)|x]$ may be considered as an element of an RKHS whose kernel is given by the augmentation-induced joint distribution .
The framework thus enables us to quantitively analyse the performance of a downstream regressor using a kernel machinery, as well as establishes a criterion of good representation (approximation to the basis of the RKHS).

**Weaknesses:**

### Clarity
A major issue is the lack of clarify. The paper is currently dense, not accessible to a general audience (as someone who is not well-versed in the area, I find the paper really challenging to read), and thus requires a major revision. Because of this, I have to admit that I have not fully understood the contribution of the paper. Specific comments are summarised below.

### Significance
There are some elements that make the evaluation of significance difficult:
* It is not clear how reasonable the assumption on the target function is. It seems that the assumption is conveniently introduced so that we can analyse the problem using a particular RKHS. Is there any motivating example for using $\mathbb{E}_A[g^*(A)|x]$ as a target? It should be more straightforward to work with (functions on ) the augmentation space $\mathcal{A}$ rather than $\mathcal{X}$, as in Johnson et al., 2023.
* The encoder has to be of the form (10), as otherwise $\hat{\Psi}$ are not included in the RKHS. Therefore, the claim that the encoder $\hat{\Psi}$ can be arbitrary is not accurate to me.
* The soft-invariance assumption is not straightforward to interpret. How does it measure "invariance"? Since $\mathbb{E}[g(A)|x]$ and $g(a)$ are functions on different sets, the invariance is not well-defined.
* The dimensionality dependency on the upper bound seems to be very pessimistic, as the authors admitted. I

Comments:
* The problem is not well presented. It would have been more helpful if the authors had provided a high-level problem description and define symbols accordingly (e.g., as in the introduction of Johnson et al., 2023). For example, how representation learning is performed is not really mentioned using symbols in the paper (e.g., $\hat{\Phi}$ is not mentioned up to page 4).
* Some symbols are loosely defined or mentioned before formally defined; e.g., $\hat{\mathcal{H}}_d$ is mentioned before the formal definition of \hat{H}_{\Gamma}.
* Presentation of technical results could be improved. For instance, Proof Sketch of Theorem 1 only decomposes Theorem 1 into two parts, repeating what is mentioned in the remark, and thus could be omitted.
* For Proposition 1, the authors might want to check Chapter 11 of "An Introduction to the Theory of Reproducing Kernel Hilbert Spaces" by Paulsen and Raghupathi.
* Using the same symbol for different density functions is confusing ($p(a)$ and $p(x)$).
* Section 5. The term "strong" augmentation is obscure. One may consider an augmentation distribution that is independent of $x$, in which case the augmentation complexity is 1. Is the dependency between $A$ and $X$ considered a measure of "strong"?

**Questions:**

* (9) requires a suitable regularisation that depends on $B$ and $\varepsilon$. In some sense, the analysis addresses a well-specified case. What happens if this is violated?
* (12): Why is $G^{-1/2}$ required?
* "As shown in Wang et al. (2022b), there is...": Is this an existence statement?

---

> ### Author Response · Authors · 2023-11-17
> **Author Response**
>
> Thank you for your review. We have uploaded a new version of our paper and we invite the reviewer to check it out. We would like to answer to your questions as follows:
>
> **Clarity**: Thank you very much for this feedback! We agree that our paper is on the denser side and needs to be improved in terms of clarity. We have updated the paper based on the comments from the reviewers, including notation clarification, added illustrative examples, and rephrasing. Please let us know if there are further changes that you would like to see; we do hope our paper provides sufficient background to be self-inclusive, and would really appreciate your feedback.
>
>
> ### Significance
> 1. **Motivation and necessity of Assumption 1**: First, we’d like to note that assuming the target function to belong to some function class is a necessity (rather than a convenience) for proving any nonparametric learning guarantee, and an RKHS is a common choice in lots of prior work. Second, in our Proposition 1 we show a duality between the eigenfunctions on $A$ and the eigenfunctions on $X$, and this essentially says that working with $A$ is equivalent to working with $X$ up to an invertible transformation. Please also refer to our discussion on Assumption 1 in Appendix A for more details.
>
> 2. **Clarification on the arbitrary encoder**: We apologize for the confusion. What we meant is that $\hat{\Phi}$ over $A$ can be an arbitrary function, while $\hat{\Psi}$ over $X$ is always defined as the average encoder of $\hat{\Phi}$ in this work. As we mentioned in Section 3, in practice during upstream people train $\hat{\Phi}$ instead of $\hat{\Psi}$, so they can use any model of any size and any architecture. We made this point clearer in Section 3 of the updated pdf.
>
> 3. **Interpreting the soft invariance**: The soft invariance is almost equivalent to Assumption 1.1 in Johnson et al. (2023), except that it adds a $\lVert g^* \rVert _{P _A}^2$ on the right so that it is homogeneous; we have clarified this in the updated pdf. Essentially, the soft invariance says that if $A, A’$ are augmented from the same original sample $X$, then the predictor would give them similar predictions. Similar conditions have appeared in lots of prior work, including Haochen et al. 2021; Saunshi et al. 2022; Johnson et al. 2023.
>     - Could the reviewer please clarify which equation they are referring to by “$E[g(A) | x]$ and $g(a)$ are functions on different sets”? We couldn’t find in which equation we are comparing functions of different spaces.
>
> 4. **Dimensionality dependency**: We’d like to reemphasize that this dependency is a common issue and not a peculiar problem of our paper, as discussed in Section 5.1 and Section 6. Specifically, note that the more well-known model complexity based generalization bounds also cannot evade the curse of dimensionality.
> ### Comments
> We would also like to thank the reviewer for the comments, and we have updated the paper accordingly. Two further clarifications:
> - **"Strong" augmentation**: the strength of an augmentation is determined by the joint distribution over $A$ and $X$, and can be quantitatively measured by the augmentation complexity introduced in this work. The reviewer pointed out that if $A$ and $X$ are independent, then the augmentation complexity is 1. This is correct, but please note that in practice any meaningful $A$ should depend on $X$ since $A$ is “augmented” from $X$, and hence the two should never be independent.
> - Regarding Chapter 11 of the book by Paulsen and Raghupathi, could you please specify which results in Chapter 11 we should focus on?
>
>
> ### Questions
> 1. **How to implement Eqn. (9) if the exact values of $B$ and $\epsilon$ are unknown**: thank you for the good question! In practice, one only needs to have an estimated upper bound $M$ of $\frac{ B }{ \sqrt{1-\epsilon} }$, and our bound of $\lVert \hat{f} - f^* \rVert$ is $O(M)$, so it will not make too much difference if $M$ has the same order as $\frac{ B }{ \sqrt{1-\epsilon} }$. This is the common practice for least-squares regression, including kernel ridge regression, where one doesn’t know the theoretical optimal regularization coefficient but uses an estimated value instead.
>
> 2. **Why is $G^{-1/2}$ required**: $G^{-1/2}$ is used for normalization. We added this clarification in the updated pdf.
>
> 3. **“As shown in Wang et al.”**: This is not an existence statement. We meant that Wang et al. showed this equation; we have edited this sentence in the updated pdf.
>
>
> We hope that this response has addressed all your questions and concerns. The reviewer writes a very detailed, thoughtful and constructive review, which we really appreciate. We welcome the reviewer to have further discussions with us.

---

> ### Comment · Reviewer_kPdq · 2023-11-17
> **Quick response**
>
> Thank you to the authors for the detailed response -- I will have a closer look.
> Please let me share a quick (and minor) response.
>
> First of all, we don't have an individual response Reviewer to dUJ6. Is this okay?
>
> Re: Chapter 11 of the book by Paulsen and Raghupathi, Section 11.2 in this chapter treats the RKHS defined by an integral operator (the range space construction), and so Proposition 11.1 is an instance of this. Also there are some superfluous notation and symbols (e.g., do we really need to display the Radon-Nikodym derivative, given that we have a concrete density function and don't use measure theoretic results?).
>
> Re: soft-invariance, the following comment is not that important, but I just want to clarify what I meant (and the reason I found the term soft-invariance a bit weird). I do not expect a response to this comment.
>
> My understanding of (1) is that this inequality controls the conditional variance of $g$ (averaged over x), and my comment is based on this interpretation.
> if a function is (exactly) invariant under some transformation, then we should expect the transformed function lies in the same set (i.e,, if $\tilde{g}$ is the transformed version of $g$, then we expect $\tilde{g} = g$ if $g$ is invariant).
> If the transformation is taking conditional expectation, then $g$ cannot be invariant since the conditional expectation maps $g$, which is a function of $a$, to a function of $x$. If $\mathcal{A}=\mathcal{X}$, I think calling the assumption soft-invariant makes sense, in that $g$ is invariant with respect to the Markov kernel (the conditional expectation); that is $g$ is soft-invariant as a function of $x$, if the distance between $\tilde{g}$ and $g$ is small, where $\tilde{g}(x) = \int g(x') P(d x'|x) $ with $ P(d x'|x) $ the augmentation distribution.

---

> > ### Author Response · Authors · 2023-11-17
> > **Thank you**
> >
> > Dear Reviewer kPdq,
> >
> > Thank you for your quick response! We are still finalizing our response to Reviewer dUJ6, which we will submit within the next 10 hours. We will read Section 11.2 of the book you mentioned more carefully, and give you another reply tomorrow. And very much thanks for your comment!
> >
> > Best regards,
> > Authors

---

> > ### Author Response · Authors · 2023-11-20
> > **Regarding Chapter 11 of the book you mentioned**
> >
> > Dear Reviewer kPdq,
> >
> > Thank you for your response. "The range of a Hilbert-Schmidt operator can form an RKHS" is a well-known result. For example, a similar theorem was stated in Theorem 4.9 of Schölkopf & Smola (2002). The purpose of Proposition 1 is to introduce to our readers the necessary background of RKHS that leads to the key isometry property.
> >
> > We have added a citation to the book you mentioned after Proposition 1.
> >
> > Best regards,
> > Authors

---

### Official Review · Reviewer_umfH · 2023-11-09

**Soundness:** 3 good
**Presentation:** 2 fair
**Contribution:** 3 good
**Rating:** 6
**Confidence:** 3

**Summary:**

The paper builds upon recent work on self-supervised representation learning to study the effect of augmentation on the generalization properties. The key insights are that 1) augmentation can be seen as introducing a natural kernel $K_X$ between the inputs, kernel which has to be estimated based on the unlabeled data, 2) the labeled data serves then to find a linear predictor, whose generalization performance is bounded by the authors, with a bound that crucially depends on some quantities related to the augmentation procedure. The main point for the proof is an assumption, called 'isometry' (w.r.t. the augmentation) by the authors, on the optimal solution. Numerical experiments on estimating these quantities are then performed.

**Strengths:**

I appreciated the goal of the paper and that it improves on the presentation of some of its competitors. The discussions and remarks were enlightening and I chuckled when reaching the Appendix (A, B) to see the anticipation of some of my questions or critiques. The article clearly involved some work and the literature review appears quite adequate. The presence of the experiments nicely complements the theoretical part, justifying in particular the interest of introducing $\kappa$ in the analysis.

**Weaknesses:**

The key proof element is Assumption 1. I wished Section 2.1 was written differently so as to improve its clarity, emphasizing the assumptions that really matter based on my understanding, that is 1) $\Gamma \Gamma^*$ is Hilbert-Schmidt 2) eq. (8) + $f^* \in R(\Gamma^*)$ rather than the more obscure (1) or Ass.1. The writing of Proposition 1 can be definitely improved.

I think the comparison with the source condition of Cabannes et al. could be more developped since the authors' assumption ($f^* \in R(\Gamma^*)$) looks quite similar. I looked for the formula corresponding to (1) in HaoChen et al. but could not find it. Also the final predictor of Definition 1 used a bound constraint on the norm, I am not sure whether it is common in some literature since I am used rather to regularized versions. These formulations require better justification I would say. I also have doubts with the comparison with the RIP.

**Questions:**

- Which formula corresponding to (1) can be found in HaoChen et al.?
- Can you quote some articles that used a bound constraint on the final predictor as you do? In general imposing constraints simplifies proofs, is this reason of your choice?
- Can you give examples of augmentations that do not lead to $\int K_X(x,x) dp(x) <\infty$? $p(a|x)=\delta_x$ for instance? Similarly, give some examples where $\Gamma \Gamma^*$ is clearly Hilbert-Schmidt (Gaussian blur I guess)? I ask this because it is not a given that $K_X(x,x)\in\mathbb{R}$ in my opinion. There are kernel integral operators that are not related to RKHSs.
- Is it true that (8) implies that $u_i \approx 0$ for $\lambda_i < 1-\epsilon$? It seems to me to be the case. Eq.(8) can then be interpreted as requiring that $f^*$ is mostly encoded only on the first eigenvalues of $\Gamma \Gamma^*$, in other words that it is 'simple/low dimensional', and that has little to do with RIP I would say. Please discuss further this aspect, and if possible suggest a reformulation of Sec 2.1.

I would upgrade my score if these questions and those of the other reviewers are answered.

The authors could underline that Cabanes et al. also restrict themselves to the quadratic loss despite the contrastive inspiration coming from classification, and use the arguments in Cabanes et al. to justify it through calibration.

Minor:
- p2 please add some context words, 'upstream/unlabeled', 'downstream/labeled' to simplify the reading for non-experts.
- p3 last line $\phi$ not $\phi(a)$ which are values
- p4 'the the'
- Quoting Forrest Gump, ``my mama always said'' never to start a sentence with a mathematical expression.

---

> ### Author Response · Authors · 2023-11-17
> **Author Response**
>
> Thank you for your review. We have uploaded a new version of our paper and we invite the reviewer to check it out. We would like to answer to your questions as follows:
>
> ### Comments
> 1. **Improving the clarity of Section 2.1**: Thank you for the suggestion! Please note that we have largely revised Section 2.1, including adding an example of BERT that explains the notations, clarifying Eqn. (1), revising Proposition 1, and a new remark after Eqn. (8).
>
> 2. **Comparing to Cabannes et al. (2023)**: As we mentioned in Section 6, the main difference is that Cabannes et al. (2023) assumed the function class to be a subset of $H$, which is the RKHS of a __pre-selected__ kernel $K$ (which they said is determined by the model architecture). On the other hand, the main goal of this work is to decouple the effect of model and augmentation, in order to derive model-free learning guarantees. As a result, our paper does not have a “pre-selected kernel”. The only element that is pre-selected in our work is the augmentation method, which gives rise to the kernels. This helps isolate the effect of augmentation, allowing us to investigate its precise role.
>
> 3. **The connection between our isometry property and RIP**: Thank you for this great question. Our isometry property can be considered as a counterpart for RIP in infinite-dimensional functional spaces: RIP says that in finite-dimensional vector spaces, for a matrix $M$ and a vector $v$, $Mv$ preserves most variance of $v$, and therefore the classical method PCA finds the principal components that keep the most variance. Similarly, our isometry property says that $\Gamma^* \Gamma f^*$ preserves most variance of $f^*$ (the fraction of variance that is lost is less than $\epsilon$). Regarding whether $f^*$ can be encoded by the top eigenfunctions depends on how fast the eigenvalues of $\Gamma^* \Gamma$ decay: If it is fast enough then this is true, otherwise this is false. We have largely revised Section 2.1. Please let us know if the new version is clearer.
> ### Questions
> 1. **Correspondence of Eqn. (1) in Haochen et al.**: We apologize for the confusion. We meant to say that the Laplacian over the augmentation graph was introduced in Haochen et al. (2021), and the same soft invariance condition was introduced in Johnson et al. (2023) in their Assumption 1.1. We have clarified this in Section 2.1 in the updated pdf.
>
> 2. **Bound on the RKHS norm in the estimator**: The norm bound can be considered as a capacity constraint and is necessary for proving any guarantees; for reference, please see Chapter 13 of Wainwright (2019). In general, there are two typical ways of estimating the regression function in nonparametric least-squares estimation: either restricting the optimization problem to some appropriately chosen subset (usually needs to be bounded), or adding a regularizer as in kernel ridge regression. In our particular case, using a regularizer is the same as adding a constraint on the RKHS norm, as shown in Lemma 35 of Cabannes et al. (2023).
> 3. **Unbounded kernels**: This is a good question. For our work,  $\Gamma \Gamma^*$ is Hilbert-Schmidt because $\lVert K \rVert_{\infty}$ is bounded (by $\kappa^2$). To the best of our knowledge, a bounded $\lVert K \rVert_{\infty}$ is required by most statistical guarantees of kernel methods, from classical ones to more recent results. For examples of unbounded kernels and kernels not associated with an RKHS, please see Chapter 4 of Christmann and Steinwart (2008), as well as other papers from Ingo Steinwart. However, the learning guarantees provable for unbounded kernels are very limited, and they are far beyond the scope of this work.
> 4. **Is $u _i \approx 0$ for $\lambda _i < 1 - \epsilon$?**
>     - Not necessarily, and here is a simple example: Suppose $\lambda _1 = 1$ and $\lambda _2 = 1 - \epsilon - \alpha$, for some very small $\alpha > 0$. And all other eigenvalues are 0. Then, Eqn. (8) is equivalent to $ \frac{\alpha}{1 - \epsilon - \alpha} u _2^2 \le \epsilon u _1^2 $, which holds as long as $\frac{u _2^2}{u _1^2} \le \frac{\epsilon (1 - \epsilon - \alpha)}{\alpha}$. Thus, $u _2$ can be very large, even much larger than $u _1$ if $\alpha$ is very small.
>     - Moreover, $f^*$ is not necessarily low-dimensional. For example, if $\lambda _1 = 1$ and $\lambda _2 = \lambda _3 = \cdots = \lambda _{1000} = 1 - \epsilon - \alpha$, then with the same argument, $f^*$ can have 1000 dimensions.
>
> Thank you also for the detailed writing suggestions in your minor comments. We have updated the draft accordingly.
>
> We hope that this response has addressed all your questions and concerns. The reviewer raises some great questions which we really appreciate, and we welcome the reviewer to have further discussions with us.

---

> > ### Comment · Reviewer_umfH · 2023-12-01
> >
> > Comments
> >
> > About Cabannes: no, a source assumption is meant to say the problem of recovering $f^*$ is well-posed since the optimal solution belongs (or close) to the range of the kernel integral operator. Thus assuming $f^* \in R(\Gamma^*)$ is precisely the same idea, and that has nothing to do with how the kernel is selected. You "pre-select" it by choosing your augmentation method and you assume that your problem has a chance of being solvable.
> >
> > About RIP: I disagree, RIP as underlined by another reviewer has to hold for every vector and characterizes a property of a matrix. Your assumption bears on a single vector $f^*$ and says that you hope that the augmentation procedure does not lose much of the "variance".
> >
> > Most of your assumptions correspond to putting yourself in a setup where you can solve the problem. There is no harm in that but you should be open about it.
> >
> > Questions
> >
> > Correspondence of Eqn. (1) in Haochen et al.: Ok.
> >
> > Bound on the RKHS norm in the estimator: Yes I know the two approaches are equivalent, though they require an ad-hoc choice of the norm bound which is much harder to find than the regularization parameter. How do you guess your $B$ or you $\epsilon$? Do you implement the constraint in practice?
> >
> > Unbounded kernels: Reviewer tXrm asked you the same question and you answered that "we assumed they were integrable". It is not because it is a classical tool to have bounded kernels that it means your setting allows for them. You have chosen not to work with a pre-defined kernel (e.g. Gaussian), so live with it, and prove that there are indeed cases where $K_A$ and $K_X$ are square integrable. I had suggested some choices of augmentation (Dirac and Gaussian) which should give (counter) examples, please test out those or others, but I would not be satisfied just with an answer saying that you make this strong assumption on a clearly large class of kernels because of proof conveniencies.
> >
> > Is $u_i$ small: take $f^*=u_i \psi_i$ for $i$ chosen such that $\lambda_i < 1-\epsilon$, do you get that $u_i=0$ based on (8)? I think that (8) prevents $f^*$ to be encoded mainly with the indices corresponding to the smaller $\lambda_i$, so that most of its variance is in the first eigenvalues. In your example if I take $\alpha$ large, you are precisely showing that the associated coefficient $u_2$ is quite small.

---

### Official Review · Reviewer_tXrm · 2023-11-09

**Soundness:** 3 good
**Presentation:** 2 fair
**Contribution:** 4 excellent
**Rating:** 8
**Confidence:** 3

**Summary:**

The work provides a statistical analysis of augmentation-based pretraining using the so-called augmentation operator. This operator is naturally introduced and several properties of this operator are stated. Upper and lower error bounds, which apply to arbitrary encoders, are derived. The augmentation complexity is emphasized as a central object contributing to error analysis in both learning and augmentation.

**Strengths:**

The paper gives a structured foundation for augmentation based learning. The augmentation operator is well motivated and its central role is well highlighted. The obtained error bounds show, in a good way, a separation in the contribution of the model and the augmentation. The analysis seems rigorous and, personally, I find the appearance of the spectral properties of the augmentation intriguing.

**Weaknesses:**

-I find the text very dense and compressed; and thus not easy to read. I acknowledge that the task of fitting the results and concepts in the 9 page limit is difficult. Therefore, I would suggest that the authors consider whether the text is better published in a journal with fewer page restrictions.
-Some concepts could be motivated more. That is particularly helpful for readers less familiar with the topic
-I find, that the assumptions and notations should be pointed out more clearly (see questions)

**Questions:**

General remarks

-The assumptions and notations should be pointed out more clearly. For instance

       -it seems it is assumed, but I did not see it stated explicitly, that $P_\mathcal{A}$ has a density, and that $p(a|x)$ and $p(x|a)$ exist.

       -What is A in E(g(A)|x). Similarly, X is not defined as a random variable (it is clear from the context but should be defined).

       -The notational difference between $p$ and $p(\cdot,\cdot)$ is small.

-Is $\hat{\psi}$ always assumed to be of the form (10) or is (10) just an example?

-Can you explain the choice of the distribution $P_{AX}$ a bit more? Particularly, it could be emphasized how the conditional distribution $P(\cdot |x)$ has to be given. You can refer to the given examples.

-Can your approach be used to build or suggest an augmentation-based pretraining? Maybe under assumptions on prior information on the distribution $P_X$?

-Proposition 1, (iv): What does it mean that g* satisfies Assumption 1? Assumption 1 is an Assumption on $f^* \in \mathcal{L}^2(P_X)$, or does the statement in (iv) refer to (1)? The proof of statement (v) is done in the proof of statement (iv); I think those parts can be moved to the proof of statement (v).

-In (1), how are A and A' drawn from $p(\cdot |x)$? A motivation for (1) would be helpful. The space $\mathcal{F}_B(\Gamma;\epsilon)$ is a central object, it would be good to motivate it more.

-Can Assumption 1 be verified; is it a restrictive assumption?

-Can $\epsilon$ or violation of Assumption 1 be inferred during the learning process?

-Definition 3: Is $G^{-1}$ well-defined, even if $d = \infty$?

-Remark after Lemma 2: What is $\|\cdot\|_n$?

-How is $d$ chosen? Equation (20) would motivate to choose $d$ such that there is a large gap between $\lambda_d$ and $\lambda_{d+1}$, but your result improves such a guess for $d$ compared to HaoChen21 and Saunshi22.

-Is $\bar{\phi}$ an eigenfunction of $\Gamma^*\bar{\Gamma}$? If so, how restrictive is the condition $\hat{\phi} = \bar{\phi}$; is this just the choice of the encoder?

-It is good to see a separation of model and augmentation in the error analysis. Do you think a combination, for instance, through augmentation based on the model, can be fruitful, too?

-Does Lemma 9 need a condition on boundedness of the matrix $QDQ^T$ or absolute convergence of the sum $\sum\limits_{i = 1}^\infty \lambda_i q_i^T q_i$?


Minor comments:

-Did you compute the ``sweet spot" mentioned in the introduction? Can it be computed?

-In Section 2.1 some reference for conditional expectation and RKHS could be added.

-Proof of Proposition 1: The proof should also contain showing that $\mathcal{H}_\Gamma$ is complete.

-Are $K_X$ and $K_A$ square integrable? It seems so from (4) and necessary from the proof of the statement that $\Gamma^*\Gamma$ and $\Gamma\Gamma^*$ are integral operators. Are conditions on P_AX$ needed that (4) holds?

-Below Definition 2: I don't think that convexity of $D_{\chi^2}$ is needed because the bound $S_\lambda\leq \kappa^2$ follows from (11), via $\kappa^2 \geq K_X(x,x) = S_\lambda$.

-First sentence of Section 3: Assumption 1 could be mentioned to recall the bound for $\|f^*\|$ in $H_{\Gamma}$.

-Lemma 5: Below Lemma 5, it is mentioned that the required 4th order bound holds. Why is the condition for such a bound included in the statement of the lemma?

-Around Definition 2 you can refer to Examples 1,2,3 for examples of the augmentation complexity.

-Are there exponential lower bounds on $\kappa$ in Examples 2,3?

Typo: Page 4: "the the", the word ``the" was used two times

---

> ### Author Response · Authors · 2023-11-17
> **Author Response**
>
> Thank you so much for your detailed review! We have uploaded a new version of our paper which we invite you to check out.
>
> ### General remarks
> 1. **Notations**: We added a concrete example in Section 2, which clarifies how $P _{AX}$ can be chosen. Regarding your questions:
>     - $P _A$, $p(a|x)$ and $p(x|a)$: In Section 2.1, paragraph 2: “Data augmentation induces a joint distribution $P _{AX}$, with marginal distributions $P _A$ and $P _X$”. Thus, $P _A$ exists from marginalization, similarly for $p(a|x)$ and $p(x|a)$.
>     - $A$ and $X$: All capital letters (in normal font) are random variables; we have added this clarification.
>     - $p$: In the updated pdf, we changed $p(x)$ to $P _X (x)$, and $p(a)$ to $P _A (a)$. The remaining $p(\cdot)$ should be clear from context.
>     -  $A, A’, F _B(\Gamma; \epsilon)$: $A, A’$ are two augmentations from the same original sample $X$. $F _B(\Gamma; \epsilon)$ comes from the soft invariance similar to prior work.
> 2. **Definition of $\hat{\Psi}$**: We would like to clarify that what we mean by “arbitrary” in Section 3 is that $\hat{\Phi}$ can be an arbitrary function; And in this work $\hat{\Psi}$ is always the average encoder of $\hat{\Phi}$ defined by Eqn. (10). As we mentioned before Eqn. (10), in practice during upstream people train $\hat{\Phi}$ instead of $\hat{\Psi}$, so they can use an arbitrary encoder of any size and architecture.
> 3. **Suggesting new augmentations**: Our method can compare between augmentations, but new augmentations depend on human prior knowledge of the downstream task.
> 4. **Whether Assumption 1 can be verified**: While it is not practically easy to find the exact value of $\epsilon$ to verify Assumption 1, we noted after Eqn. (8) that the exact value of $\epsilon$ does not affect the fact that the top-d eigenfunctions is optimal.
> 5. **Is $G^{-1}$ well defined**: $\hat{\Phi}$ is explicitly assumed to be full-rank (see the paragraph above Definition 3), so $G^{-1}$ is well-defined. We have added clarifications in the updated pdf.
> 6. **What is $\lVert \cdot \rVert _n$**: This notation is not used in the main body, but only in our proof. We have moved this to the appendix.
> 7. **Choose d with a large gap between $\lambda _d$ and $\lambda _{d+1}$**: We would like to clarify that a large gap between $\lambda _d$ and $\lambda _{d+1}$ is not necessary for our results. In fact, this is one of our major improvements over Haochen et al. (2021) and Saunshi et al. (2022). Their bounds depend on $(\lambda _d - \lambda _{d+1})^{-1}$. However, in reality $\lambda _d$ decays quite fast, so this gap is usually much smaller than $\lambda _d$. e.g. see Figure 1 of https://openreview.net/pdf?id=ByxY8CNtvr. Meanwhile, our bound does not depend on this gap but only $\lambda _d^{-1}$. Also, we are not suggesting practitioners to “guess for $d$” using our bounds. After all, our results are just upper bounds, not the real test error.
> 8. **Regarding $\hat{\phi} = \bar{\phi}$**: In Section 4, we study the special case $\hat{\phi} = \bar{\phi}$, the Monte-Carlo approximation of the optimal encoder. This is not a restriction, but rather an interesting special case because many popular self-supervised learning methods amount to this, as pointed out in Appendix C.
> 9. **Combining augmentation and model**: Yes, combining the augmentation complexity based bounds in this work, and previous model complexity based bounds, can lead to tighter bounds, as we discussed in Appendix A.
> 10. **Lemma 9**: $Q D _{\lambda} Q^{\top}$ is a $d \times d$ matrix where $Q, D$ are both bounded entrywise, hence the product $Q D _{\lambda} Q^{\top}$ is also bounded.
> ### Minor comments
> 1. **How to find the “sweet spot” in augmentation strength**: We consider this as mainly an empirical observation determined by validation, demonstrated in Figure 3(c).
> 2. **Reference for RKHS**: Added in Section 2.2.
> 3. **Are $K_X, K_A$ square integrable**: Yes: $K _X$ is assumed to be square integrable after Eqn. (4), and $K _A$ is square integrable due to Proposition 1.
> 4. **Fourth moment condition in Lemma 5**: Lemma 5 is a general result for any $\hat{\phi}$, so it needs the fourth moment condition. Lemma 6 and Theorem 2 considers the special case $\hat{\phi} _i = \bar{\phi} _i$, and the fourth moment condition holds for this special case. We have added this clarification.
> 5. **Exponential lower bounds for $\kappa$**: There is no exponential lower bound. $\kappa$ can be polynomial if the augmentation is very strong; for example, in supervised pretraining where each sample is “augmented” to its upstream class.
>
> We have also adopted your other suggestions, including updating (iv) and the proof of proposition 1, referring to examples 1-3 around Definition 2, revising the sentence after Definition 1, recalling $\lVert f^* \rVert$ in Section 3, etc.
>
> Thank you again such a detailed, thoughtful and constructive review. We hope that we have addressed your questions and concerns, and welcome any further discussions.

---

> ### Comment · Reviewer_tXrm · 2023-11-21
> **Response to authors**
>
> My comments and remarks have been well addressed. Thank you to the authors.
>
> The notations have been adapted and clarified. My questions have been mostly answered. I still have three (whose responses do not need to be included in the paper)
>
> 1. Concerning "Suggesting new augmentations": Thank you for your response. I would like to continue shortly on this subject. Once one is able to compare between different augmentations one can choose the best. This is why I was wondering if your bounds give rise to suggestions for augmentation, too.
>
> 2. I find particularly interesting one of the concerns of reviewer dUj6 on the condition about where $f^*$ lives. I have read the authors' response to the question by reviewer dUj6 and it made me wonder if the case $f^* \notin H$ just adds an additional simple term to the error analysis (such as $err(f^*,…) = err_H(f,…) + \vert\vert f-f^*\vert\vert_V$ where $V$ is a space with less regularity (making it more likely to have $f^* \in V$ than having $f^* \in H$) and $f$ is a (well-chosen) element in $H$.
>
> 3. Why would the authors not suggest using their analysis to “guess d”? Does it relate to tightness of the error bounds?

---

> ### Author Response · Authors · 2023-11-21
> **Author Response**
>
> Thank you for your discussion! We are glad that our previous response has answered most of your questions. We would like to respond to your new comments as follows:
> 1. **Suggesting new augmentations**: This is a great question. Let us provide a rough argument: For example, our analysis could explain why patchifying and discretizing an image into tokens (such as in ViT) is a good idea compared to pixel-level augmentation. For MAE, if we directly mask an image at the pixel level, then this augmentation is almost disjoint (as pointed out by Saunshi et al. 2022), and thus is quite bad. With patchifying and discretizing, we can make the masking augmentation less disjoint and thereby lower its augmentation complexity, leading to better generalization. Similarly, this argument could explain why discretizing techniques such as VQ-VAE or VQ-GAN can be helpful. Fully justifying this argument would require an extensive empirical study, which we leave to a future paper.
>
> 2. **What if $f^{\star}$ does not belong to $H$, but is very close to $H$?** We have a discussion on this question in Appendix A. Here we’d like to provide a more detailed discussion. The short answer to your question is: Assuming $f^*$ belonging to some function class is inevitable, but it is indeed possible to relax $f^* \in H$ such that $f^*$ belongs to some larger class, under some extra assumptions.
>     - First, we have to assume that $f^*$ belongs to *some* function class, which usually requires $\lVert f^* \rVert _{\infty}$ to be bounded. Only assuming $f^*$ is close to $H$ is not sufficient for proving any bound. For example, for $f^* \in H$, suppose we add a tiny perturbation $f^* \rightarrow \tilde{f}^*$, such that $\tilde{f}^*$ is unbounded on a subset $X _0$ whose measure is very small but positive. Then, even though $\lVert f^* - \tilde{f}^* \rVert _{L^2(P _X)}$ can be very small, we still cannot prove any bound unless no training sample belongs to $X _0$, whose asymptotic probability is 0 when there are infinitely many samples. Basically, a single training sample from $X _0$ would have an uncontrollable effect on the predictor.
>     - Second, relaxing “$f^* \in H$” is possible: We can assume that $f^*$ belongs to some larger function class. One such relaxation was done by Fischer & Steinwart (2020), who considered the “power spaces” $H^p$. The smaller $p$ is, the larger $H^p$ will be. And for $p < 1$, $H^p$ is not necessarily an RKHS, but still a separable Hilbert space. The reviewer could intuitively think of $H^p$ as the RKHS of $K^p$, assuming that $H^p$ is an RKHS. Then, Fischer & Steinwart proved tight statistic bounds under the assumptions that $f^* \in H^p$, the eigenvalues of $H$ decay fast enough, and $\lVert f^* \rVert _{\infty}$ is bounded (this is still inevitable).
>
> From a theoretical perspective, $f^* \in H$ is indeed relaxable, but we feel that such relaxation would provide little extra help for people to understand the role of augmentation. That’s why in this work, we just assume that $f^* \in H$.
>
> Fischer & Steinwart. Sobolev Norm Learning Rates for Regularized Least-Squares Algorithms. 2020.
>
> 3. **Why we are not suggesting to “guess d”**: In this work, we show the following trade-off controlled by $d$: A smaller $d$ leads to larger approximation error because the encoder has lower representational capacity, but a larger $d$ leads to larger estimation error because the encoder contains more noise, and thus gives rise to a higher downstream sample complexity. We do not claim that we can estimate the best $d$ that minimizes `real approximation error + real estimation error`, because what we have proved are upper bounds, and the $d$ that minimizes `approximation error bound + estimation error bound` need not to be the optimal $d$, even if the bounds are tight.
>     - In statistical learning, “an upper bound is tight” is usually in a minimax sense, which means that one can prove a lower bound by constructing an example of a very bad task, and the upper bound is tight if it coincides with this lower bound on this worst-case example. But real tasks are rarely worst-case, so the upper bound can rarely match the real test error on a real task. This is why we cannot claim that we can estimate the best $d$ just with our upper bounds, even if they are extremely tight.
>
> We are very happy to have further discussions with you, thank you again!

---

### Author Response · Authors · 2023-11-17
**[Author Response] General Message**

Dear Reviewers and AC,

We would like to thank all our reviewers for their time and effort in reviewing our paper. We really appreciate that we have got excellent reviewers who provide thoughtful comments and helpful feedback. Based on the reviews, we have made a number of improvements to our paper, and have uploaded a new pdf. We also welcome all reviewers to have further discussions with us. Just a kind reminder that **the discussion deadline is Nov 22nd.**

Best regards,
Authors

---

### Meta-Review · Area_Chair_eGuS · 2023-12-05

**Metareview:**

The reviewers provided thorough reviews for this paper and engaged in discussions among themselves and with the authors. After much deliberation they settled for a clear accept and I will honour this process by following their recommendation.

**Justification For Why Not Higher Score:**

There are a couple of more critical reviews and I believe that a spotlight is the right level for this paper.

**Justification For Why Not Lower Score:**

There are a some very positive reviews and I believe that a spotlight is the right level for this paper.

---

### Decision · Program_Chairs · 2024-01-16

Accept (spotlight)